# Deep Generative Models for Offline Policy Learning: Tutorial, Survey, and Perspectives on Future Directions

**Jiayu Chen**                                                             *chen3686@purdue.edu*
*Purdue University*
*West Lafayette, IN 47907*

**Bhargav Ganguly**                                                        *bganguly@purdue.edu*
*Purdue University*
*West Lafayette, IN 47907*

**Yang Xu**                                                                *xu1720@purdue.edu*
*Purdue University*
*West Lafayette, IN 47907*

**Yongsheng Mei**                                                          *ysmei@gwu.edu*
*The George Washington University*
*Washington, DC 20052*

**Tian Lan**                                                               *tlan@gwu.edu*
*The George Washington University*
*Washington, DC 20052*

**Vaneet Aggarwal**                                                        *vaneet@purdue.edu*
*Purdue University*
*West Lafayette, IN 47907*

**Reviewed on OpenReview:** *https://openreview.net/forum?id=Mm2cMDl9r5*

## Abstract

Deep generative models (DGMs) have demonstrated great success across various domains, particularly in generating texts and images using models trained from offline data. Similarly, data-driven decision-making also necessitates learning a generator function from the offline data to serve as the policy. Applying DGMs in offline policy learning exhibits great potential, and numerous studies have explored in this direction. However, this field still lacks a comprehensive review and so developments of different branches are relatively independent. In this paper, we provide the first systematic review on the applications of DGMs for offline policy learning. We cover five mainstream DGMs, including Variational Auto-Encoders, Generative Adversarial Networks, Normalizing Flows, Transformers, and Diffusion Models, and their applications in both offline reinforcement learning (offline RL) and imitation learning (IL). Offline RL and IL are two main branches of offline policy learning and are widely-adopted techniques for sequential decision-making. Notably, for each type of DGM-based offline policy learning, we distill its fundamental scheme, categorize related works based on the usage of the DGM, and sort out the development process of algorithms in that field. In addition, we provide in-depth discussions on DGMs and offline policy learning as a summary, based on which we present our perspectives on future research directions. This work offers a hands-on reference for the research progress in DGMs for offline policy learning, and aims to inspire improved DGM-based offline RL or IL algorithms. For convenience, we maintain a paper list on https://github.com/LucasCJYSDL/DGMs-for-Offline-Policy-Learning.

## Contents

# 1 Introduction

Offline Policy Learning is a machine learning discipline that leverages pre-existing, static datasets to learn effective policies for sequential decision-making or control. This paper covers its two primary branches: Offline Reinforcement Learning (Offline RL) and Imitation Learning (IL). Offline RL utilizes a pre-compiled batch of experience data collected from other policies or human operators, typically comprising a series of state-action-reward-next state tuples. The objective of offline RL is to develop a policy that can maximize the expected cumulative rewards, which may necessitate deviating from the behavior patterns observed in the training data. On the other hand, IL trains a policy by mimicking an expert's behaviors. The data used for IL should be demonstrated trajectories from experts. These trajectories, containing experts' responses to various scenarios, usually consist of a sequence of state-action pairs (Learning from Demonstrations, LfD) or state-next state pairs (Learning from Observations, LfO).

Generative models are a class of statistical models that are capable of generating new data instances. They learn the underlying distribution of a dataset and can generate new data points with similar characteristics. Deep generative models (DGMs) are a subset of generative models that leverage the powerful representational capabilities of deep neural networks to model complex distributions and generate high-quality, realistic samples. Notably, the application of DGMs in computer vision (CV) and natural language processing (NLP) has been highly successful. Examples include text-to-image generation using Diffusion Models (Stable Diffusion (Rombach et al. (2022))), text-to-video generation with Diffusion Models and Transformers (Sora (Brooks et al. (2024))), large language models based on Transformers (ChatGPT (OpenAI (2023))), and so on. Offline policy learning bears similarities to CV and NLP, as all these domains involve learning from a set of offline data. While CV or NLP models generate images or texts based on given contexts, offline RL/IL models produce actions or trajectories conditioned on task scenarios. This similarity suggests that applying DGMs to offline policy learning could potentially replicate the success witnessed in CV and NLP, leveraging their capability to model and generate complex data patterns.

Great advances have been made in both DGMs and offline policy learning, and there are numerous works on applying state-of-the-art DGMs to aid the development of offline policy learning. This paper provides a comprehensive review of DGM-based offline policy learning. Specifically, the main content is categorized by the type of DGMs, and we cover nearly all mainstream DGMs in this paper, including Variational Auto-Encoders, Generative Adversarial Networks, Normalizing Flows, Transformers, and Diffusion Models, where Transformers and Diffusion Models are representatives of autoregressive and score-based generative models, respectively. For each category, we present related works in both offline RL and IL as two subsections. In each subsection (e.g., VAE-based IL in Section 3.3), we abstract the core schemes of applying that DGM in IL or offline RL, categorize related works by how the DGM is utilized, and provide a summary table outlining the representative works with their key novelties and evaluation benchmarks. It is noteworthy that our paper is more than a survey: (1) For the introduction of each work, we include its key insights and objective design, to help readers get the knowledge without referring to the original papers. (2) For each group of works (e.g., Section 3.2.4 and 7.3.2), algorithms are introduced in different levels of details. The most representative ones are highlighted as a tutorial on that specific usage of DGMs for offline policy learning, while subsequent extension works are discussed more briefly to provide a comprehensive overview. (3) We present research works within the same group using unified notations, aiming to clarify the relationships between their objectives and the evolution of algorithm designs.

The main content begins with background on offline policy learning, highlighting existing challenges that can be addressed by DGMs. Notably, different DGMs could adopt distinct base offline RL/IL algorithms. For instance, three main branches of offline RL: dynamic-programming-based, model-based, and trajectory-optimization-based offline RL, are introduced and they are mainly used in Section 2.1.1, 2.1.2, 6.2.1, respectively. Then, in the order of Variational Auto-Encoders, Generative Adversarial Networks, Normalizing Flows, Transformers, and Diffusion Models, we present the background on each DGM and its application for offline policy learning, in Section 3 - 7. For the background of each DGM, we introduce its mathematical basics and specific model variants that are utilized in offline policy learning, such that necessary knowledge can be conveniently looked up. Following the main content, in Section 8, we present in-depth discussions on the main topic of this paper and provide perspectives on future research directions. For discussions (Section

8.1), we analyze both the common and unique usages of different DGMs in offline policy learning, summarize the seminal works in each category and the issues of offline policy learning that have been targeted by DGM-based algorithms. For the future works (Section 8.2), we offer our perspectives on potential research directions across four aspects: data, benchmarking, theories, and algorithm designs.

The main contributions of this paper are as follows: (1) This is the first review paper on Deep Generative Models for Offline Policy Learning. (2) This paper covers a wide array of topics, including five mainstream Deep Generative Models and their applications in both Offline Reinforcement Learning and Imitation Learning. (3) Throughout this paper, we distill key algorithmic schemes/paradigms and selectively highlight seminal research works, as tutorials on respective topics. (4) Our work showcases the evolution of DGM-based offline policy learning in parallel with the progress of generative models themselves. (5) The summary provided at the end of this paper offers valuable insights and directions for future developments in this field.

## 2  Background on Offline Policy Learning

In this section, we introduce two main branches of offline policy learning: offline reinforcement learning (RL) and imitation learning (IL). The purpose of this part is not to thoroughly review these two areas but to provide necessary background for the applications of DGMs in offline policy learning. At the end of this section, we discuss challenges in offline policy learning that can potentially be addressed by DGMs.

All content is grounded in the fundamental framework of Markov Decision Process (Puterman (2014)), denoted as $\mathcal{M} = (\mathcal{S}, \mathcal{A}, \mathcal{T}, r, \rho_0, \gamma)$. $\mathcal{S}$ is the state space, $\mathcal{A}$ is the action space, $\mathcal{T} : \mathcal{S} \times \mathcal{A} \times \mathcal{S} \to [0, 1]$ is the transition function, $\rho_0 : \mathcal{S} \to [0, 1]$ is the distribution of the initial state, $r : \mathcal{S} \times \mathcal{A} \to \mathbb{R}$ is the reward function, and $\gamma \in [0, 1]$ is the discount factor.

### 2.1  Offline Reinforcement Learning

Here, we discuss several categories of offline RL algorithms, including dynamic-programming-based, model-based, and trajectory-optimization-based ones. In particular, dynamic-programming-based offline RL has been broadly combined with VAEs, Normalizing Flows, and Diffusion Models for improvements; Model-based offline RL is mainly used with GANs; Trajectory Optimization can be implemented using Transformers or Diffusion Models.

#### 2.1.1  Dynamic-Programming-based Offline Reinforcement Learning

Dynamic-programming-based offline RL is a main branch of model-free offline RL as detailed in (Levine et al. (2020)). Given an offline dataset $D_\mu$ collected by the behavior policy $\mu(a|s)$, dynamic-programming-based offline RL usually would adopt a constrained policy iteration process as below: ($k$ denotes the index of the learning iteration.)

$$Q_{k+1}^\pi = \arg\min_Q \mathbb{E}_{(s,a,r,s')\sim D_\mu} \left[ Q(s,a) - \left( r + \gamma \mathbb{E}_{a'\sim\pi_k(\cdot|s')} Q_k^\pi(s',a') \right) \right]^2 ;$$
$$\pi_{k+1} = \arg\max_\pi \mathbb{E}_{s\sim D_\mu} \left[ \mathbb{E}_{a\sim\pi(\cdot|s)} Q_{k+1}^\pi(s,a) \right] \ \ s.t. \ \mathbb{E}_{s\in D_\mu} \left[ D(\pi(\cdot|s), \mu(\cdot|s)) \right] \leq \epsilon. \tag{1}$$

Here, the first and second equations are referred to as the policy evaluation and policy improvement steps, respectively. When updating the Q function, $(s, a, r, s')$ are sampled from $D_\mu$ but the target action $a'$ is sampled by the being-learned policy $\pi_k$. If $\pi_k(a'|s')$ differs substantially from $\mu(a'|s')$, out-of-distribution (OOD) actions, which have not been explored by $\mu$, can be sampled. Further, the Q-function trained on $D_\mu$, i.e., $Q_{k+1}^\pi$, may erroneously produce over-optimistic values for these OOD actions, leading the policy $\pi_{k+1}$ to generate unpredictable OOD behaviors. In online RL, such issues are naturally corrected when the agent interacts with the environment, attempting the actions it (erroneously) believes to be good and observing that in fact they are not. In offline RL, interactions with the environment are not accessible, but, alternatively, the over-optimism can be controlled by limiting the discrepancy between $\pi$ and $\mu$, as shown in Eq. (1). The discrepancy measure $D(\cdot||\cdot)$ has multiple candidates. For a comprehensive review, please refer to (Levine et al. (2020)). Built upon this basic paradigm (i.e., Eq. (1)), we introduce some practical and representative offline RL algorithms that focus on addressing the issue of OOD actions.

**Policy constraint methods**, such as AWR (Peng et al. (2019b)) and AWAC (Nair et al. (2020)), choose the KL-divergence as the discrepancy measure, for which the policy improvement step in Eq. (1) has a closed-form solution:

$$\pi_{k+1} = \arg\max_{\pi} \mathbb{E}_{s \sim D_{\mu}} \left[ \mathbb{E}_{a \sim \pi(\cdot|s)} Q_{k+1}^{\pi}(s,a) - \lambda(D_{KL}(\pi(\cdot|s)||\mu(\cdot|s)) - \epsilon) \right]$$
$$\Rightarrow \pi_{k+1}(a|s) = \mu(a|s) \exp\left( Q_{k+1}^{\pi}(s,a)/\lambda \right) / Z(s) \tag{2}$$

In this expression, $1/Z(s)$ serves as the normalization factor across all possible action choices at state $s$; $\lambda$ is the Lagrangian multiplier to convert the original constrained optimization problem to an unconstrained one, typically set as a constant rather than being optimized. In practice, such an policy can be acquired through weighted supervised learning from $D_{\mu}$ (Wang et al. (2020)), where $\exp\left( Q_{k+1}^{\pi}(s,a) \right)$ serves as the weight for $(s,a)$, that is:

$$\pi_{k+1} = \arg\max_{\pi} \mathbb{E}_{(s,a) \sim D_{\mu}} \left[ \log \pi(a|s) \exp\left( Q_{k+1}^{\pi}(s,a)/\lambda \right) / Z(s) \right] \tag{3}$$

**Policy penalty methods**, such as BRAC (Wu et al. (2019a)), utilize approximated KL-divergence $\widehat{D}_{KL}(\pi(\cdot|s)||\mu(\cdot|s)) = \widehat{\mathbb{E}}_{a \sim \pi(\cdot|s)} [\log \pi(a|s) - \log \hat{\mu}(a|s)]$, where $\hat{\mu}$ is learned via Behavioral Cloning (introduced in Section 2.2.1) from $D_{\mu}$ to estimate $\mu$ and $\widehat{\mathbb{E}}$ denotes the estimated expectation via Monte Carlo sampling, and modify the reward function as $\tilde{r}(s,a) = r(s,a) - \lambda \widehat{D}_{KL}(\pi(\cdot|s)||\mu(\cdot|s))$ ($\lambda > 0$). In this way, the deviation from $\mu$ is implemented as a penalty term in the reward function for the policy learning to avoid deviating from $\mu$ not just in the current step but also in future steps. This method is usually implemented in its equivalent form (Wu et al. (2019a)) as below:

$$Q_{k+1}^{\pi} = \arg\min_{Q} \mathbb{E}_{(s,a,r,s') \sim D_{\mu}} \left[ Q(s,a) - \left[ r + \gamma \left( \mathbb{E}_{a' \sim \pi_k(\cdot|s')} Q_k^{\pi}(s',a') - \lambda \widehat{D}_{KL}(\pi(\cdot|s')||\mu(\cdot|s')) \right) \right] \right]^2 ;$$
$$\pi_{k+1} = \arg\max_{\pi} \mathbb{E}_{s \sim D_{\mu}} \left[ \mathbb{E}_{a \sim \pi(\cdot|s)} Q_{k+1}^{\pi}(s,a) - \lambda \widehat{D}_{KL}(\pi(\cdot|s)||\mu(\cdot|s)) \right]. \tag{4}$$

One significant disadvantage of this approach is that it requires explicit estimation of the behavior policy, i.e., $\hat{\mu}$, and the estimation error could hurt the overall performance of this approach. Instead, a recent work (Park et al. (2024)) proposes that $\mathbb{E}_{s \sim D_{\mu}} \left[ -\lambda \widehat{D}_{KL}(\pi(\cdot|s)||\mu(\cdot|s)) \right]$ can be replaced by $\mathbb{E}_{(s,a) \sim D_{\mu}} [\lambda \log \pi(a|s)]$, which implicitly constraints $\pi$ to remain close to the behavior policy. They further show through intensive empirical results that this objective design consistently outperforms aforementioned policy constraint methods (e.g., AWR). Notably, the application of Diffusion Models in offline RL (i.e., Section 7.3.1) predominantly adheres to these policy constraint methods, suggesting potential future improvement.

**Support constraint methods**, such as BCQ (Fujimoto et al. (2019)) and BEAR (Kumar et al. (2019)), propose to confine the support of the learned policy within that of the behavior policy to avoid OOD actions, because constraining the learned policy to remain close in distribution to the behavior policy, as in the previous two methods, can negatively impact the policy performance, especially when the behavior policy is substantially suboptimal. As an example, BEAR proposes to replace the discrepancy constraint in Eq. (1) with a support constraint: $\pi \in \{\pi' : \mathcal{S} \times \mathcal{A} \to [0,1] \mid \pi'(a|s) = 0 \text{ whenever } \mu(a|s) < \epsilon \}$ [1]. With such a support constraint, the target actions $a'$ used in policy evaluation (i.e., the first equation in Eq. (1)), would all satisfy $\mu(a|s) \geq \epsilon$ and so are in-distribution actions, that is:

$$Q_{k+1}^{\pi} = \arg\min_{Q} \mathbb{E}_{(s,a,r,s') \sim D_{\mu}} \left[ Q(s,a) - \left( r + \gamma \mathbb{E}_{a' \sim \pi_k(\cdot|s') \ s.t. \ \mu(a|s) \geq \epsilon} Q_k^{\pi}(s',a') \right) \right]^2 \tag{5}$$

**Pessimistic value methods** regularize Q-function directly to avoid overly optimistic values for OOD actions, as an alternative to imposing (support or distributional) constraints on the policy. As a representative, CQL (Kumar et al. (2020)) removes the policy constraint in Eq. (1) and modifies the policy evaluation process as below:

$$Q_{k+1}^{\pi} = \arg\min_{Q} \mathbb{E}_{(s,a,r,s') \sim D_{\mu}} \left[ Q(s,a) - \left( r + \gamma \mathbb{E}_{a' \sim \pi_k(\cdot|s')} Q_k^{\pi}(s',a') \right) \right]^2 +$$
$$\lambda \left[ \mathbb{E}_{s \sim D_{\mu}, a \sim \pi_k(\cdot|s)} Q(s,a) - \mathbb{E}_{(s,a) \sim D_{\mu}} Q(s,a) \right] \tag{6}$$

---

[1]In practice, they use a sampled maximum mean discrepancy (MMD, Gretton et al. (2012)) between the being-learned policy and behavior policy to implement such a support constraint, of which the effectiveness is empirically justified.

Here, the second term (with $\lambda > 0$) can be viewed as a regularizer for the policy evaluation process, which minimizes Q-values under the being-learned policy distribution, i.e., $\pi_k(\cdot|s)$, and maximizes the Q-values for $(s, a)$ within $D_\mu$. Intuitively, this ensures that high Q-values are only assigned to in-distribution actions. Suppose $\widehat{Q}^\pi = \lim_{k\to\infty} Q_k^\pi$ and define $Q^\pi$ as the true Q-function for $\pi$, they theoretically prove that $\mathbb{E}_{a\sim\pi(\cdot|s)}\left[\widehat{Q}^\pi(s, a)\right] \leq \mathbb{E}_{a\sim\pi(\cdot|s)}\left[Q^\pi(s, a)\right], \; \forall s \in D_\mu$, with a high probability, when $\lambda$ is large enough. Thus, this algorithm mitigates the overestimation issue with a theoretical guarantee. However, it tends to learn an overly conservative Q function [2].

Notably, DM-based and NF-based offline RL algorithms usually adopt policy constraint and support constraint methods, respectively, as detailed in Section 7.3.1 and 5.3.1. While, VAE-based offline RL (i.e., Section 3.2.2) has applied all the aforementioned methods except for policy constraint ones. The motivation to involve DGMs in these methods is either improving expressiveness of the policy $\pi$, or providing better estimations of the policy or support constraints to avoid OOD actions.

### 2.1.2 Model-based Offline Reinforcement Learning

In model-based offline RL (Kidambi et al. (2020); Yu et al. (2020a)), a parametric dynamic model $\widehat{\mathcal{T}}(s'|s, a)$ and reward model $\hat{r}(s, a)$ need to be learned through supervised learning:

$$\max_{\widehat{\mathcal{T}}} \mathbb{E}_{(s,a,s')\sim D_\mu}\left[\log \widehat{\mathcal{T}}(s'|s, a)\right], \; \min_{\hat{r}} \mathbb{E}_{(s,a,r)\sim D_\mu}\left[(\hat{r}(s, a) - r)^2\right] \tag{7}$$

The approximated MDP $\widehat{\mathcal{M}} = (\mathcal{S}, \mathcal{A}, \widehat{\mathcal{T}}, \hat{r}, \hat{\rho}_0, \gamma)$, where $\hat{\rho}_0$ is the empirical distribution of initial states in $D_\mu$, can then be used as proxies of the real environment, and (short-horizon) trajectories can be collected by interacting with $\widehat{\mathcal{M}}$ using the being-learned policy $\pi_\theta$, to form another dataset $D_\pi$.

To mitigate the impact from the model bias of $\widehat{\mathcal{M}}$, an estimation of the model uncertainty, i.e., $u(s, a)$, is utilized as a penalty term to discourage the agent to visit uncertain regions (where $u$ is high) in the state-action space, by setting the reward at $(s, a)$ as $\tilde{r}(s, a) = \hat{r}(s, a) - \lambda_r u(s, a)$. ($\lambda_r > 0$ is a hyperparameter.) This conservative way to avoid OOD behaviors resembles the policy penalty method in dynamic-programming-based offline RL. The uncertainty measure can be acquired through Bayesian Neural Networks (Chua et al. (2018)), or modeled as disagreement among estimations from an ensemble of Q-functions or approximated environment models (Lu et al. (2022b)). Specifically, $D_\pi$ can be generated as follows: $s_0 \sim \hat{\rho}_0(\cdot), a_0 \sim \pi_\theta(\cdot|s_0), \tilde{r}_0 = \hat{r}(s_0, a_0) - \lambda_r u(s_0, a_0), s_1 \sim \widehat{\mathcal{T}}(\cdot|s_0, a_0)$, and so on. Finally, offline/off-policy RL algorithms can be applied to train the policy $\pi_\theta$ on an augmented dataset $D_{\text{aug}} = \lambda_D D_\mu + (1 - \lambda_D)D_\pi$. Here, $\lambda_D \in [0, 1]$ is another hyperparameter. A typical (off-policy) actor-critic framework for this process is as below (Levine et al. (2020)): ($Q_{\bar{\phi}}$ is the target Q-function.)

$$\max_\theta \mathbb{E}_{s\sim D_{\text{aug}}, a\sim\pi_\theta(\cdot|s)}\left[Q_\phi(s, a)\right], \; \min_\phi \mathbb{E}_{(s,a,\tilde{r},s')\sim D_{\text{aug}}}\left[(Q_\phi(s, a) - (\tilde{r}(s, a) + \gamma\mathbb{E}_{a'\sim\pi_\theta(\cdot|s')}Q_{\bar{\phi}}(s', a')))\right] \tag{8}$$

As detailed in Section 4.3, GANs have been utilized for model-based offline RL to approximate the policy or environment models.

In fact, with the approximated model $\widehat{\mathcal{M}}$, planning algorithms can be directly applied to find the optimal action sequence as below:

$$a_{0:T}^* = \arg\max_{a_{0:T}} \sum_{t=0}^{T} r(s_t, a_t) - \lambda_r u(s_t, a_t) \; s.t. \; s_{t+1} = \widehat{\mathcal{T}}(s_t, a_t) \tag{9}$$

The equation above is a planning objective for deterministic environments, which can be efficiently solved with algorithms like Model Predictive Control (Holkar & Waghmare (2010)), similarly for stochastic environments.

---

[2]According to Theorem 3.2 in (Kumar et al. (2020)), the minimum value of $\lambda$ is linked to $\max_{s\in D_\mu} \frac{1}{\sqrt{|D_\mu(s)|}}$, where $|D_\mu(s)|$ denotes the frequency of state $s$ in $D_\mu$. Given that $D_\mu$'s coverage could be limited and certain states would have low visitation frequencies, this can result in a high value of $\lambda$. However, as shown in Eq. (6), this same $\lambda$ value is used in the Q-update for every state, potentially leading to underestimation for other states in $D_\mu$.

Eq. (8) and (9) represent obtaining the policy through learning and planning, respectively. Notably, there is a third line of research that unifies planning and learning for efficient model-based offline RL, exemplified by MuZero and its variants (Schrittwieser et al. (2020); Hubert et al. (2021); Schrittwieser et al. (2021); Antonoglou et al. (2022)). The applications of DGMs have not been extensively explored in the last two model-based offline RL manners, suggesting a promising future research direction.

### 2.1.3 Trajectory-Optimization-based Offline Reinforcement Learning

Trajectory optimization casts offline RL as sequence modelling (Prudencio et al. (2023)). One approach in this category is to model the distribution of augmented trajectories $\{\tau_{0:T} = (s_0, a_0, R_0, \cdots, s_T, a_T, R_T)\}$ induced by the behavior policy $\mu$, where $R_t = \sum_{i=t}^{T} r_i$ denotes the return-to-go (RTG). Each trajectory $\tau_{0:T}$, instead of being treated as a single data point, can be modeled/generated using autoregressive models like Transformers, and the objective may take the form:

$$\max_{\pi} \mathbb{E}_{\tau_{0:T} \sim D_\mu} \sum_{t=1}^{T} \log \pi(\tau_t | \tau_{0:t-1}), \ \log \pi(\tau_t | \tau_{0:t-1}) = \log \pi^1(s_t, R_t | \tau_{0:t-1}) + \log \pi^2(a_t | \tau_{0:t-1}, s_t, R_t) \quad (10)$$

The implementation of $\pi^1$ and $\pi^2$ can vary in different algorithms. After training, $\pi^2$ can be used as the policy to generate trajectories of a target return $R_0$ in an autoregressive manner: $s_0 \sim \rho_0(\cdot)$, $a_0 \sim \pi^2(\cdot | s_0, R_0)$, $s_1 \sim \mathcal{T}(\cdot | s_0, a_0)$, $r_0 = r(s_0, a_0)$, $R_1 = R_0 - r_0$, $\cdots$. The rationale behind this is that by learning from a distribution of return and trajectory pairs, the RTG-based policy $\pi^2$ is expected to induce trajectories that correspond to the conditioned RTG.

Such a sequence modelling objective makes trajectory-optimization-based offline RL less prone to selecting OOD actions, because multiple state and action anchors throughout the trajectory prevent the learned policy from deviating too far from the behavior policy $\mu$. As shown above, the training is simply through supervised learning, which gets rid of the difficulty of dynamic programming or policy gradient optimization as in previous two categories of offline RL. Also, trajectory-optimization-based offline RL performs well in sparse reward settings, where temporal-difference methods typically fail, since they rely on dense reward estimates to effectively propagate Q-values over long horizons. Among the five DGMs that we cover in this paper, only Transformers belong to autoregressive models. Thus, we introduce representative works of autoregressive trajectory modelling and their extensions in Section 6.2. We also provide detailed discussions on the fundamental limitations of these methods in Section 6.2. For instance, to recover near optimal policies with these methods, environment dynamics (i.e., $\mathcal{T}$ and $r$ in $\mathcal{M}$) should be nearly deterministic and the offline dataset $D_\mu$ should provide good coverage of all possible return-trajectory pairs. These assumptions are somewhat strong compared to those needed for dynamic-programming-based approaches (Brandfonbrener et al. (2022)).

Alternatively, due to the capability of Diffusion Models to handle high-dimensional data, they have been utilized to directly model/generate complete trajectories $\{\tau_{0:T} = (s_0, a_0, \cdots, s_T, a_T)\}$ as data points. During execution, the target return $R_0$ can serve as a conditioner to guide the trajectory generation. This approach relies heavily on a deep understanding of Diffusion Models, and as such, the details are elaborated in the corresponding Section 7.3.2. Unlike the previous two categories of offline RL, trajectory-optimization-based offline RL significantly depends on the capabilities of DGMs, such as Transformers and Diffusion Models, establishing a closer connection with them.

## 2.2 Imitation Learning

Imitation Learning (IL, Zare et al. (2023)) aims at learning a mapping between states and actions, from a set of demonstrations $D_E$ for a certain task. $D_E$ is usually composed of state-action pairs demonstrated by domain experts. Next, we introduce the two primary categories of IL: Behavioral Cloning and Inverse Reinforcement Learning.

### 2.2.1 Behavioral Cloning

Behavioral Cloning (BC, Pomerleau (1991)) casts IL as supervised learning which is well-studied and computationally efficient. To be specific, the objective of BC is to train a policy $\pi$ with which the likelihood of the expert demonstrations can be maximized:

$$\max_\pi \mathbb{E}_{(s,a)\sim D_E} \left[ \log \pi(a|s) \right] \tag{11}$$

Despite its simplicity, BC has several significant drawbacks.

First, the BC agent is trained on states generated by the expert policy but tested on states induced by its own action. As a result, the distribution of states observed during testing can differ from that observed during training, and the agent could encounter unseen states, which is known as the covariate shift problem (Pomerleau (1988)). BC agents do not have the ability to return to demonstrated regions when it encounters out-of-distribution (OOD) states, so deviations from the demonstrated behavior tend to accumulate over the decision horizon, causing compounding errors. Here are some strategies for addressing this issue. Interactive IL, such as DAgger (Ross et al. (2011)), assumes access to an online expert which can relabel the data collected by the BC agent with correct actions, such that the agent can learn how to recover from mistakes. Alternatively, when the simulator is available, the BC agent can be trained to remain on the support of demonstrations through an additional RL process. For instance, Brantley et al. (2020) propose training an ensemble of policies, using the variance of their action predictions as a cost. The variance outside of the expert's support would be higher, because the ensemble policies are more likely to disagree on states not encountered in the demonstrations. An RL algorithm can then be applied to minimize this cost, in conjunction with supervised learning using BC. The RL process helps guide the agent back to the expert distribution, while BC ensures that the agent closely mimics the expert within that distribution.

Second, it is difficult for a supervised learning approach to identify the underlying causes of expert actions, leading to a problem known as causal misidentification (De Haan et al. (2019)). Consequently, the BC agent may fail to differentiate between nuisance correlates in the states and the actual causes of expert actions, making it challenging for the learned policy to generalize to OOD states. Strategies addressing this issue either incorporate the causal structure of the interaction between the expert and environment into policy learning, or identify and eliminate nuisance correlates in observations. We have seen a series of studies that apply VAEs to tackle causal misidentification, as detailed in Section 3.3.5.

Notably, Florence et al. (2021) investigate a fundamental design decision of BC that has largely been overlooked: the form of the policy. Empirically, they show that using explicit continuous feed-forward networks as policies, i.e., $a = \pi_\theta(s)$, struggles to model discontinuities, making policies incapable of switching decisively between different behaviors. Instead, they propose Implicit BC, where an energy model $E_\theta$ is utilized to represent the policy $\pi_\theta$. The policy definition and training method are as below:

$$a = \arg\min_{\tilde{a}} E_\theta(s, \tilde{a}), \ \max_\theta \mathbb{E}_{(s,a)\sim D_E, \{a_i^{\text{neg}}\}_{i=1}^N} \log \left[ e^{-E_\theta(s,a)} / \left( e^{-E_\theta(s,a)} + \sum_{i=1}^N e^{-E_\theta(s,a_i^{\text{neg}})} \right) \right] \tag{12}$$

$\{a_i^{\text{neg}}\}_{i=1}^N$ are generated negative examples (i.e., non-expert actions) corresponding to $(s,a)$. Such an InfoNCE-style objective (van den Oord et al. (2018)) is to approximately maximize the expectation of $\log \left( e^{-E_\theta(s,a)} / \int_{\tilde{a}} e^{-E_\theta(s,\tilde{a})} \right)$. As a result, the expert action $a$ would correspond to the lowest energy $E_\theta(s,a)$. During inference, $\arg\min_{\tilde{a}} E_\theta(s, \tilde{a})$ can be acquired through techniques like stochastic optimization, which brings higher computation cost but significantly better empirical performance in both simulated and real-world tasks. We have seen a lot of works that apply DGMs as the policy model to improve the policy expressiveness or handle complex input/output data, which are introduced in later sections, but whether such an implicit design could improve DGM-based policies remains to be explored.

### 2.2.2 Inverse Reinforcement Learning

Inverse Reinforcement Learning (IRL, Russell (1998)) involves an apprentice agent that aims to infer the reward function underlying expert demonstrations. Subsequently, a policy agent can be trained through

RL based on this inferred reward function, attempting to replicate the expert's behavior. Unlike BC, most IRL algorithms require not only offline demonstrations but also access to a simulator. By interacting with the simulator, the IRL agent can observe long-term consequences of its behavior and learn to recover from mistakes through RL, making them less sensitive to covariate shift than BC. Also, reward functions provide explanation for expert behaviors, which can help eliminate causal confusion. Traditional IRL approaches (Ng & Russell (2000); Abbeel & Ng (2004)) usually restrict the form of reward functions to be, e.g., a linear combination of manually-designed features, which may be overly simplistic for real-world tasks. Additionally, these approaches follow an iterative process that alternates between reward updating and policy training. At each iteration, the policy is trained to be near-optimal under the current reward function, so as to evaluate the utility of the reward and inform its next-iteration update, which is computationally expensive. Further, a policy can be optimal with respect to an infinite number of reward functions (Kim et al. (2021b)). This inherent ambiguity in the relationship between the policy and the reward function can pose significant challenges for reward learning.

DGMs have played a significant role in addressing aforementioned issues of IRL. (1) Within a GAN-like framework, GAIL (Ho & Ermon (2016)) and AIRL (Fu et al. (2017)) implement IRL as an adversarial learning process. They adopt a discriminator network as a pseudo reward function, which can take raw states as input and does not restrict the form of rewards. They also learn an actor network as the policy, which is concurrently trained alongside the discriminator. Intuitively, the training process involves a two-player game between the actor and discriminator. The discriminator attempts to distinguish agent trajectories from expert trajectories, while the actor endeavors to deceive its adversary by generating trajectories that closely resemble expert ones. At each iteration, both the actor and the discriminator are trained for only several epochs, as in GANs, which greatly reduces the computation cost compared to traditional IRL. Further, these algorithms are based on MaxEntIRL (Ziebart et al. (2010)) which cleanly resolves the ambiguity between the reward function and policy using the principle of maximum entropy. We push the mathematical details of MaxEntIRL, GAIL, and AIRL and their extensions to Section 4.2 (i.e., GAN-based IL). (2) As another representative work, AVRIL (Chan & van der Schaar (2021)) is based on the Bayesian IRL (Ramachandran & Amir (2007)) framework and VAEs. In Bayesian IRL, a posterior distribution over candidate reward functions is derived from an assumed prior distribution over the rewards and a likelihood of the reward hypothesis. Classical Bayesian IRL algorithms are computationally intractable in complex environments, since generating each posterior sample requires solving an entire MDP with RL. As introduced in Section 3.1, VAEs (Kingma & Welling (2014)) facilitate scalable estimation of Bayesian posteriors through variational inference, so AVRIL utilizes VAEs to implement scalable Bayesian IRL. Specifically, within a VAE, the encoder is used to approximate the posterior distribution over the rewards, and the decoder is employed to approximate the Q-value, which implicitly defines the policy. Compared to GAIL, AVRIL also offers computational efficiency and does not impose restrictions on the forms of rewards. However, its algorithmic design lacks robust theoretical grounding, and its empirical performance tends to be inferior, indicating that more future work is required. We find that most research on scalable IRL favor GAN-based approaches.

### 2.3 Challenges of Offline Policy Learning

In this section, we summarize the significant challenges faced by offline RL and IL, providing guidance for the application of DGMs in these areas. **Dynamic-programming-based offline RL** relies on accurate estimations of some key quantities, such as the approximated behavior policy $\hat{\mu}$ in policy penalty methods and the expert support in support constraint methods, to avoid OOD behaviors. Achieving these accurate estimations poses a significant challenge. On the other hand, the pessimistic value method addresses the overestimation issue common in other dynamic programming methods but can suffer from excessive pessimism, which may limit improvements beyond the behavior policy. Therefore, maintaining a balance between conservatism and optimism is crucial. **Model-based offline RL** necessitates an accurate world model $\widehat{\mathcal{M}}$ and explicit uncertainty estimation. Modeling MDPs for environments characterized by complex dynamics, high-dimensional state spaces, and long decision horizons is particularly challenging. Further, uncertainty estimation, which serves to quantify the distributional shift, can significantly affect the performance of model-based offline RL. **Trajectory-optimization-based offline RL** requires a large amount of training data that adequately covers possible return-trajectory pairs. Additionally, it remains uncertain

whether conditioning on the return-to-go ensures that a generator would produce trajectories achieving that specified return, highlighting some fundamental limitations of these methods.

**BC** can be improved by employing more expressive/advanced policy models or by addressing its inherent issues, such as covariant shift and causal misidentification. Methods that effectively mitigate covariant shift or compounding errors, without requiring online experts or simulators (i.e., relying solely on offline data), could be particularly beneficial. **IRL** is a more demanding technique as it seeks to infer not only the expert policy but also the underlying reward function. Currently, there is no algorithm that perfectly balances computational efficiency with theoretical soundness in reward recovery. GAN-based IRL, such as GAIL, often encounters challenges related to training stability and sample complexity. Most IRL algorithms depend on simulators to train policies through RL. However, this survey focuses on algorithms that operate exclusively with offline data, and as such, we do not provide an extensive review of applying DGMs to IRL.

From a data perspective, a significant challenge in offline RL is effectively utilizing unlabeled data, which does not contain reward labels and is more readily available. As an example, some VAE-based unsupervised skill discovery algorithms, such as (Eysenbach et al. (2019b); Ajay et al. (2021); Chen et al. (2023c)), attempt to extract skills from unlabeled data, which can then serve as policy segments for downstream RL. Similarly, a combination use of large amount of unlabeled data with a limited amount of high-quality labeled data represents a challenging yet promising direction for data-efficient offline RL. As for IL, learning from imperfect demonstrations can be a fruitful research direction. Given the difficulty in obtaining large numbers of optimal demonstrations from human experts, an IL agent may have to learn from demonstrations that vary in levels of optimality to ensure sufficient training data. As examples, there are two extensions of GAIL for learning from imperfect demonstrations (Wu et al. (2019b); Wang et al. (2021b)). They propose mechanisms to automatically weight demonstrations based on their quality and significance for training, and develop algorithms for learning from the weighted demonstrations.

Finally, several extended setups for offline policy learning can be developed, including safety-critical, multi-agent, multi-task, and hierarchical settings. The safety-critical setting involves numerous safety constraints that the learned policy must adhere to. Multi-task learning aims to develop a policy that can either be directly applied to a variety of tasks or quickly adapted to them. Hierarchical learning usually requires constructing a two-level policy, where the lower level consists of skills specific to subtasks and the high level manages the transition between these skills. This structure is particularly beneficial for complex, long-horizon tasks that can be decomposed into a sequence of subtasks. Throughout Section 3 - 7, we introduce a series of works that apply DGMs to these extended setups. Additionally, for IL, learning from observations (LfO) is a more challenging setup, where the agent only has access to trajectories of expert states without knowledge of expert actions. Algorithms for this setup enable challenging tasks such as learning from videos and learning from agents with different embodiments (who share the same state space but have distinct action sets). GAIfO (Torabi et al. (2018)) adapts the GAIL objective for LfO by matching the expert and agent's state-transition distributions instead, and there are several VAE-based algorithms designed for LfO as detailed in Section 3.3. However, the overall research applying DGMs to LfO remains sparse, suggesting further exploration in this direction.

In Section 8.1, we provide an in-depth discussion on offline policy learning and deep generative models (DGMs). In particular, we outline the base IL/offline RL frameworks to which each type of DGM has been applied in Table 12, and the issues or extensions that each DGM has tried to resolve in Table 13, corresponding to the challenges introduced in this section. We encourage readers interested in solving specific problems of offline policy learning to look up these tables and refer directly to the corresponding sections.

## 3 Variational Auto-Encoders in Offline Policy Learning

This section begins with background on Variational Auto-Encoders (VAEs), where we introduce their mathematical foundations and model variants with a particular focus on those employed in offline policy learning. Subsequently, we present a comprehensive overview of the applications of VAEs in offline RL (Section 3.2) and IL (Section 3.3) as two subsections. Within each subsection, we categorize related research works based on how VAEs are utilized and summarize the algorithm design paradigm as a tutorial. At the end of each

subsection, we provide a table to show the representative algorithms with their key novelties and evaluation tasks, serving as a reference for future research in algorithm design and evaluation.

The sections focusing on other generative models, i.e., Section 4 - 7, follow a similar structure with this section. Considering the extensive scope of content addressed in this work, notations utilized for different generative models are relatively independent. However, within the context of the same generative model, including its background and applications in offline policy learning, notations are consistent and can be cross-referenced. Consistently across these sections, we use $x \sim P_X(\cdot)$ to represent data points in the offline dataset and $z \sim P_Z(\cdot)$ to represent their latent representations.

### 3.1  Background on Variational Auto-Encoders

Variational Auto-Encoders (VAEs, Kingma & Welling (2014; 2019)) assume that the given data distribution $P_X(x)$ can be generated with a deep latent-variable model $P_{\theta^*}(x) = \int P_{\theta^*}(z)P_{\theta^*}(x|z)dz$, i.e., sampling a continuous latent variable $z$ from the prior $P_{\theta^*}(z)$ and then generating $x$ from the conditional (generative) model $P_{\theta^*}(x|z)$. It's also assumed that $P_{\theta^*}(z)$ ($P_{\theta^*}(x|z)$) comes from a parametric family of distributions $P_\theta(z)$ ($P_\theta(x|z)$). Intuitively, such $P_\theta(x)$ can be seen as an infinite mixture of some base probability distributions $P_\theta(x|z)$ (e.g., the Gaussian distribution) across a range of possible latent variables $z$, thus it can be quite flexible and powerful for modelling complex data distributions $P_X(x)$.

To learn the mapping from $z$ to $x$, an extra model that can infer the latent variables $z$ from the training data $x$ is required, as $z$'s are not provided with the offline data. However, according to the Bayes rule, the true posterior (inference) model $P_\theta(z|x) = P_\theta(z)P_\theta(x|z)/P_\theta(x)$ is intractable in most cases. For example, if $P_\theta(x|z)$ is implemented as a neural network, we do not have the analytical form for $P_\theta(x)$ and so not for $P_\theta(z|x)$. Therefore, in VAEs, a generative model $P_\theta(x|z)$ and corresponding inference model $P_\phi(z|x)$ (for approximating the true posterior) are jointly learned. The most notable contributions of VAEs are two-fold: firstly, they propose a variational lower bound as a practical learning objective, and secondly, they introduce the reparameterization trick, which facilitates end-to-end training of both the generative and inference models.

The objective function of VAEs is a variational lower bound (a.k.a., evidence lower bound (ELBO)) of the log likelihood $\log P_\theta(x)$, and its derivation process is widely adopted in algorithm designs related to latent-variable models, so we show the process here:

$$\log P_\theta(x) = \mathbb{E}_{z \sim P_\phi(\cdot|x)}[\log P_\theta(x)] = \mathbb{E}_{z \sim P_\phi(\cdot|x)}\left[\log\left[\frac{P_\theta(x,z)}{P_\theta(z|x)}\right]\right] = \mathbb{E}_{z \sim P_\phi(\cdot|x)}\left[\log\left[\frac{P_\theta(x,z)}{P_\phi(z|x)}\frac{P_\phi(z|x)}{P_\theta(z|x)}\right]\right]$$

$$= \mathbb{E}_{z \sim P_\phi(\cdot|x)}\left[\log\left[\frac{P_\theta(x,z)}{P_\phi(z|x)}\right] + \log\left[\frac{P_\phi(z|x)}{P_\theta(z|x)}\right]\right] = \mathcal{L}_{\theta,\phi}(x) + D_{KL}(P_\phi(z|x)||P_\theta(z|x)) \geq \mathcal{L}_{\theta,\phi}(x)$$

(13)

$$\mathcal{L}_{\theta,\phi}(x) = \mathbb{E}_{z \sim P_\phi(\cdot|x)}[\log P_\theta(x,z) - \log P_\phi(z|x)] = \mathbb{E}_{z \sim P_\phi(\cdot|x)}\left[\log P_\theta(x|z) - \log\left[\frac{P_\phi(z|x)}{P_\theta(z)}\right]\right]$$

$$= \mathbb{E}_{z \sim P_\phi(\cdot|x)}[\log P_\theta(x|z)] - D_{KL}(P_\phi(z|x)||P_\theta(z))$$

(14)

The inequality in Eq. (13) uses the fact that KL divergence (i.e., $D_{KL}(\cdot||\cdot)$) is non-negative (Csiszár & Shields (2004)). $P_\phi$ and $P_\theta$ are learned by maximizing $\mathbb{E}_{x \sim P_X(\cdot)}\mathcal{L}_{\theta,\phi}(x)$, which in turns maximizes $\mathbb{E}_{x \sim P_X(\cdot)}\log P_\theta(x)$. When using neural networks for both the inference model $P_\phi(z|x)$ and generative model $P_\theta(x|z)$, $\mathcal{L}_{\theta,\phi}(x)$ trains a specific type of auto-encoder, where $P_\phi(z|x)$ and $P_\theta(x|z)$ function as the encoder and decoder respectively. Intuitively, this auto-encoder is trained by maximizing the data reconstruction accuracy (i.e., the first term in $\mathcal{L}_{\theta,\phi}(x)$), while regularizing the distribution of $z$ from the encoder to be close to the prior distribution of $z$. This regularization is necessary, since $z$ is sampled from the encoder for data generation (i.e., $z \sim P_\phi(\cdot|x), \hat{x} \sim P_\theta(\cdot|z)$) when training but, during evaluation, $z$ is sampled from the prior distribution (i.e., $z \sim P_\theta(\cdot), \hat{x} \sim P_\theta(\cdot|z)$).

Usually, VAEs assume that $P_\theta(z) = \mathcal{N}(z; 0, I)$ and $P_\phi(z|x) = \mathcal{N}(z; \mu(x; \phi), \text{diag}(\sigma^2(x; \phi)))$. Here, assuming $z \in \mathbb{R}^d$, $I$ is a $d \times d$ identity matrix, $\mu(x; \phi) \in \mathbb{R}^d$ is a mean vector, and $\text{diag}(\sigma^2(x; \phi)) \in \mathbb{R}^{d \times d}$ is a diagonal covariance matrix with $\sigma^2_{1:d}(x; \phi)$ as the diagonal elements. In this case, the analytical form of

$D_{KL}(P_\phi(z|x)||P_\theta(z))$ exists and we have:

$$\mathcal{L}_{\theta,\phi}(x) = \mathbb{E}_{z \sim P_\phi(\cdot|x)}\left[\log P_\theta(x|z)\right] + \frac{1}{2}\sum_{i=1}^{d}\left[1 + \log \sigma_i^2(x;\phi) - \mu_i^2(x;\phi) - \sigma_i^2(x;\phi)\right] = \mathcal{L}_{\theta,\phi}^1(x) + \mathcal{L}_\phi^2(x) \quad (15)$$

$\nabla_\phi \mathcal{L}_\phi^2(x)$ and $\nabla_\theta \mathcal{L}_{\theta,\phi}^1(x)$ can be easily estimated with gradient backpropagation. However, when calculating $\nabla_\phi \mathcal{L}_{\theta,\phi}^1(x)$, the gradient $\nabla_z \log P_\theta(x|z)$ cannot be backpropagated to $\phi$, since $z$ are samples from the encoder rather than a function of $\phi$. One solution is the reparameterization trick. Specifically, $z$ can be reparameterized as a deterministic function of $\phi$ by externalizing the randomness from the output to the input of the encoder, i.e., $z = P_\phi(x,\epsilon) = \mu(x;\phi) + \sigma(x;\phi) \odot \epsilon,\ \epsilon \sim \mathcal{N}(0,I)$. ($\epsilon \in \mathbb{R}^d$, and $\odot$ denotes the element-wise product.) In this way, the generative models and corresponding inference models can be jointly trained using straightforward learning techniques (e.g., stochastic gradient descent (Amari (1993))).

Next, we introduce some representative variants of VAEs, which are utilized in Section 3.2 and 3.3.

- $\beta$-**VAE:** Assuming that the data $x \sim P_X(\cdot)$ is generated by a groundtruth simulation process with a number of independent generative factors, $\beta$-VAE (Higgins et al. (2017)) is proposed to learn latent variables $z$ that encode each generative factor in separate dimensions. For example, a simulator might sample independent factors corresponding to the object shape, colour, and size to generate an image of a small green apple; ideally, in the latent variable of the image, there should be separate dimensions for each of these factors. To learn such an encoder $P_\phi(z|x)$, the authors propose to constrain the KL divergence between $P_\phi(z|x)$ and an isotropic unit Gaussian distribution $P_\theta(z) = \mathcal{N}(z;0,I)$ to encourage the dimensions of $z$ to be conditionally independent, resulting in the objective:

$$\max_{\theta,\phi} \mathbb{E}_{z \sim P_\phi(\cdot|x)}\left[\log P_\theta(x|z)\right]\ s.t.\ D_{KL}(P_\phi(z|x)||P_\theta(z)) < \epsilon \quad (16)$$

  Introducing a Lagrangian multiplier $\beta > 0$, the equation above can be reformulated into an unconstrained optimization problem: $\max_{\theta,\phi,\beta} \mathbb{E}_{z \sim P_\phi(\cdot|x)}\left[\log P_\theta(x|z)\right] - \beta\left[D_{KL}(P_\phi(z|x)||P_\theta(z)) - \epsilon\right]$. They opt to fine-tune $\beta$ as a hyperparameter instead of learning it, leading to the final objective:

$$\max_{\theta,\phi} \mathbb{E}_{z \sim P_\phi(\cdot|x)}\left[\log P_\theta(x|z)\right] - \beta D_{KL}(P_\phi(z|x)||P_\theta(z)) \quad (17)$$

  When $\beta = 1$, this recovers the original VAE formulation (i.e., Eq. (14)). $\beta$ controls the tradeoff between limiting the latent channel capacity (via the second term) and preserving the information for reconstructing the data sample (via the first term). Empirically, they find that it's important to set $\beta > 1$ in order to learn the required disentangled latent variables.

- **Gated VAE:** As mentioned in $\beta$-VAE, disentangled latent variables can be viewed as data representations in which each element relates to an independent and semantically meaningful generator factor of the data. $\beta$-VAE provides an unsupervised manner to learn such disentangled representations (through constraining the KL divergence). However, consistent disentanglement has recently been demonstrated to be impossible without inductive bias or subjective validation (Locatello et al. (2019)). Thus, Gated VAE (Vowels et al. (2020)) is proposed to incorporate domain knowledge as weak supervision to encourage disentanglement. Specifically, as a form of weak supervision, data can be clustered such that each cluster comprises data sharing certain generative factors. Then, the latent variable $z$ can be split into several partitions, each of which capture the generative factors of a specific cluster. Suppose that $x_1$ and $x_2$ are from the same cluster which is assigned to the $i$-th partition of the latent variable, i.e., $z_i$, the encoder and decoder can then be trained with a modified ELBO: (which is a lower bound of $\log P_\theta(x_2)$)

$$\mathcal{L}_{\theta,\phi}(x_1,x_2) = \mathbb{E}_{z \sim P_\phi(\cdot|x_1)}\left[\log P_\theta(x_2|z)\right] - D_{KL}(P_\phi(z|x_1)||P_\theta(z)) \quad (18)$$

  Note that the entire $z$ is utilized to generate $x_2$ from $x_1$, but only gradients corresponding to $z_i$ are backpropagated to the inference model $P_\phi$. The rationale is that $x_2$ is intended to reconstruct only the shared generative factors with $x_1$, and these factors should exclusively be captured by $z_i$. Gated VAE requires domain knowledge as supervision, which may limit its general applicability.

- **CVAE:** Conditional VAE (CVAE, Sohn et al. (2015)) is a model for conditional generation. Given a set of data $x$ and their corresponding conditional variables $c$ (e.g., classification labels or desired attributes of the data), CVAE learns a conditional prior $P_\theta(z|c)$ and a conditional generative model $P_\theta(x|z,c)$, so that a data sample $x$ that satisfies a certain condition $c$ can be generated via $z \sim P_\theta(\cdot|c), x \sim P_\theta(\cdot|z,c)$. CVAE is particularly useful in tasks requiring controlled data generation, like scenarios where manipulating specific attributes of the generated data is necessary. As in VAEs, an inference model $P_\phi(z|x,c)$ is introduced for the learning process, which gives the ELBO:

$$\mathcal{L}_{\theta,\phi}(x,c) = \mathbb{E}_{z \sim P_\phi(\cdot|x,c)} \left[ \log P_\theta(x|z,c) \right] - D_{KL}(P_\phi(z|x,c) || P_\theta(z|c)) \tag{19}$$

  Compared with the original VAE formulation, a conditional variable $c$ is introduced to each function, i.e., the prior, inference, and generative model. Following a derivation process similar with Eq. (13), it can be shown that Eq. (19) is a variational lower bound for $\log P_\theta(x|c)$, i.e., the conditional generation likelihood. Further, they note that, during evaluation, latent variables are sampled from $P_\theta(z|c)$ rather than $P_\phi(z|x,c)$, as only the conditions $c$ are available. Thus, to enhance consistency of the data generation process during training and evaluation, they introduce an additional objective term to Eq. (19):

$$\mathcal{L}_{\theta,\phi}^{\text{hybrid}}(x,c) = \alpha \mathcal{L}_{\theta,\phi}(x,c) + (1 - \alpha) \mathbb{E}_{z \sim P_\theta(\cdot|c)} \left[ \log P_\theta(x|z,c) \right] \tag{20}$$

- **VQ-VAE:** Unlike aforementioned VAE variants where $z$ is continuous, VQ-VAE (van den Oord et al. (2017)) is an auto-encoder framework utilizing discrete latent variables. In particular, they assume the latent variable space is a codebook $[e_1^T, \cdots, e_k^T] \in \mathbb{R}^{k \times d}$, where $k$ is number of categories and $d$ is the dimension of the latent vector for each category (i.e., $e_i$). Such discrete representations are usually easier to model than continuous ones and can be particularly advantageous in some situations. For example, in a dataset of images with several categories, categorized latent representations are more appropriate, where each category is assigned a code from the codebook. In particular, the forward process of VQ-VAE is: $z_E = P_\phi(x), z_D = e_i$ ($i = \arg\min_j ||z_E - e_j||_2$), $\hat{x} \sim P_\theta(\cdot|z_D)$. Here, the latent variable from the encoder, i.e., $z_E$, is mapped to its nearest neighbour in the codebook, i.e., $z_D$, which is then input to the decoder for data generation. The learnable parameters of VQ-VAE include $\theta$, $\phi$, and $e_{1:k}$, for which the objective is as below: ($z_D$ and $e_i$ are defined as above; sg denotes the stop-gradient operator.)

$$\mathcal{L}_{\theta,\phi,e_{1:k}}(x) = \log P_\theta(x|z_D) - ||\text{sg}\left[P_\phi(x)\right] - e_i||_2^2 - \beta ||P_\phi(x) - \text{sg}\left[e_i\right]||_2^2 \tag{21}$$

  The generative model is trained to reconstruct the data $x$, as in other VAE variants, via the first term of Eq. (21). The codebook parameters are trained to align with the encoded latent representations $z_E = P_\phi(x)$ through the second term. As for $P_\phi$, it is trained by both the first and third terms. Specifically, they assume $\frac{\partial \log P_\theta(x|z_D)}{\partial z_E} = \frac{\partial \log P_\theta(x|z_D)}{\partial z_D}$, such that the gradient of the first term on $z_D$ can be backpropagated to $\phi$. The third term, on the other hand, is to make sure that the encoder commits to an embedding from the codebook. While empirical evidence demonstrates VQ-VAE's effectiveness, its objective design largely relies on intuition, so additional theoretical underpinning would be beneficial.

- **VRNN:** Chung et al. (2015) introduce a recurrent version of the VAE for the purpose of modeling sequences of data (i.e., $x_{\leq T} \triangleq [x_0, \cdots, x_T]$). At each time step $t$, the historical information $(x_{<t}, z_{<t})$ is involved for sequential decision making, where $z_{<t}$ are latent variables corresponding to $x_{<t}$. Instead of directly concatenating the sequence of variables, i.e., $(x_{<t}, z_{<t})$, as the history, a recurrent neural network (RNN) is adopted to embed the historical information recursively: $h_t = \text{RNN}_\omega(x_t, z_t, h_{t-1})$, $(t = 0, \cdots, T)$ [3]. To involve historical information in decision-making, the prior, generative, and inference model are conditioned on the history embedding as: $P_\theta(z_t|h_{t-1})$, $P_\theta(x_t|z_t, h_{t-1})$, and $P_\phi(z_t|x_t, h_{t-1})$, where $h_{t-1}$ embeds the history $(x_{<t}, z_{<t})$. The overall objective

---

[3] According to (Chung et al. (2015)), additional embedding layers $\text{emb}_X$ and $\text{emb}_Z$ can be used to extract features from the data $x_t$ and latent variables $z_t$, respectively.

(to maximize) for these three models and $\text{RNN}_\omega$ is as below:

$$\mathbb{E}_{P_{\phi,\omega}(z \leq T | x \leq T)} \left[ \sum_{t=1}^{T} \left( \log P_\theta(x_t | z_t, h_{t-1}) - D_{KL}(P_\phi(z_t | x_t, h_{t-1}) || P_\theta(z_t | h_{t-1})) \right) \right] \tag{22}$$

where $P_{\phi,\omega}(z \leq T | x \leq T) = \prod_{t=0}^{T} P_\phi(z_t | x_t, h_{t-1})$ with $h_t$ defined as above. This objective is a variational lower bound of the log likelihood of the data sequence, i.e., $\log P_\theta(x_{\leq T})$, as shown in Appendix A of (Chung et al. (2015)). The introduction of historical information and explicit use of the Hidden Markov Model (Rabiner & Juang (1986)) (e.g., for defining $P_{\phi,\omega}(z \leq T | x \leq T)$) significantly improve the representational capability of VRNN for sequential data.

### 3.2 Variational Auto-Encoders in Offline Reinforcement Learning

Applications of VAEs in offline RL are mainly based on the dynamic programming methods introduced in Section 2.1.1. Based on the given background, we explore specific applications of VAEs for offline RL in Section 3.2.2 -3.2.4, preceded by an overview in Section 3.2.1.

### 3.2.1 An Overview

A major use of VAEs for offline RL is to estimate the behavior policy from the offline data $D_\mu$. Specifically, a CVAE (see Section 3.1) is adopted for this estimation:

$$\max_{\theta,\phi} \mathbb{E}_{(s,a) \sim D_\mu} \left[ \mathbb{E}_{z \sim P_\phi(\cdot | a,s)} \left[ \log P_\theta(a | z, s) \right] - D_{KL}(P_\phi(z | a, s) || P_\theta(z | s)) \right] \tag{23}$$

Corresponding to Eq. (19), $s$ and $a$ here work as the condition variable $c$ and data sample $x$, respectively. As introduced in Section 3.1, the objective above constitutes a lower bound for $\mathbb{E}_{(s,a) \sim D_\mu} [\log P_\theta(a|s)]$, i.e., the typical supervised learning objective. After training, actions at state $s$ can be sampled from the estimated behavior policy $\hat\mu(\cdot|s)$ as: $z \sim P_\theta(\cdot|s), \hat{a} \sim P_\theta(\cdot|s,z)$. In practice, the reconstruction term $\log P_\theta(a|z,s)$ can be replaced with $-||a - \hat{a}||_2^2$ for continuous action spaces, and the prior $P_\theta(z|s)$ can simply be chosen as $\mathcal{N}(z|0, I)$. Following the $\beta$-VAE (i.e., Eq. (17)), a factor $\beta > 0$ is usually introduced as the weight of the KL term in Eq. (23) to balance the two objective terms.

With the estimated behavior policy $\hat\mu$, the policy penalty and support constraint offline RL methods introduced in Section 2.1.1 can be naturally applied. As a representative of support constraint methods, **BCQ** (Fujimoto et al. (2019)) trains such $\hat\mu$ and defines the policy to learn based on $\hat\mu$ as $\pi(\cdot|s) = \hat{a} + \xi(\cdot|s, \hat{a})$, where $\hat{a}$ is a sample from $\hat\mu(\cdot|s)$ and $\xi(\cdot|s, \hat{a})$ is a **bounded** and learnable residual term. $\pi$ is trained to maximize the approximated Q-values as in standard RL [4]. In this way, the action support of $\pi$ is constrained to be close to the behavior policy's. Similarly, inspired by (Shamir (2018)), **AQL** (Wei et al. (2021)) proposes to improve the estimation for the behavior policy $\mu$ using a residual generative model $W_1(W_2 G(s) + \hat{a})$, where $\hat{a} \sim \hat\mu(\cdot|s)$ (as in BCQ) and $W_{1,2}$, $G(\cdot)$ constitute a residual network. They claim that such a residual structure could effectively reduce the estimation error (compared with using $\hat{a}$) for the behavior policy $\mu$.

Regarding the benefits of using a VAE as the policy network, compared to feedforward networks composed solely of fully-connected layers, VAEs excel at capturing the multiple modalities present in $D_\mu$, which could be collected by a diverse set of policies, by utilizing the latent variable $z$. Also, the action generation process, $z \sim \mathcal{N}(0, I)$ and $a \sim P_\theta(\cdot|s, z)$, allows for stochastic action sampling. Compared to other deep generative models, VAEs may be less expressive but are more lightweight than Normalizing Flows and Diffusion Models and can provide more stable training than GANs.

Based on these background knowledge, we provide a review of VAE-based offline RL algorithms in the following subsections, with a particular focus on works whose primary novelty lies in the use of VAEs. One category of such algorithms seeks to enhance dynamic-programming-based offline RL via the use of VAEs. They either modify the learning objective to further mitigate the issue of OOD behaviors, or apply

---

[4] For BCQ, multiple Q functions are learned simultaneously, and the target value for policy evaluation is specifically designed based on these Q functions, which is though not our focus. Please refer to (Fujimoto et al. (2019)) for the details.

augmentation/conversion to the offline data for improved learning. The other category concentrates on extended offline RL setups, such as hierarchical or multi-task offline RL, leveraging the fact that the latent variable $z$ can be learned as embeddings for tasks or subgoals/subtasks within a task. Consequently, $P_\theta(a|s,z)$ can be interpreted as a task-conditioned policy (for multi-task RL) or a subtask-conditioned policy within a hierarchical policy structure (for hierarchical RL).

### 3.2.2 Addressing the Issue of Out-of-Distribution Actions

As introduced in Section 2.1.1, there are four categories of algorithms for mitigating the issue of OOD actions. VAEs have been used to improve three categories among them, which are detailed as follows.

**Applying support constraints: PLAS** (Zhou et al. (2020)) proposes that the support constraint can be simply applied to the latent space of a VAE. In particular, they first estimate the behavior policy as a CVAE $\hat{\mu}$ with Eq. (23). Ideally, after training (through maximizing the reconstruction accuracy of actions induced by the behavior policy), for latent variables $z$ which have high probabilities under $P_\theta(z|s)$, the corresponding decoder $P_\theta(a|s,z)$ should output high-probability actions under the behavior policy distribution (i.e., in-distribution actions), since $\mu(a|s)$ is estimated as $\int P_\theta(z|s)P_\theta(a|s,z)\,dz$. In this case, they learn a latent space policy $\pi(z|s)$ and use it in conjunction with the pretrained (and fixed) decoder $P_\theta(a|s,z)$ as the mapping from $s$ to $a$. The output of $\pi$ is constrained to $[-\sigma,\sigma]$, i.e., the high probability area of the prior distribution $\mathcal{N}(z;0,I)$, to ensure that $(P_\theta \circ \pi)(a|s)$ outputs in-distribution actions. As introduced in Section 5.3, NF-based offline RL algorithms adopt the same algorithm idea. **SPOT** (Wu et al. (2022)) suggests that the constraint in BEAR $(\log\mu(\pi(s)|s) \geq \epsilon, \forall s \in D_\mu)$ can be relaxed to $\mathbb{E}_{s\sim D_\mu}[\log\mu(\pi(s)|s)] \geq \epsilon'$ for practicality. Then, the constrained policy improvement step (i.e., the second equation in Eq. (1)) can be converted to: (We use $\pi$ and $Q$ to represent the policy and Q function from now on, for simplicity.)

$$\max_\pi \mathbb{E}_{s\sim D_\mu}\left[Q(s,\pi(s)) + \lambda(\log\mu(\pi(s)|s) - \epsilon')\right] \tag{24}$$

Here, $\lambda > 0$ is the Lagrangian multiplier. Again, they explicitly model $\mu$ as a CVAE $\hat{\mu} = (P_\phi, P_\theta)$ and propose to approximate $\log\mu(a|s)$ as below: $(z^{(l)} \sim P_\phi(\cdot|s,a),\ l = 1,\cdots,L.)$

$$\log\mu(a|s) \approx \log P_\theta(a|s) = \log\mathbb{E}_{z\sim P_\phi(\cdot|s,a)}\left[\frac{P_\theta(a|s,z)P_\theta(z|s)}{P_\phi(z|s,a)}\right] \approx \log\left[\frac{1}{L}\sum_{l=1}^{L}\frac{P_\theta(a|s,z^{(l)})P_\theta(z^{(l)}|s)}{P_\phi(z^{(l)}|s,a)}\right] \tag{25}$$

As introduced in Section 5.1, Normalizing Flows enable exact density estimation, which can potentially eliminate the need for sample-based approximations as in Eq. (25).

**Applying policy penalty: TD3-CVAE** (Rezaeifar et al. (2022)) proposes to replace the penalty term in Eq. (4) (i.e, $\widehat{D}_{KL}(\pi(\cdot|s)||\mu(\cdot|s))$) with a prediction error $b(s,a) = ||a - P_\theta \circ P_\phi(s,a)||$, where $\hat{\mu} = (P_\phi, P_\theta)$ is the pretrained CVAE for estimating $\mu$. Intuitively, if an action $a$ from the being-learned policy $\pi(\cdot|s)$ corresponds to a high prediction error under $\hat{\mu}(\cdot|s)$, $a$ is probably an OOD action and so $(s,a)$ should be assigned with a high penalty. Further, they theoretically show the equivalence (under certain conditions) of $b(s,a)$ and a KL-divergence penalty term $D_{KL}(\pi(\cdot|s)||\pi_b(\cdot|s))$, where $\pi_b(\cdot|s) = \text{SoftMax}(-b(s,\cdot)/\tau)$ and $\tau > 0$ is a temperature parameter. Compared with $\widehat{D}_{KL}(\pi(\cdot|s)||\mu(\cdot|s))$, $b(s,a)$ is easier to approximate and brings superior empirical performance. **BRAC+** (Zhang et al. (2021a)) points out that estimating the penalty term $D_{KL}(\pi(\cdot|s)||\mu(\cdot|s))$ as $\widehat{D}_{KL}(\pi(\cdot|s)||\mu(\cdot|s))$ in BRAC requires generating a large number of samples to reduce the estimation variance. Therefore, they propose an upper bound for $D_{KL}(\pi(\cdot|s)||\mu(\cdot|s))$, which has an analytical form:

$$D_{KL}(\pi(\cdot|s)||\mu(\cdot|s)) \approx \mathbb{E}_{a\sim\pi(\cdot|s)}\left[\log\pi(a|s) - \log\hat{\mu}(a|s)\right]$$
$$\leq \mathbb{E}_{a\sim\pi(\cdot|s)}\left[\log\pi(a|s)\right] - \mathbb{E}_{a\sim\pi(\cdot|s)}\left[\mathbb{E}_{z\sim P_\phi(\cdot|s,a)}\left[\log P_\theta(a|s,z)\right] - D_{KL}(P_\phi(z|s,a)||P_\theta(z|s))\right] \tag{26}$$
$$= \mathbb{E}_{a\sim\pi(\cdot|s),z\sim P_\phi(\cdot|s,a)}\left[\log\pi(a|s) - \log P_\theta(a|s,z)\right] - \mathbb{E}_{a\sim\pi(\cdot|s)}\left[D_{KL}(P_\phi(z|s,a)||P_\theta(z|s))\right]$$

Here, the inequality is based on the fact that a CVAE $\hat{\mu} = (P_\phi, P_\theta)$ is used to estimate $\mu$ and $\mathbb{E}_{z\in P_\phi(\cdot|s,a)}[\log P_\theta(a|s,z)] - D_{KL}(P_\phi(z|s,a)||P_\theta(z|s))$ constitutes a variational lower bound for $\log\hat{\mu}(a|s)$, as shown in Eq. (19). Given that $P_\theta(a|s,z)$, $P_\theta(z|s)$, and $P_\phi(z|s,a)$ all have Gaussian outputs and suppose

that the policy $\pi(a|s)$ is also Gaussian, then both $\log \pi(a|s) - \log P_\theta(a|s,z)$ and $D_{KL}(P_\phi(z|s,a)||P_\theta(z|s))$ have analytical forms, which can reduce the sample variance. However, sampling for $a$ and $z$ is still required.

**Applying pessimistic values: CPQ** (Xu et al. (2022a)) extends the idea of CQL (introduced in Section 2.1.1) to safe RL, where the policy is trained to maximize its Q-values while minimizing the accumulative cost $Q_c(s,a) = c(s,a) + \mathbb{E}_{s',a'\sim\pi(\cdot|s')}[Q_c(s',a')]$. In particular, they train the cost Q-function $Q_c$ to assign high costs to OOD actions, such that, by constraining $Q_c$ as in standard safe RL algorithms, OOD actions can be avoided at the same time. To realize this, the objective for $Q_c$ is designed as below, which is similar in form with Eq. (6):

$$\min_{Q_c} \mathbb{E}_{(s,a,c,s')\sim D_\mu} \left[ Q_c(s,a) - \left(c + \gamma \mathbb{E}_{a'\sim\pi(\cdot|s')} Q_c(s',a')\right)\right]^2 - \lambda \mathbb{E}_{s\sim D_\mu, a\sim\tilde{\mu}(\cdot|s)} Q_c(s,a) \tag{27}$$

Here, $\tilde{\mu}$ is defined based on $\hat{\mu} = (P_\phi, P_\theta)$. In particular, $\forall a \sim \tilde{\mu}(\cdot|s)$, $D_{KL}(P_\phi(z|s,a)||P_\theta(z|s)) \geq d$, where $P_\theta(z|s) = \mathcal{N}(z; 0, I)$ and $d$ is a predefined threshold. Intuitively, $\tilde{\mu}(\cdot|s)$ produces OOD actions $a$ of which the corresponding posterior distribution $P_\phi(z|s,a)$ deviates significantly with its training target $P_\theta(z|s)$, and the expected cost $Q_c$ of OOD actions is trained to be high. **MCQ** (Lyu et al. (2022)) explores mild but enough conservatism for offline RL to mitigate the underestimation issue of CQL. Their algorithm's design does not hinge on VAEs; instead, the VAE is employed solely for estimating $\mu$ as in previous works, so we do not provide details here. For similar reasons, we skip introductions of **UAC** (Guan et al. (2023)) and **O-RAAC** (Urpí et al. (2021)), both of which can be considered as extensions of BCQ, a support constraint method.

### 3.2.3 Data Augmentation and Transformation

**Data augmentation:** The provided offline data $D_\mu$ may have limited coverage of the state-action space or lack diversity in behavioral patterns. Consequently, constraint- or pessimism-based algorithms may learn sub-optimal policies with limited generalization capabilities in the entire environment. VAEs have been used for data augmentation, aiming at improving the coverage or diversity of the offline data. (1) **ROMI** (Wang et al. (2021a)) is a model-based data augmentation strategy utilizing reverse rollouts. They first learn the backward dynamic model $\widehat{\mathcal{T}}_{\text{rev}}(s|s',a)$, reward model $\hat{r}(s,a)$, and reverse policy $\hat{\mu}_{\text{rev}}(a|s')$ from $D_\mu$ via simple supervised learning, where $s'$ denotes the next state. Specifically, $\hat{\mu}_{\text{rev}}(a|s')$ is modeled with a CVAE $(P_\phi, P_\theta)$, and its training objective is the same as Eq. (23) but to replace $s$ with $s'$. With a random sample $s_{t+1}$ from $D_\mu$, a reverse rollout (of length $h$) can be generated as below:

$$\left[ (s_{t-i}, a_{t-i}, r_{t-i}, s_{t+1-i}) \mid a_{t-i} \sim \hat{\mu}_{\text{rev}}(\cdot|s_{t+1-i}), s_{t-i} \sim \widehat{\mathcal{T}}_{\text{rev}}(\cdot|s_{t+1-i}, a_{t-i}), r_{t-i} \sim \hat{r}(s_{t-i}, a_{t-i}) \right]_{i=0}^{h-1} \tag{28}$$

Unlike the forward generation process, such a reverse manner prevents rollout trajectories that end in OOD states. Also, the CVAE makes it possible for stochastic inference: $z \sim \mathcal{N}(0, I), a \sim P_\theta(\cdot|s', z)$, which improves the diversity. These generated rollouts are then combined with $D_\mu$ for the use of offline RL. This work is closely related to model-based offline RL (Section 2.1.2). (2) To enable the learned policy to generalize to OOD states, **SDC** (Zhang et al. (2022)) suggests training the policy on perturbed states and motivating it to revert to in-distribution states from any state deviations. In particular, a forward dynamic model $\widehat{\mathcal{T}}(s'|s,a)$ and a CVAE-based state transition model $\widehat{U}(s'|s)$ is learned from $D_\mu$ through supervised learning. At a state $\tilde{s}$ perturbed from $s$, the policy $\pi$ is trained to minimize MMD $\left(\widehat{\mathcal{T}}(\cdot|\tilde{s}, \pi(\cdot|\tilde{s}))||\widehat{U}(\cdot|s)\right)$, i.e., to produce actions that can lead it back to the next state $s'$ in the original trajectory from the perturbation $\tilde{s}$. MMD denotes the maximum mean discrepancy. (3) Han & Kim (2022) point out that the latent space of a VAE pretrained on $D_\mu$ can capture the data distribution in $D_\mu$. Based on that, they suggest selectively augmenting the data region that is sparse in the original dataset through data generation with the VAE. However, that paper does not provide details on measuring sparsity through the latent space or data generation using the VAE. (4) **KFC** (Weissenbacher et al. (2022)) suggests inferring symmetries (Hambidge (1967)) of the underlying dynamics of an environment using a VAE forward prediction model, and applying such symmetry transformations to generate new data points as data augmentation. This work requires extensive knowledge of control theory, so it will not be discussed in depth here.

**Data conversion:** VAEs have been used to transform the states or actions in the offline dataset to simplify the learning process. Here, we present two notable works in this direction. When the state space is high-dimensional (e.g., images), directly applying offline RL to the raw data could be challenging. Thus, Rafailov et al. (2021) propose using VAEs to get compact representations $z$ of high-dimensional states $s$ to improve the learning efficiency. $z$ effectively represents $s$, since the VAE is trained to reconstruct $s$ from $z$ while adhering to variational regulations (i.e., the KL term in Eq. (14)). However, this data conversion manner is not specific to VAEs. Other generative models with encoder-decoder structures, such as Normalizing Flows and Transformers, can also be employed for this purpose. On the other hand, **SAQ** (Luo et al. (2023)) proposes to convert continuous actions to discrete ones, to make it significantly simpler to implement constraint-based offline RL methods. To realize this, they train a VQ-VAE (introduced in Section 3.1) on $D_\mu$, which can map a continuous action $a$ at a given state $s$ to a discrete code (through its encoder) for training and map a given discrete code back to the original action space (with its decoder) for evaluation. This state-conditioned action discretization scheme is learned as below:

$$\max_{\theta,\phi,e_{1:k}} \mathbb{E}_{(s,a)\sim D_\mu} \left[ \log P_\theta(a|e_i) - ||\mathrm{sg}\left[P_\phi(s,a)\right] - e_i||_2^2 - \beta||P_\phi(s,a) - \mathrm{sg}\left[e_i\right]||_2^2 \right] \tag{29}$$

Here, $e_{1:k}$ is the codebook and represents the $k$ discretized actions; $i = \arg\min_j ||P_\phi(s,a) - e_j||_2$ is the index of the nearest latent code for the embedding $P_\phi(s,a)$. Applying the pretrained VQ-VAE encoder $P_\phi$ on $D_\mu$ leads to discretized actions, i.e., $(s,a) \rightarrow (s,e_i)$. For discrete action spaces, the estimation of the constraint terms in offline RL, such as the approximated behavior policy $\hat{\mu}(\tilde{a}|s)$ and KL divergence $\hat{D}_{KL}(\pi(\tilde{a}|s)||\mu(\tilde{a}|s))$ in policy penalty methods, could be easier and more accurate.

### 3.2.4 VAE-based Multi-task/Hierarchical Offline RL

Multi-task RL and hierarchical RL extend the basic RL setting. In particular, multi-task RL (Sodhani et al. (2021)) aims at learning a policy that can be directly applied to or quickly adapted to a distribution of tasks. Hierarchical RL, on the other hand, learns a hierarchical (two-level) policy for complex tasks which can usually be decomposed into a sequence of subtasks. In this case, low-level policies can be used to accomplish each subtask, while the high-level policy coordinates the subtasks and the use of low-level policies. For example, in goal-achieving tasks, a goal-conditioned policy can be considered a multi-task policy, as it can be used to reach multiple goals in an environment by changing the goal condition. However, if the goal is distant, the entire path might be partitioned into several subgoals by the high-level policy, and to reach each subgoal, a corresponding low-level (subgoal-conditioned) policy can be employed.

Next, we formally introduce these two setups and explore how VAEs can be utilized to enhance them. By introducing the most notable research in each category, i.e., BOReL for multi-task RL and OPAL & HiGoC for hierarchical RL, our aim is to present the fundamental paradigms of these research directions as a tutorial.

**Multi-task RL:** Given a multi-task offline dataset $D_\mu^M = [[\tau^{i,j} = (s_0^{i,j}, a_0^{i,j}, r_0^{i,j}, \cdots, s_T^{i,j})]_{i=1}^N]_{j=1}^M$, where $i$ and $j$ are indexes for the trajectory and task respectively, multi-task offline RL aims at learning a multi-task policy that can be adapted to unseen tasks with zero- or few-shot training. These unseen tasks are required to be in the same distribution as the training ones. As a representative, **BOReL** (Dorfman et al. (2021)) is proposed for the case where the reward and dynamic function $(r_j, \mathcal{T}_j)$ vary with the task. To train a multi-task policy, a straightforward manner is to condition that policy on the task information $(r_j, \mathcal{T}_j)$. In particular, they adopt the latent variable $z$ [5] of a CVAE as a representation of the reward and dynamic function, and learns a latent-conditioned policy $\pi(a|s,z)$ as the multi-task policy. To this end, the CVAE is trained as follows:

$$\max_{\theta,\phi} \sum_{t=0}^{T-1} \mathrm{ELBO}_t, \ \ \mathrm{ELBO}_t = \mathbb{E}_{z\sim P_\phi(\cdot|\tau_{0:t})} \left[ \log P_\theta(s_{0:T}, r_{0:T-1}|z, a_{0:T-1}) - D_{KL}(P_\phi(z|\tau_{0:t})||P_\theta(z)) \right],$$
$$\tag{30}$$
$$\log P_\theta(s_{0:T}, r_{0:T-1}|z, a_{0:T-1}) = \log P_\theta(s_0|z) + \sum_{t=0}^{T-1} \left[ \log P_\theta(s_{t+1}|s_t, a_t, z) + \log P_\theta(r_t|s_t, a_t, s_{t+1}, z) \right]$$

---

[5] Actually, they adopt the mean and variance of the latent variable (i.e., the output of the encoder) as the policy conditioner.

| Algorithm | VAE Type | VAE Usage | Evaluation Task |
|---|---|---|---|
| BCQ | CVAE | Estimating $\mu$ (Support Constraint) | MuJoCo |
| AQL | CVAE | Estimating $\mu$ (Improved Estimation for $\mu$) | D4RL (L) |
| PLAS | CVAE | Estimating $\mu$ (Support Constraint) | D4RL (L, A, K), Real Robot |
| SPOT | CVAE | Estimating $\mu$ (Support Constraint) | D4RL (L, M) |
| TD3-CVAE | CVAE | Providing prediction errors (Policy Penalty) | D4RL (L, A) |
| BRAC+ | CVAE | Estimating $\mu$ and $D_{KL}(\pi(a|s)||\mu(a|s))$ (Policy Penalty) | D4RL (L) |
| CPQ | $\beta$-CVAE | Estimating $\mu$ and using the latent space for OOD detection (Pessimistic Value) | MuJoCo |
| MCQ | CVAE | Estimating $\mu$ (Pessimistic Value) | D4RL (L) |
| UAC | CVAE | Estimating $\mu$ (Support Constraint) | D4RL (L, M, A) |
| O-RAAC | $\beta$-CVAE | Estimating $\mu$ (Support Constraint) | D4RL (L) |
| ROMI | CVAE | Estimating the reverse behavior policy $\mu_{\mathrm{rev}}(a|s')$ (Data Augmentation) | D4RL (L, M, M2d) |
| SDC | CVAE | Estimating the state transition model $U(s'|s)$ of $D_\mu$ (Data Augmentation) | GridWorld, D4RL (L) |
| KFC | VAE | Modelling $U(s'|s)$ for inference of dynamic symmetries (Data Transformation) | D4RL (L, M, A, K), MetaWorld, RoboSuite |
| SAQ | VQ-VAE | Discretizing the action space to simplify the learning (Data Conversation) | D4RL (L, M, A, K), Robomimic |
| BOReL | CVAE (T) | Embedding MDPs for Multi-task RL | GridWorld, Meta-MuJoCo |
| OPAL | $\beta$-CVAE (T) | Embedding skills for Hierarchical RL | D4RL (M, K) |
| TACO-RL | CVAE (T) | Embedding skills for Hierarchical RL | CALVIN, Real Robot |
| HiGoC | CVAE | Generating subgoals (Hierarchical RL) | D4RL (M), CARLA |
| FLAP | CVAE | Generating subgoals (Hierarchical RL) | Real Robot |

Table 1: Summary of VAE-based offline RL algorithms. In Column 1, we list representative (but not all) algorithms in this section. These algorithms are grouped by their categories. Regarding the VAE types, $\beta$-CVAE refers to an integration of CVAE and $\beta$-VAE, where a weight $\beta$ is added to the KL term in CVAE. The annotation (T) means that the VAE is implemented on trajectories rather than individual state transitions. The evaluation tasks are listed in Column 4. Most works are evaluated on D4RL (Fu et al. (2020)), which provides offline datasets for various tasks, including Locomotion (L), AntMaze (M), Adroit (A), Kitchen (K), Maze2d (M2d), etc. MuJoCo (Todorov et al. (2012)) and CARLA (Dosovitskiy et al. (2017)) are commonly-used simulators for robotic and self-driving tasks, respectively. Meta-MuJoCo (Dorfman et al. (2021)) is a multi-task version of MuJoCo. By Real Robot, we mean evaluations on real robotic platforms, which vary from one study to another. For other benchmarks, we provide their references here: GridWorld (Zintgraf et al. (2020)), MetaWorld (Yu et al. (2019c)), RoboSuite (Zhu et al. (2020)), Robomimic (Mandlekar et al. (2021a)), CALVIN (Mees et al. (2022b)).

In this context, $\tau_{0:T} = (s_{0:T}, r_{0:T-1}, a_{0:T-1})$; $a_{0:T-1}$ and $(s_{0:T}, r_{0:T-1})$ can be viewed as $c$ and $x$ in the CVAE framework. As shown in (Zintgraf et al. (2020)), $\mathrm{ELBO}_t$ constitutes a variational lower bound for $\log P_\theta(s_{0:T}, r_{0:T-1}|a_{0:T-1})$. The second equation in Eq. (30) is derived based on the Markov assumption (Puterman (2014)), and it can be observed from this equation that $z$ is trained to embed information regarding the initial state distribution, reward and dynamic functions, so as to reconstruct the task-specific trajectory $\tau_{0:T}$. Subsequently, the pretrained $P_\phi(z|\tau_{0:t})$, which is implemented as a recurrent neural network as in VRNN (see Section 3.1), can be applied to $D_\mu$ to infer the task embedding $z_t$ at each time step, leading to a dataset of transitions in the form $((s_t, z_t), a_t, r_t)$. Note that $(s_t, z_t)$ can be viewed as an extended state $\tilde{s}_t$, and thus standard offline RL algorithms can then be directly applied to this dataset to learn a latent-conditioned policy $\pi(a_t|(s_t, z_t))$.

**Hierarchical RL:** This category of algorithms try to learn a hierarchical policy $(\pi_{\mathrm{high}}(z|s), \pi_{\mathrm{low}}(a|s, z))$ from the offline dataset. Intuitively, the agent would segment the whole task into a sequence of subtasks or

subgoals, each of which is denoted by a (continuous or discrete) variable $z$ and accomplished by a corresponding subpolicy/skill $\pi_{\text{low}}(a|s, z)$. This hierarchical scheme is especially beneficial for complex tasks with long horizons. **OPAL** (Ajay et al. (2021)) is a representative algorithm in this direction. They define the horizon of each skill to be $h$ and organize the offline data as a set of trajectory segments $D_\mu = [\tau^i = [s_{0:h-1}^i, a_{0:h-1}^i]]_{i=1}^N$. Then, they learn the low-level policy $\pi_{\text{low}}$ for different subtasks $z$ as the decoder of a CVAE:

$$\max_{\theta,\phi} \mathbb{E}_{\tau \sim D_\mu} \left[ \mathbb{E}_{z \sim P_\phi(\cdot|\tau)} \left[ \sum_{t=0}^{h-1} \log \pi_\theta(a_t|s_t, z) \right] - \beta D_{KL}(P_\phi(z|\tau)||P_\theta(z|s_0)) \right], \ \pi_{\text{low}} \triangleq \pi_\theta \tag{31}$$

This objective is equivalent to a ($\beta$-)CVAE ELBO, where $s_0$ and $\tau$ work as the conditioner $c$ and data $x$ respectively [6]. However, unlike Eq. (23), here $\pi_\theta$ is trained on sets of trajectory segments rather than single-step transitions. This is because, as a skill policy, $\pi_\theta$ is expected to extend temporally. Also, the prior $P_\theta(z|s)$ is not fixed but implemented as a neural network that takes $s_0$ as input, to make sure the skill choice $z$ is predictable (by the high-level policy $\pi_{\text{high}}(z|s)$) given only the initial state $s_0$. Applying the pretrained $P_\phi(z|\tau)$ on $D_\mu$ and introducing the reward signals, a new dataset $D_\mu^{\text{high}} = (s_0^i, z^i \sim P_\phi(\cdot|\tau^i), \sum_{t=0}^{h-2} r_t^i, s_{h-1}^i)_{i=1}^N$ can be obtained for training $\pi_{\text{high}}(z|s)$ with any offline RL methods. With this hierarchical policy ($\pi_{\text{high}}, \pi_{\text{low}}$), the decision horizon of offline RL is effectively shorten (by a factor of $h$) and so OOD actions caused by the accumulated distribution shift can be mitigated. Rosete-Beas et al. (2022) propose **TACO-RL**, which is a very similar algorithm with OPAL but specifically tailored for goal-achieving tasks. **HiGoC** (Li et al. (2022)) is also a hierarchical framework for goal-achieving tasks, where the high-level part is a model-based planner [7] for generating the subgoal list and the low-level part is a goal-conditioned policy trained by offline RL to reach each subgoal sequentially. The subgoals are not labeled in the dataset, so the low-level policy is trained in an unsupervised manner. Specifically, for $\pi_{\text{low}}(a_{t_1}|s_{t_1}, s_{t_2})$, $s_{t_2}$ ($t_2 > t_1$) is the subgoal and randomly sampled from states after $s_{t_1}$ in the same trajectory. To be robust to possible OOD subgoals during evaluation, a CVAE $m(s_t|s_{t-h}) = (P_\phi(z|s_t, s_{t-h}), P_\theta(s_t|z, s_{t-h}))$ is pretrained on $D_\mu$ for generating the subgoal $s_t$ conditioned on the previous subgoal $s_{t-h}$, where $h$ is a predefined time interval for subgoal selections. With this CVAE, a perturbed subgoal can be generated based on the sampled one (i.e., $s_{t_2}$) as $P_\theta(P_\phi(s_{t_2}, s_{t_2-h}) + \epsilon, s_{t_2-h})$ ($\epsilon$ is a noise vector), which can replace $s_{t_2}$ as the subgoal of $\pi_{\text{low}}$ for robustness. In cases of high-dimensional states, such as images, adding noise to a well-defined low-dimensional embedding space (i.e., $z$ in the VAE) is a more effective and reasonable approach. The pretrained decoder $P_\theta(s_t|z, s_{t-h})$ can also be used for high-level planning. Specifically, a subgoal list can be generated by specifying a list of latent variables $z_{1:k}$: $s_{t_i} \sim P_\theta(\cdot|z_i, s_{t_i-h})$, $i \in [1, \cdots, k]$. Such a subgoal list can then be evaluated by task-relevant objectives. In this case, searching for an optimal list $z_{1:k}$ is literally a model predictive planning problem and is more efficient than directly searching on the high-dimensional state space( i.e., $s_{t_{1:k}}$). **FLAP** (Fang et al. (2022)) adopts a quite similar protocol with HiGoC, where the CAVE $m$ is referred to as the affordance model.

### 3.3 Variational Auto-Encoders in Imitation Learning

In this section, we offer an overview of VAE-based IL algorithms. First, we provide a tutorial-like overview of the four schemes of VAE-based IL. Based on this, we introduce a categorization of all related works based on how VAEs are utilized. The use of VAEs focuses on enhancing Behavioral Cloning, either from a data or algorithmic perspective.

#### 3.3.1 Core Schemes

Imitation Learning (IL) aims at recovering the expert policy $\pi(a|s)$ from a set of demonstrations $D_E$. Behavioral Cloning (BC) is a straightforward and widely-used IL framework (Pomerleau (1991)), as introduced in Section 2.2.1. With its special encoder-decoder structure, the VAE has been utilized to improve BC from multiple perspectives. Here, we provide a summary of four such schemes.

---

[6] For a standard CVAE, the reconstruction term should be $\log P_\theta(\tau|s_0, z) = \sum_{t=0}^{h-1} \log \pi_\theta(a_t|s_t, z) + \sum_{t=0}^{h-2} \log \mathcal{T}(s_{t+1}|s_t, a_t)$. However, the transition function $\mathcal{T}$ is not trainable and so would not influence the gradient calculation.

[7] Please refer to (Li et al. (2022)) for further details.

**Scheme (1):** Similar with the major use of VAEs for offline RL, a CVAE conditioned on the state can be directly used to model the expert policy:

$$\max_{\theta,\phi} \mathbb{E}_{(s,a)\sim D_E} \left[ \mathbb{E}_{z\sim P_\phi(\cdot|s,a)} \log P_\theta(a|s,z) - D_{KL}(P_\phi(z|s,a)||P_\theta(z|s)) \right] \tag{32}$$

This objective constitutes a variational lower bound for $\mathbb{E}_{(s,a)\sim D_E} \log \pi(a|s)$, i.e., the BC objective. After training, $z\sim P_\theta(\cdot|s), a\sim P_\theta(\cdot|s,z)$, where $P_\theta(z|s)$ is a predefined prior and usually set as $\mathcal{N}(z;0,I)$, can be used as the policy $a\sim \pi(\cdot|s)$. With the latent variable, the VAE-based policy can model stochastic behaviors with diverse modes. Given that $\pi(a|s) = \int P_\theta(a|s,z)P_\theta(z|s)dz$, rather than using a fixed prior distribution $P_\theta(z|s)$, Ren et al. (2020) propose an algorithm to fine-tune the prior online for specific tasks by optimizing a generalization bound from the PAC-Bayes theory.

**Scheme (2):** The second scheme is based on representation learning (RepL), where the latent variable $z$ is adopted as a representation of the state $s$ and usually can be learned via state reconstructions:

$$\max_{\theta,\phi} \mathbb{E}_{s\sim D_E} \left[ \mathbb{E}_{z\sim P_\phi(\cdot|s)} \text{Rec}_\theta(s,z) - D_{KL}(P_\phi(z|s)||P_\theta(z)) \right], \ \max_{\pi} \mathbb{E}_{(s,a)\sim D_E, z\sim P_\phi(\cdot|s)} \log \pi(a|z) \tag{33}$$

where the reconstruction objective $\text{Rec}_\theta(s,z) = \log P_\theta(s|z)$ or $-||s - P_\theta(z)||_2^2$. Note that the first objective in Eq. (33) is usually coupled with extra regularization terms to encourage the disentanglement of the learned representation, or enable the fusion of state inputs $s$ from multiple modalities, etc. Based on the representation learning, a policy $\pi(a|z)$ conditioned on the compact representation $z$ can be learned with any IL algorithm. Compared with $s$, $z$ is in a lower dimension and is less noisy for decision making. As a result, RepL can effectively reduce the need of IL for large amounts of training data. In Eq. (33), there are in total three functions (besides $P_\theta(z)$ which may not be learnable): $P_\phi(z|s)$, $P_\theta(s|z)$, and $\pi(a|z)$. One manner is to pretrain $P_\phi(z|s)$ and $P_\theta(s|z)$ within the VAE framework and then train $\pi(a|z)$ with IL as shown in the second term of Eq. (33). When training $\pi$, the parameters of $P_\phi$ can either be frozen or not. The other manner is to jointly train the three functions through an integrated objective, i.e., adding the two objectives in Eq. (33) together with an adjustable weight $\lambda$. In this case, the state reconstruction with a VAE can be viewed as an auxiliary task of IL. **EIRLI** (Chen et al. (2021a)) provides a systematic empirical investigation of representation learning for imitation. They find that RepL using VAEs can effectively improve the performance of vision-based IL. However, they also mention that the relative impact of using representation learning tends to be lower than the impact of adding data augmentations. This may be because usual RepL techniques (for images) tend to capture the most visually salient axes of variation (e.g., the color, background, or objects) but the action choices are often determined by more fine-grained, local cues in the environment, which calls for RepL that is more specific to decision making. Additional examples in this scheme include (Lee et al. (2019); Rahmatizadeh et al. (2018)).

**Scheme (3):** VAEs can be applied to directly model expert trajectories $\tau = (s_0, a_0, \cdots, s_T)$:

$$\max_{\theta,\phi} \mathbb{E}_{\tau\sim D_E} \left[ \mathbb{E}_{z\sim P_\phi(\cdot|\tau)} \left[ \log P_\theta(\tau|z) \right] - D_{KL}(P_\phi(z|\tau)||P_\theta(z)) \right],$$

$$\log P_\theta(\tau|z) = \log P_\theta(s_0|z) + \sum_{t=0}^{T-1} \left[ \log P_\theta(a_t|s_t,z) + \log P_\theta(s_{t+1}|s_t,a_t,z) \right] \tag{34}$$

The decomposition of $P_\theta(\tau|z)$ makes use of the MDP model, where the three terms correspond to the initial state distribution, policy distribution, and transition dynamic, respectively. $P_\theta(s_0|z)$ and $P_\theta(s_{t+1}|s_t,a_t,z)$, which may be independent of $z$, are usually defined within the simulator and not required to model, so $\log P_\theta(\tau|z)$ can be replaced with $\sum_{t=0}^{T-1} \log P_\theta(a_t|s_t,z)$ in the original objective. Note that $P_\theta(\tau|z)$ has multiple alternative decomposition forms. For example, **T-VAE** (Lu et al. (2019)) proposes to model it as $\log P_\theta(s_{0:T}|z) + \log P_\theta(a_{0:T-1}|z,s_{0:T})$. Specifically, two RNNs are adopted to generate the state and action sequences respectively, leading to a new definition of $\log P_\theta(\tau|z)$: (The state sequence $\hat{s}_{0:T}$ is generated before the action sequence $\hat{a}_{0:T-1}$; the generation of $\hat{s}_t/\hat{a}_t$ is conditioned on $h_{t-1}^{\hat{s}}/h_{t-1}^{\hat{s},\hat{a}}$ which embeds the history $\hat{s}_{1:t-1}/(\hat{s}_{1:t-1}, \hat{a}_{1:t-1})$.)

$$\log P_\theta(\tau|z) = \sum_{t=0}^{T} \log P_\theta(s_t|z, h_{t-1}^{\hat{s}}) + \sum_{t=0}^{T-1} \log P_\theta(a_t|z, h_{t-1}^{\hat{s},\hat{a}}) \tag{35}$$

In this way, the entire trajectory can be predicted without interacting with the environment/simulator, which is necessary in certain scenarios. In this scheme, the embedding $z$ is inferred from the entire trajectory $\tau$, necessitating that the encoder $P_\phi$ be implemented with specialized architectures, such as encoder-only transformers (see Section 6.1) or RNNs (like VRNN in Section 3.1). Different from aforementioned two schemes, such a trajectory-conditioned encoder can output embeddings $z$ that encapsulate information at the task or subtask level. Thus, the learned $P_\theta(a_t|s_t, z)$ can be viewed as a multi-task policy which varies with the task embedding $z$. As a side note, if replacing the decoder $P_\theta(\tau|z)$ with $P_\theta(\tau'|z)$, where $(\tau, \tau') \sim D_E$ are (state-only) expert trajectories and $\tau'$ is a future trajectory segment of $\tau$, Eq. (34) can be used for training a trajectory forecasting model, as detailed in **RC-VAE** (Qi et al. (2020)). By using such a temporal target, i.e, the future segment, the representation $z$ is forced to contain predictive information.

**Scheme (4):** The last scheme is based on the Variational Information Bottleneck (VIB) framework (Alemi et al. (2017)), of which the original objective function is as below:

$$\max_{f:X \to Y} I(Z, Y; f) \; s.t. \; I(X, Z; f) \leq I_c \Rightarrow \max_{f:X \to Y} I(Z, Y; f) - \beta I(X, Z; f) \tag{36}$$

where $I(\cdot)$ denotes the mutual information, $I_c$ is the information constraint, and $\beta > 0$ is the Lagrangian multiplier. The goal of this framework is to learn an encoding $Z$ that is maximally expressive about the target $Y$ while being maximally compressive about the input $X$. Substituting $X, Y$ with $s, a$, the function to learn, i.e., $f$, is then the expert policy $\pi(a|s)$. VIB-based IL can be practically solved by the following objective, which is a lower bound of Eq. (36) (see (Alemi et al. (2017))):

$$\max_{\phi, \theta, \omega} \mathbb{E}_{(s,a) \sim D_E} \left[ \mathbb{E}_{z \sim P_\phi(\cdot|s)} \log P_\theta(a|z) - \beta D_{KL}(P_\phi(z|s) || P_\omega(z)) \right] \tag{37}$$

This objective is similar with the VAE ELBO (i.e., Eq. (14)) in form but has two differences. First, the decoder $P_\theta$ here can predict variables different from the input of the encoder, whereas in VAEs, it is typically trained to reconstruct the input variable. Second, $P_\omega(z)$ is not an assumed prior distribution of $z$ as in VAEs but a variational approximation of the marginal distribution of $z$, i.e., $P_Z(z) = \int P_X(x) P_\phi(z|x) dx$, and it is usually learned as a neural network. Both Scheme (2) and Scheme (4) can be regarded as forms of imitation learning that incorporate effective representation learning. After training, to sample an action $a \sim \pi(\cdot|s)$, the policy encoder first compresses the state to a representation, i.e., $z \sim P_\phi(\cdot|s)$, and then its decoder predicts the action based on $z$, i.e., $a \sim P_\theta(\cdot|z)$. When dealing with high-dimensional and noisy observations $s$, using an information bottleneck can filter out redundant information while retaining essential knowledge for action predictions in the representation $z$. A higher compression rate on the input often results in better generalization of the learned policy across tasks.

Next, we provide an overview of the applications of VAEs in IL. Most works in this direction focus on extending BC. From a data perspective, VAEs can increase BC's data efficiency in scenarios where original, task-specific data is limited, or enable the use of multi-modal input data (e.g., from various sensor types). These applications can mitigate the issue of insufficient demonstrations for IL, as noted in Section 2.3. Regarding algorithmic advancements, VAEs have been utilized for skill discovery to enable hierarchical IL, and to address the causal misidentification issue of BC.

### 3.3.2 Improving Data Efficiency

IL, especially BC, requires a large amount of training data to cover possible task scenarios for robust performance. However, expert-level demonstrations for a specific task, i.e., $D_E$, are usually costly to acquire. Therefore, efficient data usage in IL is crucial. This can be achieved by leveraging the inherent structures within the data, or by gathering and processing task-related data from alternative sources for training. The capability of VAEs to extract latent representations plays a key role in facilitating these processes. Here, we present several notable works in this direction.

**Behavior Retrieval** (Du et al. (2023)) is proposed for the case where vast task-unlabelled demonstrations are available, i.e., $D_{\text{prior}}$. Specifically, $D_{\text{prior}}$ may contain suboptimal behaviors for the current task or demonstrations for a range of different but related tasks (following the same task distribution). To this end, they propose to adopt a $\beta$-VAE (i.e., $(P_\phi, P_\theta)$ in Eq. (17)) to learn the latent representations of

$(s, a) \sim D_{\text{prior}}$. Then, new training data that is similar with the ones in $D_E$ can be retrieved from $D_{\text{prior}}$ as the augmentation $D_{\text{ret}}$. The similarity between data points can be simply measured based on their latents from the pretrained encoder $P_\phi$:

$$f((s_1, a_1), (s_2, a_2)) = -\|z_1 - z_2\|_2, \ (s_1, a_1) \in D_{\text{prior}}, \ (s_2, a_2) \in D_E, \ z_i \sim P_\phi(\cdot|(s_i, a_i)). \tag{38}$$

They claim that this method would effectively filter out sub-optimal or task-irrelevant data. As a final step, a policy $\pi(a|s)$ can be learned on $D_E \cup D_{\text{ret}}$ through BC. This representation learning protocol resembles aforementioned Scheme (2).

For robotic control tasks, state-only demonstrations from a third-person view (e.g., videos from a demonstrator) are usually more available than the ones in the first-person view. Observations from the first- and third-person views, i.e., $s^F$ and $s^T$, correspond to the same true environment state $s$. Given a set of demonstrations synchronized across different viewpoints $[(s_i^F, s_i^{T_1}, \cdots, s_i^{T_k})]_{i=1}^N$, where $T_{1:k}$ denotes $k$ different third-person viewpoints, a common viewpoint-agnostic representation is required to make full use of these data. To realize this, **VAE-TPIL** (Shang & Ryoo (2021)) proposes to learn two VAEs for the first- and third-person view data respectively, i.e., $(P_{\phi^F}, P_{\theta^F})$ and $(P_{\phi^T}, P_{\theta^T})$, and to disentangle the latent variable into a viewpoint embedding $v$ and a viewpoint-agnostic state embedding $h$. To realize this, besides usual reconstruction objective terms, auxiliary objectives are involved: $(i \neq j, \ (h, v) \sim P_\phi(s))$

$$\mathcal{L}_v = \mathbb{E}_{s_i^T} \left[ \|P_\theta(h_i^T, v_j^T) - s_i^T\|_2 \right], \ \mathcal{L}_h = \mathbb{E}_{s^{T_i}} \left[ \|P_\theta(h^{T_j}, v^{T_i}) - s^{T_i}\|_2 \right], \ \mathcal{L}_F = \mathbb{E}_{s^T, s^F} \left[ \|h^T - \text{sg}(h^F)\|_2 \right] \tag{39}$$

The intuition behind these terms are: (a) The view embeddings $v_i^T$ and $v_j^T$ of states in the same view, i.e., $s_i^T$ and $s_j^T$, should be semantically equivalent and so $P_\theta(h_i^T, v_j^T)$ should be able to recover $s_i^T$. (b) The embeddings $h^{T_i}$ and $h^{T_j}$ corresponding to the same state from different views, i.e., $T_i$ and $T_j$, should be interchangeable for reconstructing $s^{T_i}$, since $h$ is expected to be view-agnostic. (c) Similarly, $h^T$ and $h^F$ should be close as they are viewpoint-agnostic and correspond to the same true environment state. With the pretrained $P_{\phi^F}$ and $P_{\phi^T}$, demonstrations across various viewpoints can be converted to viewpoint-agnostic latent representations (i.e., $h$). Then, a representation-based policy $\pi(a|h)$ can be learned with any IL method. This algorithm can be viewed as a method for the challenging setting: Learning form Observations (LfO) (introduced in Section 2.3).

**GIRIL** (Yu et al. (2020b)) suggests modelling the underlying transition from $(s_t, a_t)$ to $s_{t+1}$ in $D_E$ (i.e., its inherent structure) using a CVAE, where $(s_t, a_t)$ and $s_{t+1}$ work as the condition $c$ and data sample $x$ respectively, besides imitating the policy $\pi(a_t|s_t)$. Then, an intrinsic reward can be defined as $r_t = \|\hat{s}_{t+1} - s_{t+1}\|_2^2$, where $\hat{s}_{t+1} \sim P_\theta(\cdot|s_t, a_t)$. (Offline) RL can be applied to the extended demonstrations $[(s_t, a_t, r_t, s_{t+1})]$ to further improve the imitator learned via BC, i.e., $\pi(a_t|s_t)$. Intuitively, this process encourages the agent to explore states with high prediction errors to reduce the state uncertainty, making it possible to achieve better-than-expert performance.

### 3.3.3  Managing Multi-Modal Inputs

In real-world scenarios, decision-making inputs often come from multiple modalities, including RGB images, depth images, 3D point clouds, language instructions, and more. As a motivating example, consider the task of navigating a drone in an outdoor environment, where multiple modalities emerge due to visual information captured by various sensors and factors like the drone's current pose and location. Coherently managing the multi-modal input is essential for the real-life applications of IL. In this context, VAEs have proven effective for fusing or unifying inputs from multiple modalities based on the use of latent embeddings.

**RGBD-VIB** (Du et al. (2022)) proposes to fuse multi-modal inputs based on their uncertainty for evaluation, ensuring that inputs with higher certainty are given more weight in decision-making. To this end, a VAE $(P_{\phi^i}(z|s), P_{\theta^i}(a|z), P_{\omega^i}(z))$ is trained for each modality $i \in [1, \cdots, N]$ using the VIB framework (i.e., Eq. (36)). Note that these VAEs share the same action space but vary in their state spaces. The uncertainty measure can then be acquired based on the pretrained VAEs as $u^i(s) = D_{KL}(P_{\phi^i}(z|s)\|P_{\omega^i}(z))$. Intuitively, a large uncertainty $u^i(s)$ indicates that the current state in modality $i$ may be out of the distribution of the training dataset. Finally, the action is a weighted sum of the output from each VAE decoder, where the weight assigned to each modality is proportional to $\sum_{j=1}^N u^j(s) - u^i(s)$, inversely related to the uncertainty.

Demonstrations may come with contexts $c$, such as the task id, language instructions, goals, etc. In this case, a conditional policy $\pi(a|s,c)$ can be learned for specific task goals or instructions. **MCIL** (Lynch & Sermanet (2021)) is proposed for the scenario where multiple related contexts $c_i, i \in [1, \cdots, N]$ are provided for a task. For a coherent use of these contexts, MCIL learns an encoder for each context, i.e., $(P_{\phi^1}(z|c_1), \cdots, P_{\phi^N}(z|c_N))$, to embed different contexts into a unified latent space, and a decoder $P_\theta(a|s,z)$ as the latent-conditioned policy. The decoder and encoders can be trained end-to-end by $\max_{\theta,\phi^{1:N}} \mathbb{E}_{(s,a,c_i)\sim D_E, z\sim P_{\phi^i}(\cdot|c_i)} [\log P_\theta(a|s,z)]$ [8]. This leads to a policy that generalizes over multi-modal contexts: $z \sim P_{\phi^i}(\cdot|c_i), a \sim P_\theta(\cdot|s,z), i \in [1, \cdots, N]$. As an example, language instructions are usually costly to annotate, but other contexts like the goal images are relatively easy to obtain (from sensors). Through such a unified latent space, MCIL enables the use of (large-amount) demonstrations from easier sources to aid the learning of a language-conditioned policy.

Similarly with MCIL, **CM-VAE-BC** (Bonatti et al. (2020)) aims at learning a joint low-dimensional embedding for multiple data modalities. It utilizes CM-VAE (Spurr et al. (2018)), where a pair of encoder-decoder $(P_{\phi^i}(z|s), P_{\theta^i}(s|z))$ is trained for each modality $i \in [1, \cdots, N]$ via cross-modality reconstructions of which the objective is:

$$\max_{\phi^i, \theta^j} \mathbb{E}_{z\sim P_{\phi^i}(\cdot|s^i)} \left[\log P_{\theta^j}(s^j|z)\right] - D_{KL}(P_{\phi^i}(z|s^i)||P_{\theta^i}(z)) \tag{40}$$

Suppose $N = 2$, then $(i,j)$ in the equation above is an enumeration over the set $\{(1,1),(1,2),(2,1),(2,2)\}$. Note that the equation above is a variational lower bound of $\log P_{\theta^j}(s^j)$ and $s^i$ & $s^j$ corresponds to the same true environment state. This cross-modal training regime results in a single latent space that allows embedding and reconstructing multiple data modalities. After the representation learning, a policy conditioned on the latent variable $\pi(a|z)$ can be learned with any IL method (e.g., BC), following aforementioned Scheme (2), i.e., Eq. (33).

### 3.3.4 VAE-based Skill Acquisition and Hierarchical Imitation Learning

In the context of IL, skills are meaningful subsequences/patterns within the expert trajectories, which can be potentially utilized in multiple related tasks. As an example, consider IL for cooking tasks, where the tasks can be diverse but often have overlapping patterns, such as, slicing, chopping, etc. As introduced in OPAL (see Section 3.2.4), VAEs can be used to extract such patterns via learnt latent embeddings (i.e., using Eq. (31) which is a special case of aforementioned Scheme (3)). As in OPAL, after training, the decoder $P_\theta(a|s,z)$ can be viewed as the policy for skill $z$ (i.e., $\pi_{\text{low}}(a|s,z)$), and the encoder $P_\phi(z|\tau)$ can be used to parse the expert data into trajectories of $(s,z)$ pairs for training the high-level policy $\pi_{\text{high}}(z|s)$ with any IL method. Then, the hierarchical policy can be executed in a call-and-return manner: sampling a skill $z \sim \pi_{\text{high}}(\cdot|s)$, executing the corresponding policy $a \sim \pi_{\text{low}}(\cdot|s,z)$ for $h$ time steps, sampling the next skill, and so on. This whole process can be considered as a hierarchical extension of BC. Beyond this straightforward use of VAEs, some works have proposed to impose additional structures or properties, such as disentanglement, to the latent space for improved skill acquisition (e.g., in interpretability). As mentioned in Section 3.1, being disentangled means that each dimension/partition of the latent variable is independent and semantically meaningful. One way to realize this is to directly incorporate an auxiliary loss term to the standard VAE ELBO as regularization:

$$\max_{\phi,\theta,\omega} \mathcal{L}_{\theta,\phi}^{\text{ELBO}}(D_E) - \lambda \mathcal{L}_{\omega,\theta,\phi}^{\text{reg}}(D_E), \tag{41}$$

where $\mathcal{L}_{\theta,\phi}^{\text{ELBO}}(D_E)$ usually takes the form of Eq. (31); $\omega$ is the parameter of an auxiliary network; $\lambda > 0$ controls the tradeoff between the two loss terms.

**SAILOR** (Nasiriany et al. (2022)) defines $\mathcal{L}_{\omega,\theta,\phi}^{\text{reg}}(D_E) = \mathbb{E}_{(\tau_1,\tau_2)\sim D_E} \left[f_\omega\left(\mu(P_\phi(\cdot|\tau_1)), \mu(P_\phi(\cdot|\tau_2)))\right) - t\right]^2$, where $\tau_1$ and $\tau_2$ belong to the same trajectory and are separated by $t$ time steps, $\mu(P_\phi(\cdot|\tau_1))$ denotes the mean of the encoder output. Such a term can encourage the learned embedding to be predictable (for the temporal difference between skills) and consistent, which is important for downstream policy learning

---

[8]This objective ensembles the VIB framework, i.e., Eq. (36), but overlooks the regulation on $z$. Also, training a separate encoder for each context is necessary, since dealing with different modalities requires different foundation models.

with these skills. Note that the gradient from $\mathcal{L}_{\omega,\theta,\phi}^{\text{reg}}(D_E)$ can be backpropagated to the encoder, allowing this term to shape the learned skill embeddings.

Both OPAL and SAILOR would partition trajectories into segments of length $h$ (i.e., the assumed skill horizon) and then train a CVAE on these independent segments for skill acquisition (i.e., with Eq. (31)). **CKA** (Pasula (2020)), on the other hand, proposes a regularization term on the entire trajectory. In particular, if a trajectory $\tau$ can be decomposed into $m$ skills in a sequence, the learned skill embeddings $z_{1:m}$ can be regularized by a mutual information term $\mathcal{L}_{\omega,\theta,\phi}^{\text{reg}}(D_E) = -\mathbb{E}_{\tau \sim D_E, z_i \sim P_\phi(\cdot|\tau_i)} I\left(\sum_{i=1}^m z_i; \tau\right)$, where $\tau_i$ is the $i$-th segment of $\tau$. The intuition is that, as a complex behavior, $\tau$ should be a combination of these skills. By maximizing the mutual information, the interpretability of the learned embeddings for the behavior $\tau$ can be enhanced. The reason to assume a summation structure $\sum_{i=1}^m z_i$ as the combination form is to enable a practical estimation of the mutual information term (see (Pasula (2020))). Aforementioned methods either assume a fixed skill horizon of $h$ or require that the skill segmentation of the trajectory is readily available, thus only learning skill embeddings for each segment. **CompILE** (Kipf et al. (2019)) suggests a method to concurrently infer both the skill boundaries and skill embeddings directly from the trajectory.

A major benefit of skill acquisition lies in the potential for skills to be transferred across multiple related tasks. This transfer can facilitate the learning of a multi-task policy, $\pi(a|s,z,c)$, where $z$ and $c$ represent the skill and task, respectively. In this case, $(z,c)$ constitutes the policy condition and disentanglement between $z$ and $c$ are required for interpretable and controlled behavior generation. To this end, **TC-VAE** (Noseworthy et al. (2019)) is proposed for the scenario where the task variables $c$ are provided and task-irrelevant skill embeddings $z$ need to be learned. A regularization term $\max_\phi \min_\omega \mathcal{L}_{\omega,\theta,\phi}^{\text{reg}}(D_E) = \max_\phi \min_\omega \mathbb{E}_{(\tau,c) \sim D_E} ||f_\omega(\mu(P_\phi(\cdot|\tau))) - c||_1$ is adopted, where $f_\omega$ is trained to predict the task from the mean of the encoder $\mu(P_\phi(\cdot|\tau))$. Intuitively, maximizing this objective over $\phi$ will encourage the learned latent space $z$ to be non-informative about the task $c$. **SKILL-IL** (Bian et al. (2022)), on the other hand, learns $z$ and $c$ in the meantime as the latent variable of a VAE encoded from a set of multi-task demonstrations. To encourage the disentanglement, a Gated VAE (see Section 3.1) is adopted, where the latent variable is partitioned into two subdomains for $z$ and $c$ respectively. Parameters related to subdomain $z$ are updated on data comprising different skills but within the same task. Similarly, parameters related to subdomain $c$ are trained with data corresponding to the same skill but from different tasks.

### 3.3.5 Tackling Causal Confusion

Causal confusion/misidentification happens when learnt imitator policies become strongly correlated on certain nuisance variables present in the task environment, rather than identifying the causal variables that expert demonstrators actually exploited in decision-making. Thus, more information in the environment could yield worse IL performance as there would be more nuisance factors. For instance, in a driving task, the driver must brake when encountering obstacles or pedestrians. If the demonstration images include brake light indicators on the dashboard, imitators might incorrectly associate the braking behavior with these indicators. However, evaluation scenarios may not always feature these indicators, leading to potentially catastrophic outcomes (such as hitting a pedestrian). Therefore, to be maximally robust to distribution shifts (between the training and evaluation scenarios), a policy must rely solely on true causes of expert actions, thereby avoiding causal misidentification.

As a seminal work, **RCM-IL** (De Haan et al. (2019)) resolves causal misidentification in IL by learning a disentangled representation $z \in \mathbb{R}^d$ of the state $s$, each dimension of which represents a disentangled factor of variation, and then differentiating between nuisance and actual causal factors in $z$. In particular, a $\beta$-VAE is adopted to learn such a disentangled representation $z$ following Eq. (17) (with $x$ replaced by $s$). After pretraining the $\beta$-VAE on $D_E$, states $s$ in $D_E$ can be replaced by their corresponding latent embeddings $z$. Each of the $d$ dimensions of $z$ could represent either a causal or nuisance factor, and it is necessary to mask out the nuisance factors. The problem is then to find the correct mask vector $m \in \{0,1\}^d$. With $m$, the policy can be defined on the masked representations as $\pi(a|m \odot z)$, where $\odot$ denotes element-wise multiplication and $m \odot z$ keeps only actual causal factors. De Haan et al. (2019) propose methods for finding the correct $m$ by querying experts or interacting with the environment, which does not rely on the use of VAEs and so is not detailed here. **Masked** (Pfrommer et al. (2023)) adopts the same protocol as RCM-IL but improves the algorithm for finding the mask vector. Specifically, it uses a carefully-designed statistical

| Algorithm | VAE Type | VAE Usage | Evaluation Task |
|---|---|---|---|
| EIRLI | VAE | Representation Learning (2) | dm_control, Procgen, MAGICAL |
| T-VAE | VAE | Trajectory modeling (3) | 2D Navigation, 2D Circle, Minecraft |
| RC-VAE | VRNN | Trajectory forecasting (3) | Basketball Tracking, PEMS-SF Traffic, Billiard Ball Trajectory |
| Behavior Retrieval | $\beta$-VAE | Data augmentation (2) | RoboSuite, PyBullet, Real Robot |
| VAE-TPIL | VAE | Data augmentation (2) | Minecraft, PyBullet |
| GIRIL | CVAE | Data augmentation | OpenAI Atari, Pybullet |
| RGBD-VIB | VIB | Handling multi-modal input (4) | Real Robot |
| MCIL | VAE | Handling multi-modal input (1) | 3D Playroom |
| CM-VAE-BC | CM-VAE | Handling multi-modal input (2) | AirSim, Real Robot |
| SAILOR | $\beta$-VAE | Skill acquisition (3) | Franka Kitchen, CALVIN |
| CKA | VAE | Skill acquisition (3) | PyBullet |
| CompILE | VAE | Skill acquisition (3) | GridWorld, dm_control |
| TC-VAE | $\beta$-VAE | Skill acquisition (3) | Synthetic Arcs, MIME Pouring |
| SKILL-IL | Gated VAE | Skill acquisition (2)(3) | Craftworld, Real-world Navigation |
| RCM-IL | $\beta$-VAE | Addressing causal confusion (2) | OpenAI Gym/MuJoCo/Atari |
| Masked | $\beta$-VAE | Addressing causal confusion (2) | OpenAI Gym/MuJoCo |
| OREO | VQ-VAE | Addressing causal confusion (2) | OpenAI Atari, CARLA |

Table 2: Summary of representative VAE-based IL algorithms. For each algorithm, we detail the type of VAE used, how the VAE is applied, and the tasks on which it was evaluated. In Column 3, (1)-(4) refer to the 4 schemes of VAE-based IL introduced in Section 3.3.1. As for the evaluation tasks, IL algorithms has relatively more diverse choices than offline RL algorithms (as shown in Table 1). By Real Robot or Real-world Navigation, we mean evaluations on real-world platforms, which vary from one study to another. We provide the references of these benchmarks here: dm_control (Tassa et al. (2018)), Procgen (Cobbe et al. (2020)), MAGICAL (Toyer et al. (2020)), 2D Navigation/Circle (Lu et al. (2019)), Minecraft (Guss et al. (2019)), Basketball Tracking (Felsen et al. (2018)), PEMS-SF Traffic (Dua et al. (2019)), Billiard Ball Trajectory (Fragkiadaki et al. (2016)), RoboSuite (Zhu et al. (2020)), PyBullet (Coumans & Bai (2016–2021)), OpenAI Gym/Atari/MuJoCo (Brockman et al. (2016)), 3D Playroom (Lynch et al. (2019)), AirSim (Shah et al. (2017)), Franka Kitchen (Gupta et al. (2019a)), CALVIN (Mees et al. (2022b)), GridWorld (Zintgraf et al. (2020)), Synthetic Arcs (Noseworthy et al. (2019)), MIME Pouring (Sharma et al. (2018)), Craftworld (Devin (2024)), CARLA (Dosovitskiy et al. (2017)).

hypothesis testing algorithm to efficiently mask out nuisance variables among the latent dimensions produced by the $\beta$-VAE, which does not require expensive queries to the expert or environment for intervention. Under certain assumptions, this algorithm is guaranteed not to incorrectly mask factors that causally influence the expert. Please see (Pfrommer et al. (2023)) for details.

Alternatively, Park et al. (2021) present **OREO** to ameliorate causal misidentification in IL, of which the main idea is to encourage the policy to uniformly attend to all semantic factors in the state to prevent it from strongly exploiting certain nuisance factors. In particular, a VQ-VAE is trained on states from demonstrations using Eq. (21), where $x$ is replaced with $s \sim D_E$. Since there could be multiple (say $n$) semantic factors within $s$, the latent representations from the VQ-VAE, i.e., $z_E$ and $z_D$, are set to be 2D matrices in $\mathbb{R}^{n \times d}$, where $d$ is the dimension of the representation for each single factor. After pretraining, given a state $s$, its multi-factor representation $z_D$ can be acquired by querying the closet code for each row of $z_E$: ($e_{1:k}$ is the codebook, $e_j \in \mathbb{R}^d$, and $z_{E,i} \in \mathbb{R}^d$ is the $i$-th row of $z_E$.)

$$z_E \sim P_\phi(\cdot|s), \ z_D = [e_{q(1)}^T, \cdots, e_{q(n)}^T], \ q(i) = \underset{j \in [1, \cdots, k]}{\arg\min} \|z_{E,i} - e_j\|_2. \tag{42}$$

The next step is to learn a policy through BC based on the multi-factor representation $z_D$ while reducing the influence of nuisance factors. Specifically, at each state $s$, a set of $k$ binary random variables $M_i \in \{0, 1\}, i \in [1, \cdots, k]$ are iid sampled from a Bernoulli distribution, and then a mask vector $m$ corresponding to $s$ can

be created as: $m = [M_{q(1)}, \cdots, M_{q(n)}]$, where $q(1), \cdots, q(n)$ are defined as above. Finally, the IL objective is: $\max_{\pi, f} \mathbb{E}_{(s,a) \sim D_E, m} \log \pi(a | m \odot f(s))$, where $f$ is initialized with the pretrained encoder $P_\phi$ and updated during the policy training. Suppose the $q(i)$-th factor is a nuisance factor that is strongly correlated with decision-making, then it would occur frequently in the latent representations, i.e., $e_{q(i)}^T$ in $z_D$. By randomly setting the corresponding mask $M_{q(i)}$ to 0, this nuisance factor can be effectively dropped out, thereby mitigating the issue of causal misidentification. Randomized dropping provides a simple yet effective way to regularize the uniform treatment of each generative factor, i.e., either a causal variable that explains expert actions or a nuisance variable that just shows strong correlation with the demonstrations.

The representative VAE-based offline RL and IL algorithms are summarized in Table 1 and 2, respectively. We notice that most applications of VAEs in offline policy learning are based on the learned embedding $z$. For example, it can be used to identify OOD actions (Section 3.2.2), transform high-dimensional states or continuous actions for improved learning efficiency (Section 3.2.3), represent tasks or skills for multi-task or hierarchical learning (Section 3.2.4), fuse input from multiple modalities (Section 3.3.3), provide disentangled representations of the states for tackling causal confusion (Section 3.3.5), and so on. On the other hand, we note that some directions of VAE-based offline policy learning can be further explored. For instance, as introduced in Scheme (2) of Section 3.3.1, the VAE can be used to extract compact representations of the states, which can then be used for learning a representation-based policy $\pi(a|z)$. However, EIRLI (Chen et al. (2021a)) suggests that reward- and value-prediction would benefit more (than policy learning) from the representation $z$ that captures mostly coarse-grained visual differences. Thus, it's worth investigating whether VAE-based representation learning would improve offline RL/IL that requires reward- or value-predictions. Moreover, RCM-IL and Masked in Section 3.3.5 assume that $\beta$-VAEs can extract well-disentangled representations of the states as the basis of mitigating causal confusion. However, as noted in Gated VAE (Section 3.1), consistent disentanglement with $\beta$-VAEs, such an unsupervised manner, has been demonstrated to be impossible. Thus, further improvements can be made on tackling causal confusion, by using more advanced variants of VAEs for learning disentangled state representations.

# 4 Generative Adversarial Networks in Offline Policy Learning

Similar to the previous section, we divide the content here into three parts: background on GANs (Section 4.1), the applications of GANs in IL (Section 4.2) and offline RL (Section 4.3). Compared to GAN-based offline RL, GAN-based IL algorithms are relatively more. GAN-based IL primarily extends two fundamental IRL algorithms: GAIL and AIRL, while GAN-based offline RL is mainly based on the model-based offline RL.

## 4.1 Background on Generative Adversarial Networks

The Generative Adversarial Network (GAN, Goodfellow et al. (2014a)) is a generative model based on the minimax optimization. It consists of a generator $G$ and discriminator $D$ that are trained simultaneously to compete against each other. The generator tries to capture the distribution of real data $P_X(x)$ and generate new data examples $x$ from initial noise $z \sim P_Z(\cdot)$. The discriminator is usually a binary classifier used to discriminate generated examples from real data as accurately as possible. Formally, the generator can be modeled as a differentiable function that maps noise $z$ following a certain prior distribution $P_Z(\cdot)$ to samples $x$ in the data space $X$: $z \sim P_Z(\cdot), x = G(z) \sim P_G(\cdot)$. While, the discriminator's output $D(x)$ estimates the probability of a data point $x$ being sampled from the true data distribution $P_X(\cdot)$ rather than generated by the generator. The training objective of GAN is formulated as a minimax game between $G$ and $D$:

$$\min_G \max_D \mathbb{E}_{x \sim P_X(\cdot)}[\log D(x)] + \mathbb{E}_{z \sim P_Z(\cdot)}[\log(1 - D(G(z)))] \tag{43}$$

Here, $D$ is trained to maximize the probability of assigning correct labels to both real data and fake ones from the generator: $D(x) \to 1, D(G(z)) \to 0$. For a certain $G$, the optimal discriminator is given by: $D_G^*(x) = \frac{P_X(x)}{P_X(x) + P_G(x)}$. Plugging this back in Eq. (43), the minimax problem can be converted to an optimization problem with regard to the generator $G$: $\min_G JS(P_X(\cdot) || P_G(\cdot))$, where $JS(\cdot)$ denotes the Jensen-Shannon divergence (Menéndez et al. (1997)). Please refer to (Goodfellow et al. (2014a)) for the detailed derivation. Thus, in essence, $G$ is trained to generate samples that match the real data distribution.

Next, we introduce several variants of GANs, as the necessary background for Section 4.2 and 4.3:

- **Conditional GAN (CGAN) & InfoGAN:** CGAN (Mirza & Osindero (2014)) trains its generator and discriminator together with extra information $y$ as conditions:

$$\min_G \max_D \mathbb{E}_{x \sim P_X(\cdot)}[\log D(x|y)] + \mathbb{E}_{z \sim P_Z(\cdot)}[\log(1 - D(G(z|y)))] \tag{44}$$

  Similarly, this objective can be converted to $\min_G JS(P_X(\cdot|y)||P_G(\cdot|y))$, i.e., matching the conditional distribution of the real data. With these conditional models, CGAN has the advantage of handling not only unimodal datasets but also multimodal ones like Flickr (Plummer et al. (2015)), which consists of diverse labeled image data. With the class label or text as the condition $y$, CGAN can be used for conditional generation. However, when the dataset contains multiple modalities but the labels $y$ are not given, InfoGAN (Chen et al. (2016)) proposes to introduce a latent code $c$ (following an assumed distribution $P_C(\cdot)$) to the generator. In this way, the data generation process is modified into $c \sim P_C(\cdot), z \sim P_Z(\cdot), x = G(z, c) \sim P_G(\cdot|c)$, where $c$ is for capturing distinct modes in the dataset and $z$ targets the unstructured noise shared across the modes. The generator $G$ is trained not only with $V(D, G)$ (i.e., Eq. (43)) but also to maximize the mutual information between the mode variable $C$ and the respective samples $X|_C = G(Z, C)$: $\min_G \max_D V(D, G) - \lambda I(C, X|_C)$), such that $G(Z, C)$ would not degenerate to a unimodal model. By replacing the mutual information with its practical lower bound, we have the objective for InfoGAN as below:

$$\min_{G,Q} \max_D V(D, G) - \lambda \left[ \mathbb{E}_{c \sim P_C(\cdot), z \sim P_Z(\cdot), x = G(z,c)} [\log Q(c|x)] + H(C) \right] \tag{45}$$

  Here, $Q(x|c)$ is a variational posterior neural network trained to approximate the real but intractable posterior distribution $P(c|x)$, and $H(C)$ denotes the entropy of the mode variable $C$ [9].

- **$f$-GAN:** A large class of different divergences, including the KL divergence and JS divergence, are the so-called $f$-divergences (Csiszár & Shields (2004)). Specifically, given two continuous distributions $P(\cdot)$ and $G(\cdot)$, the $f$-divergence between them is defined as $D_f(P(\cdot)||Q(\cdot)) = \int Q(x) f\left(\frac{P(x)}{Q(x)}\right) dx$, where $f : \mathbb{R}^+ \to \mathbb{R}$ should be a convex, lower-semicontinuous function satisfying $f(1) = 0$ [10]. Given a data distribution $P_X(\cdot)$, a corresponding generative model $G$ can be learned through minimizing $D_f(P_X(\cdot)||P_G(\cdot))$. As proposed in $f$-GAN (Nowozin et al. (2016)), this can be approximately solved via a minimax optimization problem:

$$\min_G \max_V \mathbb{E}_{x \sim P_X(\cdot)} [g_f(V(x))] + \mathbb{E}_{x \sim P_G(\cdot)} [-f^*(g_f(V(x)))] \tag{46}$$

  Here, $V$ is a differentiable function without any range constraints on the output, $g_f$ is an output activation function specific to the $f$-divergence used, and $f^*$ is the convex conjugate function of $f$ (Hiriart-Urruty & Lemaréchal (2004)). With specific choices of $f$, $f^*$ and $g_f$, various types of $f$-GAN can be achieved (see Table 6 of (Nowozin et al. (2016))). For example, the original GAN can be recovered by setting $f^*(t) = -\log(1 - \exp(t))$ and $g_f(v) = -\log(1 + \exp(-v))$ [11]. In this way, $f$-GAN provides a generalization of multiple variants of GANs, including the original GAN, LSGAN, and WGAN.

- **Least Squares GAN (LSGAN):** During the early training stage, the generated examples would substantially differ from the real data, but the original GAN objective provides only very small penalties for updating $G$, as shown in Figure 16 of (Goodfellow (2017)). To overcome this vanishing gradient problem, LSGAN (Mao et al. (2017)) replaces the cross-entropy loss used in the original GAN objective with least-square losses:

$$\min_D \mathbb{E}_{x \sim P_X(\cdot)} \left[ (D(x) - j)^2 \right] + \mathbb{E}_{z \sim P_Z(\cdot)} \left[ (D(G(z)) - i)^2 \right], \ \min_G \mathbb{E}_{z \sim P_Z(\cdot)} \left[ (D(G(z)) - k)^2 \right] \tag{47}$$

---

[9] $I(C, X|_C) = H(C) - H(C|X|_C) = -\int P_C(c) \log P_C(c) \, dc + \int P_C(c) P_G(x|c) \log P(c|x) \, dc \, dx \geq -\int P_C(c) \log P_C(c) \, dc + \int P_C(c) P_G(x|c) \log Q(c|x) \, dc \, dx$. The same trick as in Eq. (13) is adopted here to get this inequality.

[10] As an instance, when $f(a) = a \log a$, $D_f$ recovers the KL divergence.

[11] The discriminator in the original GAN $D(x)$ would be a function of $V(x)$, i.e., $D(x) = 1/(1 + \exp(-V(x)))$. $V(x) \in \mathbb{R}$ and thus $D(x) \in (0, 1)$.

Here, $i, j$ are the labels for the generated and real examples, respectively; $k$ is the value that $G$ hopes for $D$ to believe for generated examples. As shown in (Mao et al. (2017)), when $j - i = 2$ and $j - k = 1$, Eq. (47) is equivalent to minimizing the Pearson $\chi^2$ divergence, which is a type of $f$-divergence, between the real and generated data distribution, i.e., $P_X(x)$ and $P_G(x)$.

- **Wasserstein GAN (WGAN):** As mentioned earlier, for the original GAN, the optimization with respect to $G$ is equivalent to minimizing the JS divergence $JS(P_X(\cdot)||P_G(\cdot))$. WGAN (Arjovsky et al. (2017)) proposes to minimize the Wasserstein distance instead, which is defined as $\sup_{||f||_L \leq K} \mathbb{E}_{x \sim P_X(\cdot)}[f(x)] - \mathbb{E}_{x \sim P_G(\cdot)}[f(x)]$. $||f||_L \leq K$ denotes the set of $K$-Lipschitz functions. In this case, the objective function of WGAN is as below:

$$\min_G \max_D \mathbb{E}_{x \sim P_X(\cdot)}[D(x)] - \mathbb{E}_{x \sim P_G(\cdot)}[D(x)] \tag{48}$$

  Here, $D$ is used to estimate the Wasserstein distance, hence its output is not limited to $[0, 1]$ as in the original GAN. Also, the neural network $D$ is applied with weight clipping, ensuring that $D$ is $K$-Lipschitz. Compared with the original GAN, WGAN improves the learning stability and provides meaningful learning curves for parameter fine-tuning.

- **Triple-GAN:** To realize data classification and conditional generation in the meantime, Triple-GAN (Li et al. (2017a)) proposes to introduce a classifier $C(y|x)$ to the original GAN framework. Here, $x$ and $y$ denote the data and label, respectively. Given a dataset of $(x, y)$, empirical joint and marginal distributions of the real data and labels can be acquired as $P_{X,Y}(x, y)$, $P_X(x)$, and $P_Y(y)$. $P_{X,Y}(x, y)$ can be approximated as either $P_X(x)P_C(y|x)$ or $P_Y(y)P_G(x|y)$, corresponding to the classification with $C$ and conditional generation with $G$, respectively. Thus, $C$ and $G$ can be trained to match the joint distribution of $(x, y)$. Here is the objective:

$$\min_{C,G} \max_D \mathbb{E}_{(x,y) \sim P_{X,Y}(\cdot)}[\log D(x, y)] + \lambda_G \mathbb{E}_{y \sim P_Y(\cdot), x \sim P_G(\cdot|y)}[\log(1 - D(x, y))] +$$
$$(1 - \lambda_G)\mathbb{E}_{x \sim P_X(\cdot), y \sim P_C(\cdot|x)}[\log(1 - D(x, y))] + \mathbb{E}_{(x,y) \sim P_{X,Y}(\cdot)}[-\log P_C(y|x)] + \tag{49}$$
$$\lambda_C \mathbb{E}_{y \sim P_Y(\cdot), x \sim P_G(\cdot|y)}[-\log P_C(y|x)]$$

  The first three terms resemble the GAN framework, where $C$ and $G$ are trained to fool the discriminator $D$ by generating $(x, y)$ close to the real data. The last two (cross entropy) terms provide extra supervision for $C$ using samples from the real and generated distributions. Note that the last objective term is only utilized for training $C$. It is shown in (Li et al. (2017a)) that $P_{X,Y}(x, y) = P_X(x)P_C(y|x) = P_Y(y)P_G(x|y)$ if and only if the equilibrium among the three players $(C, G, D)$ is achieved in Eq. (49).

### 4.2 Generative Adversarial Networks in Imitation Learning

This section starts with a detailed introduction of the fundamental algorithms in this direction: GAIL and AIRL, as a tutorial, and then present extensions for each of them. Finally, we provide a spotlight on the integrated use of GANs and VAEs for IL, as examples of synthesizing the use of different generative models for offline policy learning.

#### 4.2.1 Fundamental GAN-based Imitation Learning Algorithms: GAIL and AIRL

Nearly all works regarding applying GANs for IL can be viewed as extensions of GAIL (Ho & Ermon (2016)) or AIRL (Fu et al. (2017)). Both works are based on the maximum causal entropy inverse RL (MaxEntIRL) framework (Ziebart et al. (2010)), of which the objective is as below:

$$\max_c \left[ \min_\pi -H(\pi) + \mathbb{E}_{(s,a) \sim \rho_\pi(\cdot)}[c(s, a)] \right] - \mathbb{E}_{(s,a) \sim \rho_{\pi_E}(\cdot)}[c(s, a)] \tag{50}$$

Here, $\rho_\pi(s, a) = \pi(a|s) \sum_{t=0}^\infty \gamma^t P(s_t = s|\pi)$ is the discounted occupancy measure (i.e., visit frequency) of $(s, a)$ when taking $\pi$, and $H(\pi) = \mathbb{E}_{(s,a) \sim \rho_\pi(\cdot)}[-\log \pi(a|s)]$ denotes the policy entropy. Intuitively, this framework looks for a cost function $c$ that assigns low costs to the expert policy $\pi_E$ and high costs to any

other policy, thereby allowing the expert policy to be learned via minimizing the expected cost, i.e., through the inner RL process in Eq. (50) (i.e., $\min_\pi$). $\pi_E$ is not directly accessible and usually represented as a set of expert demonstrations.

**GAIL** proposes that, to ensure the expressiveness of $c$ while avoiding overfit on the set of demonstrations, $c$ can be any real function (i.e., $\mathbb{R}^{\mathcal{S}\times\mathcal{A}}$) but an extra regularizer $\psi : \mathbb{R}^{\mathcal{S}\times\mathcal{A}} \to \mathbb{R}$ on $c$ should be introduced to Eq. (50). This $\psi$-regularized MaxEntIRL problem is equivalent to an occupancy measure matching problem:

$$\min_\pi -H(\pi) + \psi^*(\rho_\pi - \rho_{\pi_E}), \ \ \psi^*(\rho_\pi - \rho_{\pi_E}) = \sup_{c \in \mathbb{R}^{\mathcal{S}\times\mathcal{A}}} (\rho_\pi - \rho_{\pi_E})^T c - \psi(c) \tag{51}$$

That is, $\psi$-regularized MaxEntIRL implicitly seeks a policy, whose occupancy measure is close to the expert's, as measured by the convex conjugate of the regularizer, i.e., $\psi^*$. As introduced in Section 4.1, GANs provide a scalable manner for distribution matching. GAIL proposes that, with a certain design of $\psi$ (i.e., Eq. (13) in (Ho & Ermon (2016))), Eq. (51) can be converted to a GAN objective:

$$\min_\pi \max_{D \in (0,1)^{\mathcal{S}\times\mathcal{A}}} -H(\pi) + \mathbb{E}_{(s,a)\sim\rho_\pi(\cdot)}[\log(1 - D(s,a))] + \mathbb{E}_{(s,a)\sim\rho_{\pi_E}(\cdot)}[\log D(s,a)] \tag{52}$$

Compared with Eq. (50), we can see that $c(s,a)$ is defined with the discriminator as $\log(1-D(s,a))$ [12]. Also, the policy $\pi$ works as the generator $G$, so $D$ and $\pi$ can be alternatively trained as in GANs to optimize Eq. (52). Specifically, in each training episode, $\pi$ is trained for several iterations to maximize the expected return $Q(s_0, a_0) = \sum_{t=0}^\infty \gamma^t(-\log(1 - D(s_t, a_t)) - \lambda \log \pi(a_t|s_t))$, $\forall (s_0, a_0)$, with a widely-adopted RL algorithm – TRPO (Schulman et al. (2015)) [13], and then $D$ is trained for iterations to correctly classify samples from $\pi$ or $\pi_E$ with Eq. (52). Compared with the original MaxEntIRL, which requires solving a complete RL problem as the inner loop for each update of $c$, GAIL ensures both expressiveness and scalability, especially useful for tasks in high-dimensional state/action spaces. Further, GAIL provides solid theoretical results on the connection between IRL and GANs. As for the global optimality and convergence rate of GAIL, Zhang et al. (2020b) provide an analysis under the condition of using two-layer neural networks (with ReLU between the two layers) as function estimators and applying natural policy gradient for policy updates, when the underlying MDP belongs to the class of linear MDPs.

**AIRL** also adopts a GAN-like framework as follows:

$$\max_D \mathbb{E}_{(s,a)\sim\rho_\pi(\cdot)}[\log(1 - D(s,a))] + \mathbb{E}_{(s,a)\sim\rho_{\pi_E}(\cdot)}[\log D(s,a)],$$
$$\min_\pi \mathbb{E}_{(s,a)\sim\rho_\pi(\cdot)}[\log(1 - D(s,a)) - \log D(s,a)] \tag{53}$$

However, instead of applying a sigmoid function (i.e., $D(s,a) = \frac{\exp(f(s,a))}{\exp(f(s,a))+1}$) as the activation function on the last layer output (i.e., $f(s,a)$) of the discriminator as in GAN and GAIL, AIRL adopts $D(s,a) = \frac{\exp(f(s,a))}{\exp(f(s,a))+\pi(a|s)}$. They claim that, with this design, AIRL is equivalent to MaxEntIRL, under the assumption that trajectories $\tau = \{s_0, a_0, \cdots, s_T, a_T, s_{T+1}\}$ follow the Boltzmann distribution (i.e., $P(\tau) \propto \rho_0(s_0) \prod_{t=0}^{T-1} \mathcal{T}(s_{t+1}|s_t, a_t) \exp(\gamma^t r(s_t, a_t)))$. However, the derivation provided in (Fu et al. (2017)) is not rigorous [14]. Further, they propose to replace $D(s,a)$ with $D(s,a,s') = \frac{\exp(f(s,a,s'))}{\exp(f(s,a,s'))+\pi(a|s)}$, where $f(s,a,s') = g(s) + \gamma h(s') - h(s)$. They state that, at optimality, $g(s)$ and $h(s)$ can recover the reward and value function, respectively, if the real reward function is a function of $s$ only and the environment is deterministic [15]. While its theoretical foundation is not robust, AIRL can be considered as a GAN-based IL algorithm drawing inspiration from MaxEntIRL and well substantiated by empirical results.

---

[12] We equivalently substitute all $D(s,a)$ in (Ho & Ermon (2016)) as $1 - D(s,a)$ to be in line with GAN, where the true and fake data should be labeled as 1 and 0, respectively, by the discriminator $D$.

[13] $\max_\pi Q(s_0, a_0) = \sum_{t=0}^\infty \gamma^t(-\log(1-D(s_t,a_t)) - \lambda \log \pi(a_t|s_t))$ is equivalent to $\min_\pi -\lambda H(\pi) + \mathbb{E}_{(s,a)\sim\rho_\pi(\cdot)}[\log(1-D(s,a))]$. $\lambda > 0$ is introduced here to balance the two objective terms.

[14] In Appendix A.1 of (Fu et al. (2017)), the step $\frac{\partial}{\partial\theta} \log Z_\theta = \mathbb{E}_{p_\theta}\left[\sum_{t=0}^T \frac{\partial}{\partial\theta} r_\theta(s_t, a_t)\right]$ is questionable. Also, in its Appendix A.2, they erroneously mix up the use of $\mu_t$ and $\hat{\mu}_t$.

[15] In Appendix A.4 of (Fu et al. (2017)), they claim that the global minimum of the discriminator objective is reached when $\pi = \pi_E$, which is backed up by the original GAN paper (Goodfellow et al. (2014b)). However, the objective design of AIRL is significantly distinct from the original GAN. Moreover, this claim forms the basis of the proof presented in its Appendix C on $g(s)$ and $h(s)$ recovering the reward and value functions respectively.

There is an extensive amount of works developed based on GAIL or AIRL. Instead of providing details on each of them, we put an emphasis on the representative ones by presenting their objective designs and key novelties in tables, and briefly enumerate other works for comprehensiveness.

### 4.2.2 Extensions of GAIL

In Table 3, we provide an overview of representative works that extend GAIL. They either substitute the original GAN framework with more advanced GAN variants (as introduced in Section 4.1) or expand the use of GAIL to other IL settings, including the multi-agent, multi-task, hierarchical, and model-based scenarios. Next, to provide a thorough review, we briefly enumerate other related works, which are further categorized by their base algorithms.

**GAIL:** Extensions of GAIL concentrate on tackling key challenges such as increasing the sample efficiency for policy learning, ensuring balanced training of the discriminator and policy, etc. (1) **DGAIL** (Zuo et al. (2020)) and **SAM** (Blondé & Kalousis (2019)) replace the stochastic policy $a \sim \pi(\cdot|s)$ in GAIL with a deterministic one $a = \pi(s)$ and, correspondingly, proposes a DDPG-based (Lillicrap et al. (2016)) updating rule for $\pi$. As an off-policy algorithm, DDPG is more sample efficient than TRPO (used in GAIL). **BGAIL** (Jeon et al. (2018)) improves the sample efficiency by approximating the posterior distribution of the discriminator's parameters in the ideal case where correct labels are assigned to both expert and generated behaviors. A more accurate estimation of the cost function can then be acquired through sampling from that distribution, consequently improving the policy training. (2) **VAIL** (Peng et al. (2019a)) suggests limiting information flow in the discriminator of GAIL (or AIRL) using the Variational Information Bottleneck framework (Tishby & Zaslavsky (2015)), to prevent the discriminator from converging too quickly and so providing uninformative gradients to the generator. **TRAIL** (Zolna et al. (2020)) proposes to regularize the discriminator through an auxiliary task, with the aim to discourage the agent to exploit spurious patterns in observations which are associated with the expert label but task-irrelevant. (3) For applications demanding reliability and robustness, **RS-GAIL** (Lacotte et al. (2019)) presents a risk-sensitive version of GAIL by introducing a constraint to the original GAIL objective, such that the conditional value-at-risk (Duffie & Pan (1997)) of the learned policy is at least as well as that of the expert. This algorithm is supported by rigorous theoretical validation.

**Multi-agent/Hierarchical GAIL: IGASIL** (Hao et al. (2019)) extends GAIL to fully-collaborative multi-agent scenarios. It adopts a similar objective design with MAGAIL's (listed in Table 3) but suggests two major improvements for learning efficiency: training the policy with DDPG for enhanced sample efficiency and adopting high-return trajectories from the policy as demonstrations for self-imitation learning. Like Option-GAIL introduced in Table 3, **OptionGAN** and **Directed-Info GAIL** extend GAIL to the hierarchical learning setting, taking advantage of the Mixture-of-Experts framework (Masoudnia & Ebrahimpour (2014)) and directed information maximization (Massey et al. (1990)), respectively. However, these two algorithms are not suitable for learning from demonstrations segmented by sub-tasks. Also, Directed-Info GAIL trains $\pi_H$ and $\pi_L$ in two separate stages, leading to suboptimality of the hierarchical policy $\pi = (\pi_H, \pi_L)$.

**InfoGAIL:** Hausman et al. (2017) and Peng et al. (2022) address the same challenge as InfoGAIL, i.e., IL from multi-modal but unstructured demonstrations. Hausman et al. (2017) achieve the same objective function as InfoGAIL but derive it through a distinct perspective; Peng et al. (2022) proposes an alternative manner to update $Q$ (in InfoGAIL) through a VAE framework, where $Q$ and $\pi$ serve as the encoder and decoder, respectively. **Ess-InfoGAIL** (Fu et al. (2023)) extends InfoGAIL to manage demonstrations from a mixture of experts, wherein the quantity of demonstrations from each expert is imbalanced. **Burn-InfoGAIL** (Kuefler & Kochenderfer (2018)) improves InfoGAIL to reproduce expert behaviors over extended time horizons.

### 4.2.3 Extensions of AIRL

Extensions of AIRL are relatively fewer, but we follow the content structure in Section 4.2.2, providing a summarization table as Table 4 and a brief review of other related works as below.

AIRL adopts the function design $f(s, a, s') = g(s) + \gamma h(s') - h(s)$, where $g(s)$ is assumed to recover the reward function and $h(s)$ can be viewed as the potential function for reward shaping (Ng et al. (1999)). **EAIRL**

| Algo (GAN Type) | Objective | Key Novelty |
|---|---|---|
| CGAIL
(CGAN)
(Zhang et al. (2019)) | $\min_\pi \max_D -\lambda H(\pi(\cdot|c)) + \mathbb{E}_{\rho_{\pi_E}(\cdot|c)}[\log D(s,a|c)]+$
$\mathbb{E}_{\rho_\pi(\cdot|c)}[\log(1-D(s,a|c))]+$
$\mathbb{E}_{\rho_{\pi_E}(\cdot|c)}[\log(1-D(s,a|c'))]$ | Recovering a policy $\pi(a|s,c)$
for each condition $c$;
Demonstrations are shareable
among similar task conditions,
so the data efficiency can be high;
$\max_D \mathbb{E}_{\rho_{\pi_E}(\cdot|c)}[\log(1-D(s,a|c'))]$
discourages the mismatch between
$c'$ and $(s,a)$ for consistency. |
| InfoGAIL
(InfoGAN)
(Li et al. (2017b)) | $\min_{\pi,Q} \max_D -\lambda_1 H(\pi) + \mathbb{E}_{\rho_\pi}[\log(1-D(s,a))]+$
$\mathbb{E}_{\rho_{\pi_E}}[\log D(s,a)] - \lambda_2 L_I(\pi,Q),$
$L_I(\pi,Q) = \mathbb{E}_{c\sim P_C(\cdot),a\sim\pi(\cdot|s,c)}[\log Q(c|s,a)] + H(C),$
$P_C(\cdot)$ is an assumed prior distribution. | Demonstrations are multi-modal but
labels $c$ are not provided, unlike CGAIL;
$L_I(\pi,Q)$ is a lower bound of
the mutual information $I(c;(s,a))$, by
maximizing which we can disentangle
trajectories corresponding to different $c$;
Policies for each latent modality
$\pi(a|s,c)$ can then be recovered. |
| $f$-GAIL
($f$-GAN)
(Zhang et al. (2020a)) | $\min_\pi \max_{f^*\in\mathcal{F}^*,T} \mathbb{E}_{\rho_{\pi_E}}[T(s,a)] - \mathbb{E}_{\rho_\pi}[f^*(T(s,a))]$
$-H(\pi),\ f^*$ is the convex conjugate of $f$ (which
defines a certain $f$-divergence) and satisfies:
convexity and $\inf_{u\in\mathrm{dom}_{f^*}}\{f^*(u)-u\} = 0.$
Like $\pi$ and $T$, $f^*$ is modeled as a neural network
but specially designed to fulfill its two constraints. | GAIL is a special case of $f$-GAIL, when
$T(x) = \log D(x),\ f^*(x) = -\log(1-e^x)$;
Maximizing over $f^*\in\mathcal{F}^*$ aims to find
the largest $f$-divergence between $\rho_\pi$ and
$\rho_{\pi_E}$, which can better guide $\pi$ to $\pi_E$. |
| Triple-GAIL
(Triple-GAN)
(Fei et al. (2020)) | $\min_{\pi,C} \max_D -\lambda_1 H(\pi) + \mathbb{E}_{\rho_{\pi_E}}[\log D(s,a,c)]+$
$\lambda_2\mathbb{E}_{\rho_\pi}[\log(1-D(s,a,c))]+$
$(1-\lambda_2)\mathbb{E}_{\rho_C}[\log(1-D(s,a,c))]+$
$\lambda_4\mathbb{E}_{\rho_{\pi_E}}[-\log C(c|s,a)] + \lambda_5\mathbb{E}_{\rho_\pi}[-\log C(c|s,a)];$
$c$ can be viewed as labels for $(s,a)$, provided in
demonstrations; $C(c|s,a)$ is utilized to classify $(s,a)$,
while $\pi(a|s,c)$ is the policy for the class $c$. | The development and theoretical validation
of Triple-GAIL exactly follow Triple-GAN
(see Section 4.1). $C$ and $\pi$ are trained by
matching the approximated joint occupancy
measures of $(s,a,c)$: $\rho_\pi(\cdot)$ and $\rho_C(\cdot)$,
with $\rho_{\pi_E}(\cdot)$. The last two objective
terms serve as extra supervision for $C$.
Viewing $c$ as options/skills, $C$ and $\pi$
then constitute a hierarchical policy. |
| MGAIL
(GAN)
(Baram et al. (2017)) | The objective of $D$ is the same as the one of GAIL.
$\pi(a|s;\theta)$ is trained by maximizing the return
$J(s_0,a_0;\theta)$, where the policy gradient, i.e., $J_\theta^0$,
can be approximated recursively as: $(t=T\to 0)$
$J_\theta^t = R_a\pi_\theta + \gamma(J_\theta^{t+1} + J_{s'}^{t+1}f_a\pi_\theta),$
$J_s^t = R_s + R_a\pi_s + \gamma J_{s'}^{t+1}(f_s + f_a\pi_s);$
$f$ is the learned dynamics: $s' = f(s,a),\ R(s,a)$
is the reward defined with $D$, $R_s \triangleq \frac{\partial R}{\partial s}$ and so on. | This is a model-based version of GAIL.
A dynamic model $f$ is learned, such that
the agent can look one-step ahead to
additionally adopt the gradient from $f$,
i.e., $\gamma J_{s'}^{t+1}f_a\pi_\theta$, as part of $J_\theta^t$.
(Typically, $J_\theta^t = R_a\pi_\theta + \gamma J_\theta^{t+1}$.)
Thus, fewer expert data and policy samples
would be required, compared with GAIL. |
| MAGAIL
(GAN)
(Song et al. (2018)) | $\min_\pi \max_D \mathbb{E}_{\rho_\pi}\left[\sum_{i=1}^N \log(1-D_i(s,a_i))\right]+$
$\mathbb{E}_{\rho_{\pi_E}}\left[\sum_{i=1}^N \log D_i(s,a_i)\right],\ \pi=(\pi_1,\cdots,\pi_N),$
$\pi_E = (\pi_E^1,\cdots,\pi_E^N).$ All agents share the state $s$
but choose their own action $a_i \sim \pi_i(\cdot|s).$
The other agents $-i$ can be viewed as part of
the environment for agent $i$. However, if $\pi_{-i}$
is unknown, the training environment would be
non-stationary for $i$, as $\pi_{-i}$ are also being updated. | A multi-agent extension of GAIL, which
learns a $D_i$ and $\pi_i$ for each agent.
As agents interact with each other,
$D_i$ and $\pi_i$ are not learned independently.
The paper's theoretical results assume
$\pi_E^{-i}$ is known, enabling the learning for
each agent $i$ to be treated separately.
Yet, in practice, better management for the
non-stationarity than MAGAIL is required. |
| Option-GAIL
(GAN)
(Jing et al. (2021)) | $\min_\pi \max_D -\lambda H(\pi) + \mathbb{E}_{\rho_\pi}[\log(1-D(o',s,o,a))]$
$+\mathbb{E}_{\rho_{\pi_E}}[\log D(o',s,o,a)],\ \pi=(\pi_H,\pi_L),$
$o \sim \pi_H(\cdot|s,o'),\ a \sim \pi_L(\cdot|s,o).$
The agent decides on its current option (a.k.a.,
skill) $o$ with $\pi_H$ based on $s$ and $o'$ (i.e., the
previous option), and then samples actions $a$
with $\pi_L$ subject to $o$. | A hierarchical learning version of GAIL.
Long-horizon tasks can be segmented to a
sequence of sub-tasks, each of which can be
done with an option $o$ (i.e., sub-policy).
This algorithm can recover a hierarchical
policy, useful for long-horizon tasks,
from the expert data through matching
the occupancy measure of $(o',s,o,a)$
between $\pi$ and $\pi_E$, which follows GAIL.
An EM version of Option-GAIL is proposed,
in case the expert's options are not labeled. |

Table 3: Summary of representative works following GAIL. In Column 1, we list the algorithms and the type of GANs they utilize. Their key objectives and novelties are listed in Column 2 and 3, respectively.

| Algo (GAN Type) | Objective | Key Novelty |
|---|---|---|
| $f$-IRL (GAN) (Ni et al. (2020)) | $\max_D \mathbb{E}_{s\sim\rho_{\pi_E}(\cdot)}\left[\log D(s)\right] + \mathbb{E}_{s\sim\rho_\theta(\cdot)}\left[\log(1-D(s))\right]$, $\min_\theta L_f(\theta), L_f(\theta) = D_f(\rho_{\pi_E}(s)\|\rho_\theta(s)), \rho_\theta(s) \propto$ $\int \rho_0(s_0)\prod_{t=0}^{T-1}\mathcal{T}(s_{t+1}|s_t,a_t)\exp(\frac{r_\theta(s_{t+1})}{\alpha})\eta_\tau(s)d\tau$, $\eta_\tau(s) = \sum_{t=1}^T \mathbb{1}(s_t = s)$; The training alternates between $\max_D$ and $\min_\theta$; With the learned reward function $r_\theta$, $\pi$ is trained by RL. $\min_\theta L_f(\theta)$ is realized by gradient descents: $\nabla_\theta L_f(\theta) = \frac{1}{\alpha T}\text{cov}_{\tau\sim\rho_\theta(\cdot)}\left(\sum_{t=1}^T f(u_t) - f'(u_t)u_t, \sum_{t=1}^T \nabla_\theta r_\theta(s_t)\right)$, $u_t = \frac{\rho_{\pi_E}(s_t)}{\rho_\theta(s_t)} \approx \frac{D(s_t)}{1-D(s_t)}$, $\text{cov}(\cdot)$ denotes covariance. | This work provides a generalization of AIRL by applying the the general $f$-divergence. The reward $r_\theta(s)$ is trained by state marginal matching, i.e., $\min_\theta D_f(\rho_{\pi_E}(s)\|\rho_\theta(s))$. $D$ is used to estimate $u_t$ in $\nabla_\theta L_f(\theta)$, based on the fact: $D^*(s) = \rho_{\pi_E}(s)/(\rho_{\pi_E}(s) + \rho_\theta(s))$. So, each update of $\theta$ requires a near-optimal $D^*$, which is time-consuming. |
| MA-AIRL (GAN) (Yu et al. (2019a)) | $\max_D \sum_{i=1}^N \mathbb{E}_{\rho_{\pi_E}}\left[\log D_i(s,a)\right] + \mathbb{E}_{\rho_\pi}\left[\log(1-D_i(s,a))\right]$, $\min_{\pi_i} \mathbb{E}_{\rho_\pi}\left[\log(1-D_i(s,a)) - \log D_i(s,a)\right], i = 1,\cdots,N$; $D_i(s,a) = \exp(f(s,a))/(\exp(f(s,a)) + \pi_i(a_i|s))$, $a = (a_1,\cdots,a_N), \pi = (\pi_1,\cdots,\pi_N)$. All agents share a state $s$ and have access to action decisions from the other agents, i.e., $a_{-i}$, during the training process. | This is a multi-agent version of AIRL, which trains $D_i$, $\pi_i$ and applies AIRL to each agent. States and joint actions are shared among agents. Theoretical results are based on Markov games and logistic stochastic best response equilibrium. However, viewing $a_{-i}$ as part of the environment, the problem and derivations degenerate to the single-agent AIRL case. |
| MH-AIRL (GAN) (Chen et al. (2023e)) | $\max_D \mathbb{E}_{\pi_E}\sum_{t=0}^{T-1}\log D(\tilde{s}_t,\tilde{a}_t|c) + \mathbb{E}_\pi\sum_{t=0}^{T-1}\log(1-D(\tilde{s}_t,\tilde{a}_t|c)), (\tilde{s}_t,\tilde{a}_t) = ((o_{t-1},s_t),(o_t,a_t))$; $\min_\pi \mathbb{E}_\pi\left[\sum_{t=0}^{T-1}\log(1-D_t^c) - \log D_t^c\right] - \lambda_1 I(\tau_\pi;C) - \lambda_2 I(\tau_\pi \to O_{0:T}|C), \tau_\pi = \{(s_t,a_t)\}_{t=0:T-1}$, $D_t^c = \frac{\exp(f(\tilde{s}_t,\tilde{a}_t|c))}{\exp(f(\tilde{s}_t,\tilde{a}_t|c))+\pi(\tilde{a}_t|\tilde{s}_t,c)}, \pi = (\pi_H,\pi_L)$ and $\pi(\tilde{a}_t|\tilde{s}_t,c) = \pi_H(o_t|s_t,o_{t-1})\pi_L(a_t|s_t,o_t)$. $c$ and $o$ denote the task and option variable, respectively. MH-AIRL can also be adopted, when $c$ and $o$ are not labeled in demonstartions, via an Expectation–Maximization (EM) design. | This algorithm integrates multi-task learning, hierarchical learning, and AIRL. $\pi$ is a hierarchical policy that can be applied to multiple tasks by conditioning on corresponding $c$. The objective design can be viewed as AIRL on the extended state-action space $(\tilde{s}_t,\tilde{a}_t)$. As regularization, $\pi$ is trained to maximize the mutual /directed information $I(\tau_\pi;C)$ /$I(\tau_\pi \to O_{0:T}|C)$ to build the causal relationship between $\pi$ and the task /option variables. |

Table 4: Summary of representative works following AIRL.

(Qureshi et al. (2019)) proposes to implement $h(s)$ as the empowerment function (Mohamed & Rezende (2015)), maximizing which would induce an intrinsic motivation for the agent to seek the states that have the highest number of future reachable states. However, as claimed in AIRL, $h(s)$ is designed to recover the value function at optimality, which is different from the empowerment function in definition. AIRL has been extended to various IL setups, including model-based, off-policy, multi-task, and hierarchical IL. **MAIRL** (Sun et al. (2021)) simply replaces the GAIL objective in MGAIL (introduced in Table 3) with the AIRL's and gets a model-based version of AIRL. Following the same intuition as DGAIL and SAM (introduced in Section 4.2.2), **Off-policy-AIRL** suggests adopting off-policy RL algorithms (specially, SAC (Haarnoja et al. (2018))) to train the generator $\pi$ in AIRL for improved sample efficiency. Table 4 introduces MH-AIRL, which extends AIRL to support both multi-task and hierarchical learning. As related works, **PEMIRL** (Yu et al. (2019b)) and **SMILe** (Ghasemipour et al. (2019a)) are proposed for multi-task AIRL, while **oIRL** (Venuto et al. (2020)) and **H-AIRL** (Chen et al. (2023d)) focus on hierarchical AIRL.

### 4.2.4 Integrating VAEs and GANs for Imitation Learning: A Spotlight

Lastly, we spotlight some papers that utilize VAEs to enhance GAIL, serving as examples of integrating different generative models for advanced imitation learning. GANs are recognized for their capacity to generate sharp image samples, as opposed to the blurrier samples from VAE models. However, GANs, unlike VAEs, are susceptible to the mode collapse problem, meaning that they may only capture partial modes in a multi-modal dataset. Therefore, the abilities of GANs and VAEs are highly complementary. As mentioned in Table 3, one approach, such as CGAIL and InfoGAIL, for solving the mode collapse issue is to train a discriminator and policy for each mode by conditioning them on the mode embedding. Such embeddings could be acquired from a pretrained VAE. In particular, a VAE $(P_\phi, P_\theta)$ can be trained to

imitate expert trajectories $\tau = (s_0, a_0, \cdots, s_T) \sim D_E$ through an ELBO objective:

$$\max_{\theta,\phi} \mathbb{E}_{\tau \sim D_E} \left[ \mathbb{E}_{c \sim P_\phi(\cdot|\tau)} \left[ \log P_\theta(\tau|c) \right] - D_{KL}(P_\phi(c|\tau)||P_C(c)) \right] \tag{54}$$

where $P_C(c)$ is the predefined prior distribution, $P_\phi(c|\tau)$ and $P_\theta(\tau|c)$ work as the encoder and decoder of the VAE. $\log P_\theta(\tau|c)$ can be decomposed as $\log \rho_\theta(s_0|c) + \sum_{t=0}^{T-1} \left[ \log \pi_\theta(a_t|s_t, c) + \log \mathcal{T}_\theta(s_{t+1}|s_t, a_t, c) \right]$ based on the MDP model. Through this unsupervised training process, the VAE is expected to capture the multiple modes in the dataset and its encoder $P_\phi(c|\tau)$ can be employed to provide mode embeddings for the expert trajectories. Then, the mode-conditioned discriminator and policy can be trained via an objective similar with CGAIL:

$$\min_\pi \max_D \mathbb{E}_{\tau \sim D_E, c \sim P_\phi(\cdot|\tau)} \left[ \frac{1}{T} \sum_{t=0}^{T-1} \log D(s_t, a_t|c)] + \mathbb{E}_{\rho_\pi(\cdot|c)}[\log(1 - D(s, a|c))] - \lambda H(\pi(\cdot|c)) \right] \tag{55}$$

The pretrained VAE policy $\pi_\theta(a|s, c)$ can be used to initialize the GAIL policy $\pi(a|s, c)$. Through further training with GAIL (i.e., Eq. (55)), the VAE policy can be refined by addressing the tendency to produce blurry imitations. **Diverse GAIL** (Wang et al. (2017)) and **VAE-ADAIL** (Lu & Tompson (2020)) are notable examples of this paradigm. A more straightforward manner to integrate VAEs and GANs for IL is adding a regularizer $\min_\pi \mathbb{E}_{s \sim \rho_\phi(\cdot)} D_{KL}(\pi(a|s)||\pi_{\text{VAE}}(a|s))$ to the GAIL objective, where $\pi_{\text{VAE}}(a|s)$ is the VAE-based policy pretrained on $D_E$ (see Section 3.3.1). In this way, the GAIL policy $\pi$ is encouraged to be close to the potentially multi-modal VAE policy. **SAIL** (Liu et al. (2020)) explores in this direction. Another significant advantage of VAEs over GANs is that they can provide semantically meaningful latent embeddings for the input data. As introduced in Section 3.2.3, such embeddings can be used as transformed training data to lower the difficulty of GAIL training. As an example, **LAPAL** (Wang et al. (2022b)) proposes to adopt a CVAE to transform high-dimensional actions into low-dimensional latent vectors and apply GAIL on the latent space instead, in order to stabilize and accelerate the learning process.

### 4.3 Generative Adversarial Networks in Offline Reinforcement Learning

For model-based offline RL (i.e., Section 2.1.2), both a policy and a world model of the environment need to be learned from the offline dataset. The generator $G$ in a GAN framework can be trained to approximate the policy or environment model, as detailed in Section 4.3.1 and 4.3.2 respectively.

### 4.3.1 Policy Approximation using GANs

As in dynamic-programming-based offline RL, to avoid OOD states and actions, the learned policy $\pi_\theta$ should be close to the underlying behavior policy $\mu$, which is typically realized through introducing a regularization term. Thus, the off-policy learning objective in model-based offline RL (i.e., the first objective in Eq. (8)) can be improved as:

$$\max_\theta \lambda_Q \mathbb{E}_{s \sim D_{\text{aug}}, a \sim \pi_\theta(\cdot|s)} \left[ Q_\phi(s, a) \right] - f(\pi_\theta, \mu) \tag{56}$$

where $\lambda_Q > 0$ is the regularization coefficient. **SGBCQ** (Dong et al. (2023)) proposes to directly constrain the action distribution of $\pi_\theta$ at each state to be close to the one of $\mu$ by implementing $f(\pi_\theta, \mu)$ as $\mathbb{E}_{s \sim D_\mu} \left[ d(\pi_\theta(\cdot|s)||\mu(\cdot|s)) \right]$, where $d(\cdot)$ denotes a statistical divergence, resembling policy constraint methods (i.e., Eq. (2)). As introduced in Section 4.1, GANs can be used to model and minimize the JS divergence between two data distributions. Thus, SGBCQ gives out a practical framework to solve Eq. (56) based on a CGAN as below:

$$\max_D \mathbb{E}_{(s,a) \sim D_\mu}[\log D(a|s) + \mathbb{E}_{z \sim P_Z(\cdot)} \log(1 - D(G(z|s)))],$$
$$\min_G -\lambda_Q \mathbb{E}_{s \sim D_{\text{aug}}, z \sim P_Z(\cdot)} \left[ Q_\phi(s, G(z|s)) \right] + \mathbb{E}_{s \sim D_\mu, z \sim P_Z(\cdot)}[\log(1 - D(G(z|s)))] \tag{57}$$

Compared with the CGAN objective (i.e., Eq. (44)), $s$ and $a$ here work as the conditional information $y$ and data point $x$, respectively. Intuitively, the stochastic policy $a \sim \pi_\theta(\cdot|s)$ is implemented as a generator $a = G(z|s), z \sim P_Z(\cdot)$ and the generator is trained to maximize the expected Q-values and fool the discriminator

$D$ at the same time. Involving such a GAN training process encourages the policy to generate behaviors close to the demonstrated ones in distribution.

**SDM-GAN** (Yang et al. (2022b)) follows a similar protocol but chooses to regularize the stationary $(s, a)$ distribution induced by $\pi_\theta$ to be close to the $(s, a)$ distribution in $D_\mu$ to avoid OOD behaviors. Specifically, they define $f(\pi_\theta, \mu)$ in Eq. (57) as $d(\rho_\mu^{\mathcal{T}}(\cdot) \| \rho_{\pi_\theta}^{\mathcal{T}}(\cdot))$, where $\rho_\mu^{\mathcal{T}}(s, a)$ can be approximated as the empirical distribution of $(s, a)$ in $D_\mu$, $\rho_{\pi_\theta}^{\mathcal{T}}(s, a) = \lim_{T \to \infty} \frac{1}{T+1} \sum_{t=0}^{T} P(s_t = s, a_t = a | s_0 \sim \rho_0(\cdot), a_t \sim \pi_\theta(\cdot|s_t), s_{t+1} \sim \mathcal{T}(\cdot|s_t, a_t))$ is the stationary $(s, a)$ distribution of $\pi_\theta$ under the real dynamic $\mathcal{T}$. Assuming the dynamic function $\widehat{\mathcal{T}}$ is well-fitted (through the process shown in Section 2.1.2), $d(\rho_\mu^{\mathcal{T}}(\cdot) \| \rho_{\pi_\theta}^{\mathcal{T}}(\cdot))$ is upper bounded by: (Please refer to (Yang et al. (2022b)) for the derivation and definitions of the function classes $\mathcal{G}_{1,2}$.)

$$\sup_{g \sim \mathcal{G}_1} |\mathbb{E}_{(s,a) \sim \rho_\mu^{\mathcal{T}}(\cdot)}[g(s,a) - \mathbb{E}_{s' \sim \widehat{\mathcal{T}}(\cdot|s,a), a' \sim \pi_\theta(\cdot|s')} g(s',a')]| + \sup_{g \sim \mathcal{G}_2} |\mathbb{E}_{(s,a) \sim \rho_\mu^{\mathcal{T}}(\cdot)}[g(s,a) - \mathbb{E}_{s \sim \rho_\mu^{\mathcal{T}}(\cdot), a \sim \pi_\theta(\cdot|s)} g(s,a)]| \tag{58}$$

The equation above is in a similar form with the Wasserstein distance (defined in Section 4.1) between samples stored in $D_\mu$ and generated by $\pi_\theta$, which inspires the use of GANs for the estimation. In particular, the objective is as below:

$$\max_D \mathbb{E}_{(s,a) \sim D_\mu}[\log D(s,a)] + \mathbb{E}_{(s,a) \sim D_G}[\log(1 - D(s,a))],$$
$$\min_G -\lambda_Q \mathbb{E}_{s \sim D_{\text{aug}}, z \sim P_Z(\cdot)}[Q_\phi(s, G(z|s))] + \mathbb{E}_{(s,a) \sim D_G}[\log(1 - D(s,a)] \tag{59}$$

Here, inspired by Eq. (58), $D_G$ contains $(s, a)/(s', a')$ pairs collected in this way: $s \sim D_\mu, a \sim \pi_\theta(\cdot|s), s' \sim \widehat{\mathcal{T}}(\cdot|s, a), a' \sim \pi_\theta(\cdot|s')$, where $a \sim \pi_\theta(\cdot|s)$ is implemented as $a = G(z|s), z \sim P_Z(\cdot)$. The only difference between Eq. (57) and (59) is thus the way to generate samples. For Eq. (57), the $(s, a)$ pairs are simply generated by $s \sim D_\mu, a \sim \pi_\theta(\cdot|s)$. The same authors have proposed a model-free variant of this algorithm (Yang et al. (2022c)), sharing similar motivations and designs.

**AMPL** (Yang et al. (2022d)) adopts the same objective (i.e., Eq. (59)) for training $\pi_\theta$, but proposes that the environment models $\widehat{\mathcal{T}}$ and $\hat{r}$ can be periodically updated by collecting new data with $\pi_\theta$ and applying supervised learning (e.g., with Eq. (7)) to update $\widehat{\mathcal{T}}$ and $\hat{r}$. Interestingly, they theoretically show that the objective for $\pi_\theta$, which combines the Q-function and distribution discrepancy, approximates an upper bound for $-J(\pi_\theta, \mathcal{M})$, i.e, the negative expected return of the policy on the real MDP. On the other hand, **DASCO** (Vuong et al. (2022)) proposes that the two terms in the objective of $\pi_\theta$ may conflict with each other, since fooling the discriminator requires mimicking all in-distribution actions, which can be suboptimal, but maximizing the Q function would mean avoiding low-return behaviors. Thus, they propose to introduce an auxiliary generator $G_{\text{aux}}$ to generate low-return samples and modify the objective as follows:

$$\min_{G, G_{\text{aux}}} \max_D \mathbb{E}_{(s,a) \sim D_\mu}[\log D(s,a)] + \mathbb{E}_{(s,a) \sim D_{G,G_{\text{aux}}}}[\log(1 - D(s,a))] - \lambda_Q \mathbb{E}_{s \sim D_{\text{aug}}, z \sim P_Z(\cdot)}[Q_\phi(s, G(z|s))] \tag{60}$$

Here, samples in $D_{G,G_{\text{aux}}}$ are generated through: $s \sim D_\mu, z \sim P_Z(\cdot), a = (G(z|s) + G_{\text{aux}}(z|s))/2$. Both $G$ and $G_{\text{aux}}$ are adopted to mimic the demonstrations, but only the primary generator $G$ is trained to maximize the Q-values. Theoretically, they prove that the optimal solution for the primary generator $G$ will maximize the probability mass of in-support samples that maximize the Q-function.

All aforementioned algorithms utilize a very similar objective design, which can be viewed as **a paradigm of using GANs for policy approximations in (model-based) offline RL**. **GOPlan** (Wang et al. (2023a)) proposes an advantage-weighted CGAN objective, which is different in form from previous ones, for capturing the multi-modal action distribution in goal-conditioned offline planning:

$$\min_G \max_D \mathbb{E}_{g \sim D_\mu} \left[ \mathbb{E}_{(s,a) \sim D_\mu^g}[w(s,a,g) \log D(s,a|g)] + \mathbb{E}_{s \sim D_\mu^g, z \sim P_Z(\cdot)}[\log(1 - D(s, G(z|s)|g))] \right] \tag{61}$$

Here, $D_\mu^g$ denotes the partition of $D_\mu$ that takes $g$ as the goal; $w(s, a, g) = \exp(A^\mu(s, a, g))$ denotes the weight function, where $A^\mu(s, a, g)$ is a separately trained advantage function. This mechanism encourages the policy to produce actions that closely resemble high-quality (i.e., high-advantage) actions from the offline dataset, but no theoretical guarantee is provided. Although algorithms introduced in this category are proposed for model-based offline RL, these methods can be transferred to the model-free setting seamlessly.

### 4.3.2 World Model Approximation using GANs

Beyond policy modeling, GANs can also be used to approximate the environment model $\widehat{\mathcal{M}}$, as illustrated in MOAN (Yang et al. (2023a)), TS (Hepburn & Montana (2022)), S2P (Cho et al. (2022)), which is expected to outperform traditional supervised learning methods (i.e., using Eq. (7)). To be specific, **MOAN** proposes the following objective:

$$\max_D \mathbb{E}_{(s,a,r,s')\sim D_\mu}[\log D(s,a,r,s')] + \mathbb{E}_{(s,a)\sim D_\mu,(\hat{r},\hat{s}')\sim G(\cdot|s,a)}[\log(1 - D(s,a,\hat{r},\hat{s}'))],$$
$$\min_G -\lambda_N \mathbb{E}_{(s,a,r,s')\sim D_\mu}\left[-\log G(r,s'|s,a)\right] + \mathbb{E}_{(s,a)\sim D_\mu,(\hat{r},\hat{s}')\sim G(\cdot|s,a)}[\log(1 - D(s,a,\hat{r},\hat{s}'))] \tag{62}$$

This design is similar with Eq. (59) in form, but replaces the Q-function term with a negative log-likelihood (i.e., $-\log G(r,s'|s,a)$) that is commonly used for environment model approximation. In this CGAN framework, the conditional generator $G(\cdot|s,a)$ is learned to predict $s'$ and $r$, working as the approximated reward $\hat{r}(s,a)$ and transition function $\widehat{\mathcal{T}}(s'|s,a)$. In particular, the generator $G(\cdot|s,a) = \mathcal{N}(\text{mean}(s,a),\text{std}(s,a))$ is implemented as a stochastic Gaussian model, and the standard deviation of the predictions, i.e., $\text{std}(s,a)$, can be used as uncertainty measure (i.e., $u(s,a)$ in Section 2.1.2). MOAN proposes to further include the confidence level given by the discriminator, i.e., $u(s,a) = \text{std}(s,a) + \sqrt{2(1 - D_{(\hat{r},\hat{s}')\sim G(\cdot|s,a)}(s,a,\hat{r},\hat{s}'))}$. Intuitively, if the variance of the estimation is high or the discriminator classifies the prediction as generated data, i.e., $D \to 0$, the uncertainty level at this point $(s,a)$ should be high. However, this uncertainty measure design is not backed up by theoretical analysis.

**TS** proposes to learn environment models for stitching high-value segments from different trajectories to form higher-quality trajectories for offline RL. To realize this, they model the distribution $P(a,r,s'|s) = P(s'|s)P(a|s',s)P(r|a,s',s)$. That is, they search for a next state $s'$ from the neighbourhood of $s$, which can come from a different trajectory and should have a higher probability $P(s'|s)$ & value $V(s')$ than the original next state of $s$. Then, they identify the most probable intermidiate $a$, $r$. In particular, they learn $P(s'|s)$, $P(a|s',s)$, $P(r|a,s',s)$ with the traditional supervised learning, CVAE, and WGAN (see Section 4.1), respectively. The WGAN objective is: ($G(z|s,a,s')$ works as the approximation of $P(r|a,s',s)$.)

$$\min_G \max_D \mathbb{E}_{(s,a,s',r)\sim D_\mu}[D(s,a,s',r)] - \mathbb{E}_{(s,a,s')\sim D_\mu,z\sim P_Z(\cdot)}[D(s,a,s',G(z|s,a,s'))] \tag{63}$$

This algorithm can be viewed as a model-based data augmentation method for offline RL using GANs. **S2P**, on the other hand, is a model learning approach for vision-based offline RL. It first approximates the transition and reward model (i.e., $\widehat{T}(s'|s,a)$, $\hat{r}(s,a)$) for the underlying states through traditional supervised learning, and then learns a generator to synthesize the image $I/I'$ that perfectly represents the corresponding state $s/s'$. The generator is conditioned on both the current state and previous image, i.e., $I' = G(z|s,I)$, and trained within a WGAN framework. Here, the GAN is used for data transformation to simplify the (vision-based) environment model approximation.

In Table 5, we provide a summary of GAN-based offline RL algorithms. GANs can be employed for both policy and world model approximations in model-based offline RL, primarily due to their ability to match a distribution (hidden in the offline data) by minimizing some statistical discrepancy.

## 5 Normalizing Flows in Offline Policy Learning

Applications of Normalizing Flows (NFs) in offline policy learning are relatively few, particularly for NF-based offline RL. Thus, in Section 5.3, we slightly broaden the scope to include NF-based online RL methods that utilize offline data.

### 5.1 Background on Normalizing Flows

Normalizing Flows (NFs, Kobyzev et al. (2021)) allow for generating data samples following complex distributions and exact density estimation of given data points, which are essential in probabilistic modeling tasks. Formally, NFs transform a simple, known probability distribution in the latent space (i.e., $z \sim P_Z(\cdot)$) to a complex, desired data distribution (i.e., $x \sim P_X(\cdot)$) through a sequence of **invertible and differentiable**

| Algorithm | GAN Type | GAN Usage | Evaluation Task |
|---|---|---|---|
| SGBCQ | CGAN | Policy | D4RL (L) |
| SDM-GAN | GAN | Policy | D4RL (L, M, A) |
| AMPL | GAN | Policy | D4RL (L, M, A) |
| DASCO | GAN (auxiliary generator) | Policy | D4RL (L, M) |
| GOPlan | CGAN (advantage-weighted) | Policy | MuJoCo Robotic Manipulation |
| MOAN | GAN | Environment Model | D4RL (L) |
| TS | WGAN | Environment Model | D4RL (L) |
| S2P | WGAN | Environment Model | dm_control |

Table 5: Summary of GAN-based offline RL algorithms. Most algorithms in this category have been evaluated on D4RL (Fu et al. (2020)), which provides multiple datasets for data-driven RL tasks, including Locomotion (L), AntMaze (M), Adroit (A), etc. MuJoCo Robotic Manipulation (Yang et al. (2023b)) provides offline datasets for a series of goal-conditioned robotic manipulation tasks, several of which can be utilized to assess the OOD generalization capabilities of the policy. dm_control (Tassa et al. (2018)) includes 6 environments, which are typically used for evaluating vision-based RL algorithms.

transformations. The mathematical representation of the transformation from $x$ to $z$ is given by a series of bijections: $F = F_N \circ F_{N-1} \circ \cdots \circ F_1$. The data flow $[x_0, x_1, \cdots, x_N]$ ($x_0 = x$, $x_N = z$) within NFs adheres to the following process: for all $1 \leq i \leq N$, $x_i = F_i(x_{i-1})$, $x_{i-1} = F_i^{-1}(x_i)$. The core principle behind the transformation is the change of variable formula. Specifically, the probability density functions (PDF) of $x$ and $z$ are related as:

$$P_Z(z) = P_X(F^{-1}(z))|\det D(F^{-1}(z))|, \ P_X(x) = P_Z(F(x))|\det D(F(x))| \tag{64}$$

where $\det D(F(x))$ denotes the determinant of the Jacobian matrix of $F(x)$. Given the transformation function $F$ and the basic distribution $P_Z$, the density of a data sample $x$ can be exactly acquired as $P_X(x)$ defined in Eq. (64). Thus, the ability in exact density estimation of NFs relies on the fact that $F$ is invertible and differentiable. Conversely, the data generation can be done by first sampling a latent $z \sim P_Z(\cdot)$ and then apply the generator function $G = F^{-1}$. As for training, it is usually through maximizing the log-likelihood of the target data distribution (Dinh et al. (2017)):

$$\max_F \mathbb{E}_X[\log(P_X(x))], \ \log(P_X(x)) = \log(P_Z(F(x))) + \log(|\det D(F(x))|) \tag{65}$$

For NFs, the generator $G$ should be sufficiently expressive to accurately model the distribution of interest, while the transformation $F$ should be invertible and the computation of the two directions mentioned above (i.e., data generation and density estimation), particularly the determinant calculation, must be efficient. Based on these requirements, various types of flows have been developed. Here, we briefly present some representative variants. As mentioned above, $G$ is composed of a series of generator modules $G_i = F_i^{-1}$. For each variant, we introduce its specific design of $G_i$, and we use $m$ and $n$ to denote the input and output of $G_i$, respectively. Note that we do not present complex mathematical details regarding calculations or objective designs, as they are not essential for understanding the applications of NFs in offline policy learning, which are introduced in Section 5.2 and 5.3.

- **Coupling Flows** enable highly expressive transformations via a coupling method. In particular, the input $m \in \mathbb{R}^D$ is partitioned into two subspaces: $(m^A, m^B) \in \mathbb{R}^d \times \mathbb{R}^{D-d}$, and the generator module $G_i$ is defined in the format: $G_i(m) = (H(m^A; \Theta(m^B)), m^B)$. Here, $H(\cdot; \theta) : \mathbb{R}^d \to \mathbb{R}^d$ is a bijection, and $\Theta$ is the parameter function. $H$ of such form is also called the coupling function. In this case, the Jacobian of $G_i$ is simply the Jacobian of $H$, and for efficient computation of determinants, $H$ is usually designed to be an element-wise bijection, i.e., $H(m^A; \theta) = (H_1(m_1^A; \theta_1), \cdots, H_d(m_d^A; \theta_d))$ where each $H_i(\cdot; \theta_i) : \mathbb{R} \to \mathbb{R}$ is a scalar bijection. The expressiveness of a coupling flow lies in the complexity of its parameter function $\Theta$, which, in practice, is usually modeled as neural networks. Algorithms in this category include NICE (Dinh et al. (2015)), RealNVP (Dinh et al. (2017)), Glow (Kingma & Dhariwal (2018)), NSF (Durkan et al. (2019)), Flow++ (Ho et al. (2019)), etc.

- **Autoregressive Flows** utilize autoregressive models for generation. To be specific, each entry of $n = G_i(m)$ conditions on the previous entries of the input: $n_t = H(m_t; \Theta_t(m_{1:t-1}))$, where $H(\cdot; \theta) : \mathbb{R} \to \mathbb{R}$ is a scalar bijection and $\Theta_t$ is a parameter function. In this case, each $n_t$ relies solely on $m_{1:t}$, resulting in a triangular Jacobian maxtrix of $G$. This structure allows for efficient computation of the determinant and parallel computation of each entry in $n$. However, the inverse process, represented as $m_t = H^{-1}(n_t; \Theta_t(m_{1:t-1}))$, must be performed sequentially, since the parameter for the subsequent step (i.e., $\Theta_{t+1}(m_{1:t})$) uses the output of the current step, i.e., $m_t$, as input. Algorithms like IAF (Kingma et al. (2016)), MAF (Papamakarios et al. (2017)), and NAF (Huang et al. (2018)), fall in this category.

- **Residual Flows** employ residual networks (He et al. (2016)) as the generator component, characterized by $G_i(m) = m + R(m)$. Here, $R(\cdot)$ is the residual block, which can be implemented as any type of neural network that is invertible. Several studies (Gomez et al. (2017); Chang et al. (2018); Jacobsen et al. (2018)) have tried to develop invertible network architectures suitable for use as residual blocks. Nonetheless, these architectures present a significant challenge in efficiently calculating the Jacobian determinants. Research works such as iResNet (Behrmann et al. (2019)) and Residual Flow (Chen et al. (2019)) propose an invertible $G_i$ by limiting the Lipschitz constant of the residual block to be less than 1 and compute its inverse through fixed-point iterations. However, controlling the Lipschitz constant of a neural network is also quite challenging.

- **ODE-based Flows** aim to learn a continuous dynamic system shown as a first-order ordinary differential equation (ODE): $\frac{d}{dt}m(t) = R(m(t); \theta(t))$, $t \in [0, 1]$. It's worthy noting that the residual flows mentioned above can be interpreted as a discretization of this ODE. Assuming the uniform Lipschitz continuity in $m$ and setting the initial condition as $m(0) = z$, the solution of that ODE at each time point $\Phi^t(z)$ exists and is unique (Arnold (1992)). As proposed in NODE (Chen et al. (2018b)), the map at time one $\Phi^1(z)$ can serve as the generator, i.e., $x = \Phi^1(z)$, which is equivalent to an "infinitely deep" neural network with stacked weights $\theta(t), t \in [0, 1]$. Thus, typically, continuous ODE-type flows require fewer parameters to match the performance of discrete ones (Grathwohl et al. (2018)). As for the invertibility of $\Phi^1$, it is naturally ensured by the theorem of existence and uniqueness pertaining to the solution of the ODE. However, $\Phi^1$ must be orientation preserving - that is, its Jacobian determinant must be positive, which may constrain its representational capacity. In order to give freedom for the Jacobian determinant to remain positive, the authors of ANODE (Dupont et al. (2019)) propose augmenting the original ODE with supplementary variables $\hat{m}(t)$, resulting in $\frac{d}{dt}[m(t)||\hat{m}(t)] = \hat{R}(m(t)||\hat{m}(t); \theta(t))$, $t \in [0, 1]$, with $||$ signifying concatenation.

There are other types of flows which are not widely used in practice and so not introduced here, such as Planar and Radial Flows (Rezende & Mohamed (2015); van den Berg et al. (2018)) and Langevin Flows (continuous and SDE-based) (Welling & Teh (2011); Chen et al. (2018a)). Coupling Flows and Autoregressive Flows facilitate straightforward invertible transformations and efficient determinant computation, yet lag in expressiveness compared to Residual Flows which are computationally costly. ODE-based Flows provide a potentially elegant and parsimonious representation with fewer parameters. However, they necessitate the resolution of ODEs during training, a process that can be computationally demanding and sensitive to the choice of numerical solver settings. Compared with VAEs and GANs, NFs allow for exact density estimation of a generated sample and can avoid training issues like mode collapse, vanishing gradients, etc (Salimans et al. (2016)). However, its requirements for bijection functions and determinant calculations can restrict its capacity in efficiently modeling complex data distributions (Cornish et al. (2020)).

## 5.2 Normalizing Flows in Imitation Learning

NF-based IL algorithms either employ NFs for exact density estimation or leverage NFs' proficiency in modelling complex policies to manage challenging task scenarios, which correspond to the normalizing direction (i.e., $x \to z$) and the generation direction (i.e., $z \to x$) of NFs respectively. Next, we present a comprehensive review of works from both categories.

### 5.2.1 Exact Density Estimation with Normalizing Flows

As a paradigm, IL methods that adopt NFs as density estimators usually start with standard IL objectives, and optimize them as RL problems wherein the rewards necessitate estimation of specific distributions. In this case, NFs are introduced to model those distributions and provide exact density inference for reward calculation, which greatly enhance the training stability and performance. To be specific, we present some representative works here.

As proposed in (Ghasemipour et al. (2019b)), most IL methods can be viewed as matching the agent's state-action distribution with the expert's, by minimizing some f-divergence $D_f$. As an example, **CFIL** (Freund et al. (2023)) chooses to minimize the KL divergence between the agent's and expert's state-action occupancy measure (as defined in Section 4.2.1):

$$\arg\min_{\pi} D_{KL}(\rho_\pi(s,a)||\rho_E(s,a)) = \arg\max_{\pi} \mathbb{E}_{(s,a)\sim\rho_\pi(\cdot)}\left[\log\frac{\rho_E(s,a)}{\rho_\pi(s,a)}\right] = \arg\max_{\pi} J(\pi, r = \log\frac{\rho_E}{\rho_\pi}) \quad (66)$$

where $J(\pi, r)$ denotes the expected return (Sutton & Barto (2018)) of the policy $\pi$ under the reward function $r$. An MAF (Masked Autoregressive Flow, Papamakarios et al. (2017)) is used to model the distributions $\rho_E(s,a)$ and $\rho_\pi(s,a)$ based on which the rewards can be acquired. Specifically, $(s,a)$ can be viewed as $x$ in the NF framework and the densities at $(s,a)$, i.e., $\rho_E(s,a)$ and $\rho_\pi(s,a)$, can be estimated using Eq. (64). However, rather than optimizing these two NFs independently with corresponding MLE objectives (i.e., Eq. (65)), they couple the modelling of these two distributions based on the optimality point of the Donsker-Varadhan form of the KL divergence (Donsker & Varadhan (1976)):

$$D_{KL}(\rho_\pi(s,a)||\rho_E(s,a)) = \sup_{f:S\times A\to\mathbb{R}} \mathbb{E}_{(s,a)\sim\rho_\pi(\cdot)}[f(s,a)] - \log\mathbb{E}_{(s,a)\sim\rho_E(\cdot)}\left[e^{f(s,a)}\right] \quad (67)$$

In the equation above, optimality occurs at $f^*(s,a) = \log\frac{\rho_\pi(s,a)}{\rho_E(s,a)} + C$ where $C \in \mathbb{R}$. Thus, through maximizing the right-hand side of the equation above, the log ratio used as the reward function can be recovered. Specifically, they model $f$ as $f_{\psi,\phi}(s,a) = \log\rho_\pi^\psi(s,a) - \log\rho_E^\phi(s,a)$ with the two flows $\rho_\pi^\psi$ and $\rho_E^\phi$. Given this improved estimator, the overall IL objective can be written as: (which combines Eq. (66) and (67))

$$\arg\max_{\pi} \min_{\rho_\pi^\psi,\rho_E^\phi} \log\mathbb{E}_{(s,a)\sim D_E(\cdot)}\left[\frac{\rho_\pi^\psi(s,a)}{\rho_E^\phi(s,a)}\right] - \mathbb{E}_{(s,a)\sim\rho_\pi(\cdot)}\left[\log\frac{\rho_\pi^\psi(s,a)}{\rho_E^\phi(s,a)}\right] \quad (68)$$

In this case, the learning alternates between $\max_\pi$ and $\min_{\rho_E^\phi,\rho_\pi^\psi}$. $\max_\pi$ is realized with an RL process using $\log\frac{\rho_E^\psi(s,a)}{\rho_\pi^\phi(s,a)}$ as the reward function. While, $\rho_E^\psi(s,a)$ and $\rho_\pi^\phi(s,a)$ are updated based on the expert data (i.e., $(s,a) \sim D_E$) and rollout data collected by $\pi$ (i.e., $(s,a) \sim \rho_\pi(\cdot)$).

**IL-flOw** (Chang et al. (2022)) focuses on Learning from Observations (LfO), where the agent only gets access to a dataset of state sequences. This work also starts with the KL divergence but processes to decouple the policy optimization from the reward learning to avoid minimax optimization as in Eq. (68) and improve the training stability. In particular, the learning objective for the policy $\pi$ is to maximize $-D_{KL}(P_\pi(s'|s)||P_E(s'|s))$ which equals to:

$$\mathbb{E}_{s_{0:T}\sim P_\pi}\left[\sum_{t=0}^{T-1}(\log P_E(s_{t+1}|s_t) - \log P_\pi(s_{t+1}|s_t))\right] = \mathbb{E}_{s_{0:T}\sim P_\pi}\left[\sum_{t=0}^{T-1}(\log P_E(s_{t+1}|s_t) + H(\pi(\cdot|s_t)))\right] \quad (69)$$

The equality holds when environment dynamics are deterministic and invertible. The right hand side of Eq. (69) can be optimized with RL by setting the reward as $r_t = \log P_E(s_{t+1}|s_t)$ while maximizing the entropy of the policy, i.e., $H(\pi(\cdot|s_t))$. As a separate stage, before the RL training, $P_E(s'|s)$ is modeled with a conditional NF (i.e., conditioning the transformation function $F$ on a fixed state variable $s$: $s' = F(z|s)$, $z \sim P_Z(\cdot)$) based on the demonstrations $D_E$. Specifically, they select NSF (Durkan et al. (2019)) as the density estimator.

**SOIL-TDM** (Boborzi et al. (2021b; 2022)) also focuses on LfO and starts with the same objective as IL-flOw. However, SOIL-TDM gets rid of the requirements for deterministic and invertible dynamics by estimating

$P_\pi(s_{t+1}|s_t)$ as $\mathcal{T}_\pi(s_{t+1}|s_t, a_t)\pi(a_t|s_t)/\pi'(a_t|s_{t+1}, s_t)$, based on the Bayes Theorem. With this new definition, the objective for $\pi$ is converted into:

$$\max_\pi \mathbb{E}_{(s_{0:T}, a_{0:T}) \sim P_\pi} \sum_{t=0}^{T-1} [r(s_t, a_t) + H(\pi(\cdot|s_t))] \tag{70}$$

where $r(s_t, a_t) = \mathbb{E}_{s_{t+1} \sim \mathcal{T}_\pi(\cdot|s_t, a_t)} \left[ -\log \mathcal{T}_\pi(s_{t+1}|s_t, a_t) + \log \pi'(a_t|s_{t+1}, s_t) + \log P_E(s_{t+1}|s_t) \right]$. In this case, besides the expert state transition model $P_E(s_{t+1}|s_t)$, they adopt conditional NFs to model the agent's transition dynamics $\mathcal{T}_\pi(s_{t+1}|s_t, a_t)$ and the posterior distribution associated with $\pi$, i.e., $\pi'(a_t|s_{t+1}, s_t)$, for which they choose Real NVP (Dinh et al. (2017)). Note that the approximation of $P_E(s_{t+1}|s_t)$ is trained on $D_E$, while approximations of $\mathcal{T}_\pi(s_{t+1}|s_t, a_t)$ and $\pi'(a_t|s_{t+1}, s_t)$ are trained on rollout data collected by $\pi$ during the RL process.

**SLIL** (Zhang et al. (2021b)) is based on the Likelihood-based Imitation Learning (LIL) framework:

$$\max_\pi \mathbb{E}_{(s,a) \sim D_E} [\log P_\pi(s, a)] \; s.t. \; P_\pi = \arg\max_P \mathbb{E}_{(s,a) \sim \rho_\pi} [\log P(s, a)] \tag{71}$$

Intuitively, $P_\pi(s, a)$ represents the probability of observing an expert state-action pair $(s, a) \in D_E$ when executing the learner policy $\pi$. Directly solving this bilevel optimization problem requires jointly learning $\pi$ and $P_\pi$, which brings training instability. Thus, they propose to maximize its tight lower bound:

$$\max_\pi \mathbb{E}_{(s,a) \sim D_E} [\log \pi(a|s)] + \mathbb{E}_{s \sim \rho_\pi} [\log P_E(s)] \; s.t. \; P_E = \arg\max_P \mathbb{E}_{s \sim D_E} [\log P(s)] \tag{72}$$

where $\rho_\pi(s)$ denotes the occupancy measure of $s$ induced by the policy $\pi$. The training process can then be divided into two stages. At stage 1, they train an NF to model the expert state distribution $P_E(s)$ based on the demonstration set $D_E$, which is, at stage 2, adopted to optimize the policy $\pi(a|s)$ using Eq. (72). Intuitively, this objective combines Behavioral Cloning (i.e., $\max_\pi \mathbb{E}_{(s,a) \sim D_E} [\log \pi(a|s)]$) and RL with $\log P_E(s)$ as the reward (i.e., $\max_\pi \mathbb{E}_{s \sim \rho_\pi} [\log P_E(s)]$). It is worthy noting that they utilize a Continuous Normalizing Flow – DCNF (Zhang et al. (2021b)) as density estimators, due to the enhanced modeling capabilities and fewer model restrictions that Continuous Normalizing Flows offer Grathwohl et al. (2018).

**GPRIL** (Schroecker et al. (2019)) also utilizes the LIL framework but optimizes it in a different manner:

$$\max_{\pi_\theta} \mathbb{E}_{(s,a) \sim D_E} [\log P_{\pi_\theta}(s, a)] = \max_{\pi_\theta} \mathbb{E}_{(s,a) \sim D_E} [\log \pi_\theta(a|s) + \log P_{\pi_\theta}(s)] \tag{73}$$

Here, the first term of the right-hand side is simply Behavioral Cloning. While, for the second term, according to (Ross & Bagnell (2010)), the gradient from it can be estimated as $\nabla_\theta \log P_{\pi_\theta}(s) \propto \mathbb{E}_{(s'', a'') \sim \mathcal{B}_{\pi_\theta}(\cdot|s)} [\nabla_\theta \log \pi_\theta(a''|s'')]$. $\mathcal{B}_{\pi_\theta}(\cdot|s)$ models the distribution of states and actions that, following the current policy $\pi_\theta$, will eventually reach the given state $s$. That is, $\mathcal{B}_{\pi_\theta}(s'', a''|s) = (1 - \gamma) \sum_{j=0}^{\infty} \gamma^j P_{\pi_\theta}(s_t = s'', a_t = a''|s_{t+j+1} = s)$. In GPRIL, $\mathcal{B}_{\pi_\theta}(s'', a''|s)$ is modeled as $\mathcal{B}_{\pi_\theta}^{\omega_1}(s''|s) \cdot \mathcal{B}_{\pi_\theta}^{\omega_2}(a''|s, s'')$, i.e., the product of two density functions each of which is modelled by a conditional MAF (Papamakarios et al. (2017)). To train these models, they collect training data using self-supervised roll-outs. In particular, they sample states, actions and target-states (i.e., $(s'', a'', s)$), where the separation in time between the $s''$ and $s$ is selected randomly from a geometric distribution parameterized by $\gamma$. In this way, the collected triplets theoretically satisfy: $(s'', a'') \sim \mathcal{B}_{\pi_\theta}(\cdot|s)$. As a subsequent research, the work (Schroecker & Jr. (2020)) utilizes the same objective function, but introduces an alternative method for estimating $\nabla_\theta \log P_{\pi_\theta}(s)$ from a goal-conditioned RL perspective. However, this algorithm is still in an early stage of development.

### 5.2.2 Policy Approximation using Normalizing Flows

As a potent generative model, Normalizing Flows can be directly adopted as policy networks for tackling challenging tasks. In this regard, recent works in off-policy RL have demonstrated that NFs can be used to model continuous policies (Haarnoja et al. (2018); Ward et al. (2019); Mazoure et al. (2019)), which can lead to faster convergence and higher returns by enhancing exploration and supporting multi-modal action

| Algorithm | Flow | Flow Type | Flow Usage | Objective Type | Evaluation Task |
|---|---|---|---|---|---|
| CFIL | MAF | AR | DE $(\rho_E(s,a),\ \rho_\pi(s,a))$ | KL | MuJoCo |
| IL-flOw | NSF | CP | DE $(P_E(s'\|s))$ | KL | MuJoCo |
| SOIL-TDM | Real NVP | CP | DE $(P_E(s'\|s),\ \mathcal{T}_\pi(s'\|s,a),\ \pi'(a\|s',s))$ | KL | MuJoCo |
| SLIL | DCNF | ODE | DE $(P_E(s))$ | LIL | MuJoCo |
| GPRIL | MAF | AR | DE $(\mathcal{B}_\pi(s'',a''\|s))$ | LIL | Robotic Insertion |
| Flow DAC | Real NVP | CP | G (Policy Network) | AIL | High-D & MujoCo |
| FlowPlan | NAF | AR | G (Trajectory Planner) | KL | HES-4D |

Table 6: Summary of NF-based IL algorithms. AR, CP, and ODE represent autoregressive, coupling, and ODE-based continuous flows, respectively. DE and G represent the two manners that NFs can be used, corresponding to the density estimator and generator, respectively. For DE, we specifically point out the densities estimated, for which the definitions are available in Section 5.2.1. Regarding the types of IL objectives, KL, LIL, and AIL denote the KL divergence, Likelihood-based IL, and Adversarial IL, respectively. Examples for AIL include GAIL and AIRL, which are introduced in Section 4.2.1. For the benchmarks, MuJoCo (Todorov et al. (2012)) provides a series of robotic locomotion tasks; Robotic Insertion (Vecerík et al. (2017)) is a specific type of robotic manipulation task built on the MuJoCo engine; High-D (Krajewski et al. (2018)) and HES-4D (Meyer et al. (2019)) are two real-word driving dataset.

distributions. Compared to commonly-employed diagonal Gaussian policies, where each action dimension is independent of the others, those based on NFs offer greater expressiveness.

In the context of IL, **Flow DAC** (Boborzi et al. (2021a)) extends DAC (an off-policy IL algorithm Kostrikov et al. (2019)) by adopting a conditional version of Real NVP as the policy network. Specifically, the action $a$ at state $s$ is generated by: $z \sim P_Z(\cdot)$, $a = G(z|s)$ where $G$ denotes the NF generator and $s$ is the conditioner. The stochasticity of this policy comes from the latent distribution $P_Z(\cdot)$, and the action output of $G$ can be in any complex distribution. Similarly, **FlowPlan** (Agarwal et al. (2020)) adopts NFs as a trajectory generator/planner through sequentially generating the next state embedding based on historical states, and an NAF (Huang et al. (2018)) is adopted as the generative model, which is trained by minimizing the KL divergence between the expert's and generated trajectories.

In Table 6, we provide a summary of NF-based IL algorithms. For each algorithm, we provide key information including what type of NF and how the NF is utilized, its underlying IL framework, and evaluation tasks.

### 5.3 Normalizing Flows for Reinforcement Learning with Offline Data

As mentioned in the beginning of Section 5, we broaden the scope of RL-related works to include both offline RL (Section 5.3.1) and online RL with offline data (Section 5.3.2). We believe that the approaches developed for using offline data in online RL could be effectively adapted to enhance offline RL. Similar with NF-based IL methods, algorithms in Section 5.3.1 and 5.3.2 can be categorized based on the usage of NFs, i.e., working as either a density estimator or a function generator.

### 5.3.1 Adopting Normalizing Flows in Offline Reinforcement Learning

As introduced in Section 2.1.1, major challenges in offline RL include (1) extrapolation error caused by approximating the value of state-action pairs not well-covered by the training data $D_\mu$ and (2) distributional

shift between behavior and inference policies. The common practice to tackle these problems is to induce conservatism, through either keeping the learned policy close to the behavior policy or constructing pessimistic value functions. However, this may lead to over-conservative and sub-optimal policies. By introducing Normalizing Flows, an agent can utilize offline data more effectively and circumvent out-of-distribution (OOD) actions through its generative (Akimov et al. (2022); Yang et al. (2023d)) or density estimation capacity (Zhang et al. (2023)).

CNF (Akimov et al. (2022)) and LPD (Yang et al. (2023d)) employ a similar algorithm idea – using NFs to extract latent variables underlying primitive actions in the offline dataset. Specifically, they train a Normalizing Flow $G$ conditioned on $s$ to model the mapping from $z$ to $a$, following $z \sim P_Z(\cdot)$, $a = G(z|s)$, through supervised learning (i.e., Eq. (65)). After training, $G$ can work as an action decoder to convert a latent variable $z$ (following a simple base distribution) to a primitive action. In this case, only a high-level policy $z \sim \pi_{\text{high}}(\cdot|s)$ needs to be learned with offline RL, and the function composition $G \circ \pi_{\text{high}}$ can work as a hierarchical policy mapping from $s$ to $a$. CNF and LPD are driven by different motivations. In particular, **CNF** pretrains a Real NVP conditioned on $s$ as the action decoder. To avoid OOD actions caused by the long tail effect, the latent space (i.e., the support of $P_Z(\cdot)$), which is also the output space of the high-level policy, is designed to be bounded, specifically an $n$-dim interval $(-1, 1)^n$. Through pretraining on $D_\mu$, each latent variable in this bounded space should correspond to an in-distribution action through the decoder $G$. As a result, the hierarchical policy (i.e., $G \circ \pi_{\text{high}}$) would not generate OOD actions, even without clipping the high-level policy's output (as in PLAS introduced in Section 3.2.2), which avoids possible sub-optimality. **LPD**, instead, focuses on improving the offline RL performance in challenging, long-horizon tasks, by adopting NFs to model the transformation from $z$ to a fixed-length sequence of actions starting from $s$, i.e., a skill. Provable benefits can be gained when the learned skills are expressive enough to recover the original policy space (Yang et al. (2023d)), and, as an advanced generative model, Normalizing Flows can facilitate the learning of such skills. With this pretrained transformation, the dataset can be relabeled in terms of skills as $D_{\text{high}} = [(s_0^i, z^i, \sum_{t=0}^{h-1} \gamma^t r_t^i, s_h^i)]_{i=1}^N$, where $h$ is the skill length and $z^i$ can be acquired as $G^{-1}(a_{0:h-1}^i|s_0^i) = F(a_{0:h-1}^i|s_0^i)$ (i.e., via the normalizing direction of the NF). Then, offline RL can be adopted to learn $\pi_{\text{high}(z|s)}$ from $D_{\text{high}}$. In this case, the decision horizon of the offline RL policy is reduced by a factor of $h$, which can effectively mitigate cumulative distribution shifts.

**APAC** (Zhang et al. (2023)) adopts NFs (specifically Flow-GAN (Grover et al. (2018))) as density estimators, rather than action decoders as in aforementioned algorithms, to filter out OOD actions while avoiding being over-conservative. To be specific, the density of the point $(s, a)$ can be approximated as $P_Z(G^{-1}(a|s))$, where $G$ is the NF: $a = G(z|s)$. Points with higher density are more likely to be generated by the behavior policy and so can be viewed as "safe" in-distribution points for training use. To be specific, they modify the policy improvement step in Eq. (1) as:

$$\pi_{k+1} = \arg\max_\pi \mathbb{E}_{s \sim D_\mu} \left[ \mathbb{E}_{a \sim \pi(\cdot|s)} Q_{k+1}^\pi(s, a) \right] \ \ s.t. \ P_Z(G^{-1}(a|s)) > \epsilon(s, a). \tag{74}$$

where $\epsilon(s, a)$ is a defined threshold. In this way, samples for updating the policy are restricted within the safe area, which could contain both observed and unobserved points in the offline dataset. Thus, the policy can potentially perform exploration out of the given dataset and avoid visiting OOD actions, simultaneously. Note that this constraint on $\pi$ is more relaxed and practical compared with the one of BEAR (introduced in Section 2.1.1): $\pi \in \{\pi' : \mathcal{S} \times \mathcal{A} \to [0, 1] \mid \pi'(a|s) = 0 \text{ whenever } \mu(a|s) < \epsilon\}$, to avoid being over-conservative.

### 5.3.2 Adopting Normalizing Flows in Online Reinforcement Learning with Offline Data

To deepen the understanding of applying NFs to RL, we incorporate another line of works on online RL utilizing offline data. Still, these works can be categorized based on the usage of NFs: potent generators (Mazoure et al. (2019); Yan et al. (2022); Slack et al. (2022)) or density estimators (Wu et al. (2021)).

With the same insights as Flow DAC (introduced in Section 5.2.2), **SAC-NF** (Mazoure et al. (2019)) extends SAC, an off-policy RL algorithm (Haarnoja et al. (2018)), by adopting NFs (IAF (Kingma et al. (2016))) as the policy network, which could potentially improve exploration and support multi-modal action distributions. On the other hand, CEIP (Yan et al. (2022)) and SAFER (Slack et al. (2022)) leverage the offline data by first learning a transformation from latent variables $z$ to actions $a$ using NFs as an action

| Algorithm | Flow | Flow Type | Flow Usage | Evaluation Task |
|---|---|---|---|---|
| CNF | Real NVP | CP | G (Action Decoder) | D4RL (L,M) |
| LPD | Real NVP | CP | G (Skill Decoder) | D4RL (M, A, K) |
| APAC | Flow-GAN | CP | DE ($P_G(s,a)$) | D4RL (L,M) |
| SAC-NF | IAF | AR | G (Policy Network) | MuJoCo |
| CEIP | Real NVP | CP | G (Action Decoder) | Kitchen, Office, FetchReach |
| SAFER | Real NVP | CP | G (Action Decoder) | Operation |
| NF Shaping | MAF | AR | DE ($P_G(s,a)$) | Robotic Insertion, Pick Place |

Table 7: Summary of NF-based (Offline) RL algorithms. AR and CP represent autoregressive and coupling flows, respectively. DE and G represent the two manners that NFs can be used, corresponding to the density estimator and generator, respectively. Specifically, $P_G(s,a)$ denotes the state-action pair distribution in the offline dataset. For the benchmarks, all the evaluation tasks shown in this table are built on the MuJoCo (Todorov et al. (2012)) engine; D4RL (Fu et al. (2020)) provides a bunch of challenging (robotic) tasks specifically for offline RL, including Locomotion (L), AntMaze (M), Adroit (A), Kitchen (K); Office (Pertsch et al. (2021)), FetchReach (Plappert et al. (2018)), Kitchen (from D4RL), Robotic Insertion, Pick Place (Vecerík et al. (2017)), and Operation (Slack et al. (2022)) all involve the manipulation of a robot arm.

decoder and then training a high-level policy $\pi_{\text{high}}(z|s)$ atop the transformation function using online RL, akin to CNF. Yet, in comparison to CNF, they extend the action decoder learning from different perspectives. In **CEIP**, the generator is conditioned on a concatenation of the current and next states, i.e., $u = [s, s']$, instead of the state alone, which can better inform the agent of the state it should try to achieve with its current action. Moreover, they propose a manner to utilize data from various yet related tasks, which are more accessible than task-specific ones. Specifically, they first train a flow for each task $i$ with corresponding demonstrations. The generator is defined as $a = G_i(z|u) = \exp\{c_i(u)\} \odot z + d_i(u)$ where $c_i$ and $d_i$ are trainable deep nets, $\odot$ refers to the Hadamard product. This structure design follows Real NVP (Dinh et al. (2017)). Subsequently, they train a combination flow on demonstrations of the target task, with the generator defined as $G(z|u) = (\sum_{i=1}^n \mu_i(u) \exp\{c_i(u)\}) \odot z + (\sum_{i=1}^n \lambda_i(u)d_i(u))$. At this stage, they fix $c_i$ and $d_i$ ($i = 1, \cdots, n$) and train the weighting functions $\mu_i$ and $\lambda_i$ to optimize the combination flow for the designated task via supervised learning (i.e., Eq. (65)). In this way, they efficiently utilize demonstrations from related tasks. While, **SAFER** emphasizes the safety aspect of the learned transformation. Specifically, the transformation should prevent selecting unsafe actions $a_{\text{unsafe}}$ in the environment, which are labeled in the training dataset. To achieve this, the authors propose conditioning the flow on both the state $s$ and a safety context $c$. It's important to note that $c$ is not provided directly but must be inferred from the information available in the environment. Thus, SAFER simultaneously learns (1) an inference network $P_\phi(c|\Lambda)$ to determine the current safety context $c$ based on a sliding window of states $\Lambda$ and (2) a generator $a = G_\theta(z|s,c)$ (Real NVP) conditioned on $s$ and $c$, with the following objective:

$$\max_{\theta,\phi} \mathbb{E}_{s,a,a_{\text{unsafe}} \sim D_\mu,\ c \sim P_\phi(\cdot|\Lambda)} \left[\log P_\theta(a|s,c) - \lambda \log P_\theta(a_{\text{unsafe}}|s,c)\right] - D_{KL}(P_\phi(c|\Lambda)||P_C(c)) \qquad (75)$$

Here, $P_\theta(a|s,c) = P_Z(G_\theta^{-1}(a|s,c))$, $P_C(c)$ denotes the assumed prior distribution of $c$. Intuitively, the first two terms encourage safe actions while deterring unsafe ones, and, in conjunction with the final term, the variable $c$ is compelled to encapsulate useful information regarding safety. This algorithm serves as an illustration of how NFs and VAEs can be integrated. In this setup, $P_\phi$ and $P_\theta$ function as the encoder and decoder of the VAE, respectively, with $P_\theta$ being implemented as a Normalizing Flow.

Unlike SAC-NF, CEIP, and SAFER, **NF Shaping** (Wu et al. (2021)) adopts NFs as density estimators. This is a method that combines (online) reinforcement learning and imitation learning by shaping the reward function (for RL) with a potential term learned from demonstrations. To be specific, they first train an NF (specifically MAF Papamakarios et al. (2017)) to estimate the density of state-action pairs in the demonstration dataset, i.e., $P_G(s,a) = P_Z(G^{-1}(a|s))$. Then, they shape the reward function as $\widetilde{r}_t = r(s_t, a_t, s_{t+1}) + \gamma\Phi(s_{t+1}, a_{t+1}) - \Phi(s_t, a_t)$, where $\Phi(s,a) = \beta\log(P_G(s,a) + \epsilon)$, $\beta > 0$ adjusts the weight, and $\epsilon > 0$ is a small constant to prevent numerical issues. According to (Ng et al. (1999); Wiewiora (2003)), reward shaping can intensify the reward and greatly improve the learning efficiency. Intuitively, this potential

term highlights regions where the demonstrated policy visits more frequently, signaling areas that should be explored more.

To sum up, we note that the algorithms proposed in (Akimov et al. (2022); Yang et al. (2023d); Yan et al. (2022); Slack et al. (2022)) adopt similar ideas. They first learn an NF-based transformation from the latent space to the real action space based on the offline data, and then train a high-level policy over the latent variables with online or offline RL. The prelearned transformation provides a mapping from a simpler, more controllable distribution to the target distribution, which is usually complex and unknown. Also, we find that Normalizing Flows are primarily used as exact density estimators in IL (Section 5.2), while mainly functioning as expressive generators in (Offline) RL (Section 5.3). From Table 6 and 7, we can observe that most works adopt autoregressive or coupling flows, which is probably due to their computation efficiency. As introduced in Section 5.1, continuous flows can potentially be more expressive in modelling complex data distribution. Integrating these types of flows with challenging offline policy learning tasks, such as vision-based ones, could be a future direction. On the other hand, while almost all algorithms are evaluated on robotic benchmarks such as MuJoCo or D4RL, there is a notable absence of comparative analysis among these algorithms. Given the diverse perspectives from which these algorithms are developed, it is challenging to compare them at the algorithmic level, thus comprehensive and fair empirical evaluations are essential to determine their effectiveness and suitability for practical applications.

# 6 Transformers in Offline Policy Learning

Transformers can be used for offline RL (Section 6.2) or IL (Section 6.3) by modeling them as a problem of predicting the next token based on historical data. Notably, most works in this section follow a similar algorithm design with Decision Transformer – a seminal work of trajectory-optimization-based offline RL (Section 2.1.3). Due to their architectural design, transformers are particularly well-suited and effective for modeling time-series data such as decision trajectories, as introduced in Section 6.1.

## 6.1 Background on Transformers

Transformer (Vaswani et al. (2017)) is an important foundation model that has shown exceptional capability across various areas, such as natural language processing (Kalyan et al. (2021)), computer vision (Han et al. (2022)), time series analysis (Wen et al. (2023)), and so on. Recently, there has been a growing trend in developing IL and offline RL algorithms based on transformers, with the hope to achieve sequential decision-making based on next-token predictions as in natural language processing. As shown in Figure 1, the transformer follows the widely-adopted encoder-decoder structure. The encoder maps the input $(x_1, \cdots, x_n)$ to a sequence of embeddings $(z_1, \cdots, z_n)$, and the decoder generates the output sequence autoregressively. That is, the decoder predicts one token $y_{m+1}$ at a time, based on the input embeddings $(z_1, \cdots, z_n)$ and previously-generated outputs $(y_1, \cdots, y_m)$. Both the encoder and decoder are composed of a stack of $L$ identical modules. For clarity, we present each layer in the module sequentially, following the data flow.

- **Positional Encoding:** After the token embedding layer, which is shared by $(x_1, \cdots, x_n)$ and $(y_1, \cdots, y_m)$, each element is converted to a $d$-dim vector. To enable the model to make use of the order of tokens, information about the relative or absolute positions of tokens within the sequence is injected through a positional encoding. For the $i$-th token, its positional encoding is also a $d$-dim vector, of which the $j$-th dimension is $\sin(\frac{i}{10000^{j/d}})$ if $j$ is even and $\cos(\frac{i}{10000^{(j-1)/d}})$ otherwise. The positional encoding and token embedding are then combined through summation. Per Vaswani et al. (2017), this (periodic-function-based) positional encoding design embeds relative position information and enables the model to manage sequences longer than those experienced in training.

- **Self Multi-Head Attention (MHA):** This component utilizes the attention mechanism. Given $s$ queries and $t$ key-value pairs, the attention function maps each query to a weighted-sum of the values, where the weight assigned to each value is computed as the compatibility of the query with the key paired with that value. The queries, keys, and values can be represented as $Q \in \mathbb{R}^{s \times d_q}$, $K \in \mathbb{R}^{t \times d_k}$, and $V \in \mathbb{R}^{t \times d_v}$, respectively, and the outputs for all queries (i.e., $O$) can be computed

in parallel as: (For conformability of matrix multiplication, $d_k = d_q$.)

$$O = \text{Attention}(Q, K, V) = \text{SoftMax}\left(\frac{QK^T}{\sqrt{d_k}}\right)V, \; O_i = \text{SoftMax}\left(\frac{\langle Q_i, K_1\rangle}{\sqrt{d_k}}, \cdots, \frac{\langle Q_i, K_t\rangle}{\sqrt{d_k}}\right)V \quad (76)$$

Here, SoftMax is a row operator, and the output for query $i$, i.e., $O_i$, is a weighted sum of the rows of $V$, where the weight for row $j$ is propotional to the similarity of $Q_i$ (i.e., the $i$-th row of $Q$) and $K_j$ measured by their inner product $\langle Q_i, K_j\rangle$. The dot products grow large with $d_k$, which would push the SoftMax function to regions with vanished gradients, so the factor $1/\sqrt{d_k}$ is introduced. The input embeddings $H_x$ (in Figure 1) can potentially be used as the matrices $Q$, $K$, and $V$ for self attention. However, to enable the model to jointly attend to information from different representation subspaces, a multi-head attention mechanism is adopted:

$$\text{MHA}(Q, K, V) = \text{Concat}(\text{head}_1, \cdots, \text{head}_h)W^O, \; \text{head}_i = \text{Attention}(QW_i^Q, KW_i^K, VW_i^V) \quad (77)$$

where $Q = K = V = H_x \in \mathbb{R}^{n \times d}$, $W_i^Q \in \mathbb{R}^{d \times d'}$, $W_i^K \in \mathbb{R}^{d \times d'}$, $W_i^V \in \mathbb{R}^{d \times d'}$, $W_i^O \in \mathbb{R}^{d \times d}$, $d' = d/h$, and Concat represents the concatenation operation. $W_i^Q, W_i^K, W_i^V$ convert $H_x$ to matrices in the $i$-th attention head, providing a distinct representation subspace. It's worthy noting that the output of MHA belongs to $\mathbb{R}^{n \times d}$, that is, the same in shape as its input.

- **Add & Normalize:** A residual connection (He et al. (2016)) is employed around each attention layer, followed by a layer normalization (Ba et al. (2016)). As common practice, these two operations are adopted to stabilize training of (very) deep networks (by alleviating ill-posed gradients and model degeneration). Suppose the previous layer is $F$, which can be an MHA or Feed-forward Network as shown in Figure 1, and its input is $X$, then the calculation of this Add & Normalize layer can be denoted as $\text{LayerNorm}(F(X) + X)$.

- **Point-wise Feed-forward Network (FFN):** FFN layers are important for a transformer to achieve good performance. Dong et al. (2021) observe that simply stacking MHA modules causes a rank collapse problem (i.e., leading to token-uniformity), and FFN is an important building block to mitigate this issue. Specifically, a two-layer fully-connected network with a ReLU activation function in the middle is applied to each of the $n$ token embeddings separately, leading to $n$ new $d$-dim embeddings as output.

- **Masked Self MHA & Cross MHA:** For the decoder, $H_x$ is replaced by $H_y \in \mathbb{R}^{m \times d}$, i.e., embeddings of previously-generated outputs. For rationality, the query (of the Masked Self MHA) at each position is only allowed to attend with positions up to and including that position, as the other queries correspond to outputs yet to generate. This is realized within the Masked Self MHA by masking out corresponding compatibility values, i.e., $\langle Q_i, K_j\rangle = -\infty, \; \forall j > i$ (in Eq. (76)). As for the Cross MHA, its queries come from the previous decoder layer and key-value pairs are from the output of the encoder, i.e., $H_z \in \mathbb{R}^{n \times d}$ in Figure 1. In this way, every query for predicting the next token can attend over all positions in the input sequence (via the Cross MHA) and all previously-generated tokens (via the Masked Self MHA).

Each encoder module (listed above) outputs an $n \times d$ matrix, and each decoder module outputs an $m \times d$ matrix. Here, $n$ and $m$ denote the counts of elements in the input and output sequences, respectively, while $d$ represents the embedding dimension. This uniformity in matrix sizes enables the stacking of multiple encoder (or decoder) modules to create a deep model. Each encoder (or decoder) module will take the output from the previous encoder (or decoder) module. Generally, the transformer architecture can be used in three different ways: **encoder-only**, **decoder-only**, or **encoder-decoder**. When using only the encoder, its output serves as a representation of the input sequence, suitable for natural language understanding tasks such as text classification. While, if only the decoder is employed, the Cross MHA modules are removed and this architecture is suitable for sequence generation tasks like language modeling. Applications of transformers in IL or offline RL usually adopt this decoder-only way. The overall encoder–decoder architecture is equipped with the ability to perform both natural language understanding and generation, typically used in sequence-to-sequence modeling (e.g., neural machine translation).

As a foundation model, the transformer architecture has significant advantages. Compared to **fully-connected layers**, the transformer is more flexible in handling variable-length inputs (i.e., with different $n$) and more parameter-efficient, since the amount of network parameters in a transformer is irrelevant to the sequence length $n$ [16]. **The convolutional layer** uses a convolution kernel of size $k < n$ to connect pairs of tokens, so, to link distant token pairs, a stack of convolutional layers is required. However, with the attention mechanism, each pair of tokens can be connected in one MHA layer and the time complexity for matching their corresponding query and key is constant (i.e., irrelevant to $n$). This makes the transformer excel at capturing long-range dependencies within a sequence. **Recurrent neural networks** produce a sequence of hidden states $h_i$, each of which is dependent on the previous hidden state $h_{i-1}$ and the input at position $i$. This sequential nature makes parallel computation impossible. However, in transformers, computations at each position are independent, allowing for parallel execution that are essential for processing long sequences. Last, both convolutional and recurrent models make structural assumptions over the inputs, suitable for image-like and time

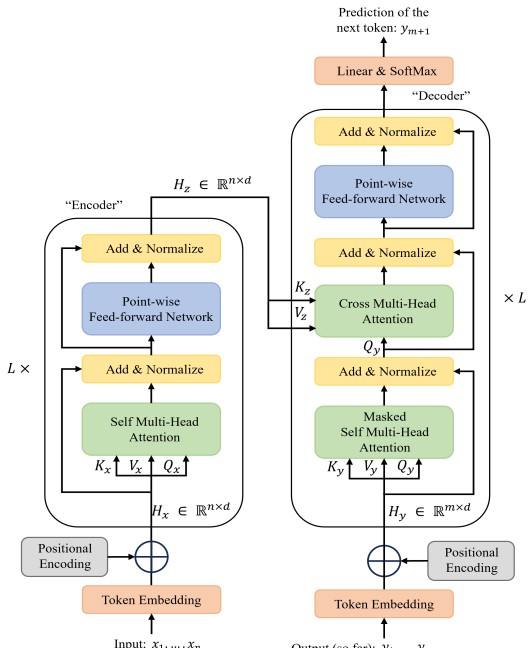

Figure 1: The Transformer Architecture

series data, respectively. However, the transformer has few prior assumptions on the data structure and thus is a more universal model. Empirical results have shown that the transformer has superior performance on a wide-range of tasks and a larger capacity to handle a huge amount of training data.

As listed above, the transformer contains multiple types of modules. Numerous studies have explored modifications or replacements of these modules in the standard transformer architecture for improvements. For a comprehensive review, please refer to (Lin et al. (2022c)). On the other hand, the computation and memory complexity of MHA modules are quadratic to the length of the input sequence (i.e., $n$), leading to inefficiency at processing extremely long sequences. Various extensions have been developed to enhance either the computational or memory efficiency. Additionally, due to the minimal structural assumptions made by the transformer about the input data, transformers are prone to overfitting when trained on small-scale datasets. Attempts like pretraining the transformer on large-scale unlabeled data (Brown et al. (2020)), introducing sparsity assumptions (Child et al. (2019)) such as limiting the number and positions of key-value pairs that each query can attend to, have been made to solve these issues. Finally, extensions to adapt the transformer for particular downstream applications are possible. Studies detailed in Section 6.2 and 6.3 can be viewed as examples of this type of extensions.

## 6.2 Transformers in Offline Reinforcement Learning

The content of this section is arranged as follows. First, we highlight pioneering studies on transformer-based offline RL in Section 6.2.1, along with a series of follow-up works for improvements in Section 6.2.2 and 6.2.3. Then, we explore the use of transformers in extended problem setups in Section 6.2.4, such as multi-agent and multi-task offline RL. In particular, for the multi-task setting, we focus on developing generalist agents based on the pretraining & fine-tuning scheme. Finally, we reflect on existing problems in this field and discuss potential future research directions in Section 6.2.5.

---

[16]The FFN layers are applied to each position rather than the whole sequence, and the MHA layers only require updating the weights: $W_i^Q, W_i^K, W_i^V, W^O$, the sizes of which are irrelevant with $n$.

### 6.2.1 Representative Works

Inspired by the great success of high-capacity sequence generation models (such as the transformer) in natural language processing (NLP), Chen et al. (2021b) propose to view offline RL as a sequence modeling problem and solve it with a **Decision Transformer** (DT). This algorithm exemplifies trajectory-optimization-based offline RL (introduced in Section 2.1.3), so we provide a detailed introduction of it, along with two other seminal works, as a brief tutorial on this particular branch of offline RL methods. Given offline trajectories $\{\tau = (s_0, a_0, R_0, \cdots, s_T, a_T, R_T)\}$, where $R_t = \sum_{i=t}^{T} r_i$ denotes the return-to-go (RTG), the DT is trained to predict the next action based on the previous $k+1$ transitions:

$$\min_{\pi} \mathbb{E}_{\tau} \left[ \sum_{t=0}^{T} (a_t - \pi(\tau_{t-k:t}))^2 \right], \text{ or } \min_{\pi} \mathbb{E}_{\tau} \left[ \sum_{t=0}^{T} -\log \pi(a_t | \tau_{t-k:t}) \right] \tag{78}$$

where $\tau_{t-k:t} = (s_j, a_j, R_j, \cdots, s_t, R_t)$ ($j = \min(t-k, 0)$) is the input sequence; the two terms above are for tasks with continuous and discrete actions, respectively. DT adopts a GPT-like architecture (Radford et al. (2018)), which is a decoder-only transformer as introduced in Section 6.1. During evaluation rollouts, an initial target return $R_0$ must be specified, which can be the highest achievable return for the task. The inference trajectory is then generated autoregressively using the DT as follows: $s_0 \sim \rho_0(\cdot)$, $a_0 \sim \pi(\cdot | s_0, R_0)$, $s_1 \sim \mathcal{T}(\cdot | s_0, a_0)$, $r_0 = r(s_0, a_0)$, $\cdots$, $R_t = R_{t-1} - r_{t-1}$, $a_t \sim \pi(\cdot | \tau_{t-k:t})$, $\cdots$, where $(\rho_0, \mathcal{T}, r)$ are components of the MDP. DT greatly simplifies offline RL by eliminating the necessity to fit value functions through dynamic programming or compute policy gradients as in classical offline RL. Using such a (supervised-learning based) sequence modeling objective makes it less prone to selecting OOD actions, because multiple state and action anchors throughout the trajectory prevent the learned policy from deviating too far from the behavior policy $\mu(a|s)$. Additionally, as a high-capacity model, the transformer has the potential to enhance both the scalability and robustness of the learned policy, similar to its applications in NLP. Per Chen et al. (2021b), DT shows competitive performance compared with prior offline RL methods, such as Conservative Q-Learning (CQL, Kumar et al. (2020)), especially for tasks with sparse and delayed reward functions. However, Emmons et al. (2022) investigate what is essential for offline RL via supervised learning (RvS) through extensive experiments. Their findings indicate that a two-layer MLP model, utilizing a simpler input format (i.e., $(s_t, R_t)$ instead of $\tau_{t-k:t}$) and the same objective (i.e., Eq. (78)), exhibits competitive, and in some cases superior, performance compared to DT, and DT underperforms prior offline RL algorithms on most benchmark tasks. This raises the questions whether it is necessary to adopt such a high-capacity model, i.e., the transformer, as the policy network, and under what scenarios RvS methods might outperform dynamic-programming-based offline RL methods.

**Trajectory Transformer** (TT, Janner et al. (2021)) follows DT but utilizes more techniques from NLP, including tokenization, discretization, and beam search. In particular, they treat each dimension of the state and action as a token and discretize them independently. Suppose the state and action have $N$ and $M$ dimensions respectively, the objective is $\min_{\pi} \mathbb{E}_{\tau} [\mathcal{L}(\tau)]$, where $\mathcal{L}(\tau)$ is defined as follows:

$$\sum_{t=0}^{T} \left[ \sum_{i=1}^{N} \log \pi(s_t^i | s_t^{<i}, \tau_{<t}) + \sum_{j=1}^{M} \log \pi(a_t^j | a_t^{<j}, s_t, \tau_{<t}) + \log \pi(r_t | a_t, s_t, \tau_{<t}) + \log \pi(R_t | r_t, a_t, s_t, \tau_{<t}) \right] \tag{79}$$

This objective involves supervision on each dimension of the state/action, the reward, and the return-to-go. For the inference process, a beam search technique (Freitag & Al-Onaizan (2017)) is utilized. Specifically, $B$ most-likely samples are kept when sampling $a_t^j$ and $R_t$. Then, samples $(a_t^{1:M}, R_t)$ with the highest cumulative reward plus return-to-go (i.e, the estimated trajectory return $\sum_{i=0}^{t-1} r_i + R_t$) is selected for execution. Note that, during the inference, $s_t^{1:N}$ and $r_t$ are acquired from the simulator. TT shows superior performance than DT in some benchmarks as reported in (Janner et al. (2021)), but treating each dimension separately would introduce learning and sample inefficiencies, especially for high-dimensional tasks.

Instead of modeling the policy, **Q-Transformer** (Chebotar et al. (2023)) proposes to learn a transformer-based Q-network. Similarly to TT, each dimension of the action is discretized and treated as a separate time step. When training, they update the Q-function in a temporal difference (TD) manner, i.e., minimizing the disagreement between the predicted Q-value $Q(s_{t-k:t}, a_t^{1:i})$ and target Q-value $\widehat{Q}(s_{t-k:t}, a_t^{1:i})$.

If $i = M$ (i.e., the action dimension), $\widehat{Q}(s_{t-k:t}, a_t^{1:i}) = r_t + \max_{a_{t+1}^1} Q(s_{t-k+1:t+1}, a_{t+1}^1)$; otherwise, $\widehat{Q}(s_{t-k:t}, a_t^{1:i}) = \max_{a_t^{i+1}} Q(s_{t-k:t}, a_t^{1:i}, a_t^{i+1})$. Note that, for the Bellman optimality (Puterman (2014)) of this per-dimension updating rule to hold, it requires the assumption that the choice of $a_t^{1:i}$ does not influence $\max_{a_t^{i+1}} Q(s_{t-k:t}, a_t^{1:i}, a_t^{i+1})$. However, this is probably not true since dimensions of an action are likely to be correlated. Thus, this method lacks theoretical support. During evaluation, the action is also generated dimension by dimension, based on the trained Q-function.

So far, we have introduced some fundamental algorithm designs to cast offline RL as sequence modelling problems based on transformers. Next, we present follow-up improvements of these methods. As illustrated in (Emmons et al. (2022)), **carefully aligning the model capacity with the training data to prevent overfitting or underfitting** and **choosing which information to condition the policy on** are critical for RvS performance. The discussions concerning the following works focus on these two aspects.

### 6.2.2 Balancing Model Capacity with Training Data

The performance of transformer-based offline RL relies heavily on the quantity and quality of the training data. A series of studies have been proposed focusing on augmenting the offline dataset across various task scenarios. (1) Based on TT, **Bootstrapped Transformer** (Wang et al. (2022a)) proposes to generate trajectories from $\pi$, i.e., the model being learned, and adopt them as additional training data to expand the amount and coverage of the offline dataset. The self-generated trajectories are filtered by their log likelihood under the policy $\pi$, which indicates the quality and reliability of the training data. (2) **SS-DT** (Zheng et al. (2023)) introduces a data augmentation method for semi-supervised settings, where most trajectories lack action labels and are in the format $(s_0, r_0, \cdots, s_T, r_T)$. Their approach simply involves training an inverse dynamics model, i.e., $\mathcal{T}_{\text{inv}}(a_t|s_t, s_{t+1})$, on trajectories with action labels and then applying this model to predict actions for the unlabeled trajectories. (3) Given only sub-optimal trajectories, trajectory-optimization-based offline RL would fail. In this case, the agent needs to learn to stitch segments from different trajectories for an optimal policy. Value-based RL does not have the same issue as it pools information for each state across trajectories through the Bellman backup. In this case, **QDT** (Yamagata et al. (2023)) suggests learning a Q-function via CQL and replacing the RTG values (i.e., $R_t$) in the offline dataset with the learned Q-values as augmentation for DT training. (4) **CDT** (Liu et al. (2023b)) proposes a data augmentation method for safe RL. Specifically, in a Constrained MDP (Altman (1998)), the DT conditioned on both $R(\tau)$ and $C(\tau) = \sum_{i=0}^T c_t$ (i.e., the constraint-violation cost of the trajectory) is expected to achieve the target trajectory return while ensuring that the accumulated cost remains lower than $C(\tau)$. However, during inference, unachievable $(R(\tau), C(\tau))$ pairs may be given as conditions, which are not represented in the offline dataset. Thus, as an augmentation, they suggest using the trajectories from the offline dataset that achieve the highest return (which is lower than the unachievable return $R(\tau)$) without violating $C(\tau)$ as the corresponding trajectories of $(R(\tau), C(\tau))$ for offline training. (5) For tasks with sparse and delayed rewards, DT suffers from model degradation, since the RTG does not change within a trajectory. **DTRD** (Zhu et al. (2023a)) proposes to learn a reward shaping function $r_\phi(s, a)$ to redistribute the delayed reward to each time step. This is achieved by solving a bi-level optimization problem as below:

$$\min_\phi \mathcal{L}_{\text{val}}(\theta^*(\phi), \phi) \; s.t. \; R(\tau) = \sum_{t=0}^T r_\phi(s_t, a_t), \; \forall \tau, \text{ and } \theta^*(\phi) = \arg\min_\theta \mathcal{L}_{\text{train}}(\theta, \phi) \tag{80}$$

Here, $\theta$ is the parameter of the policy $\pi$; $\mathcal{L}_{\text{train}}(\theta, \phi)$ denotes the DT objective on the training dataset, which is augmented by replacing the original sparse rewards $r_t$ with $r_\phi(s_t, a_t)$; $\mathcal{L}_{\text{val}}(\theta^*(\phi), \phi)$ is the DT objective on the augmented validation dataset. Intuitively, $r_\phi$ is updated to ensure that the policy learned from the dataset, augmented with corresponding dense rewards, is optimal. They propose a practical alternative training framework for $\theta$ and $\phi$, which however lacks convergence guarantee in theory.

Another line of works in this category focus on modifying the DT architecture to explicitly make use of the structural patterns within the training trajectories. As mentioned in Section 6.1, introducing strutual assumptions/priors to the input data can greatly improve the learning efficiency and mitigate the overfitting issue when learning from small-scale datasets. (1) Based on the observation that each trajectory is a sequence of state-action-reward triplets, **StARformer** (Shang et al. (2022)) proposes to first extract representations

$z_t$ from each $(r_{t-1}, a_{t-1}, s_t)$, and then apply DT on $(z_1, \cdots, z_t)$ to predict $a_t$. Note that they adopt $r_t$ rather than $R_t$, which is a counterintuitive design, as RL decision-making typically relies on information about long-term returns. (2) **GDT** (Hu et al. (2023b)) follows the same motivation as StARformer, recognizing that naively attending to all previous tokens, as in DT, can overlook the causal relationships among certain types of tokens and thus hurt the performance. In particular, they introduce relation embeddings to the compatibility calculation (i.e., Eq. (76)) for each pair of tokens $(x_i, x_j)$. With this modification, the compatibility between the query $Q_i$ and key $K_j$ is calculated as $\langle Q_i + re_{i \to j}, K_j + re_{j \to i} \rangle$, where $re$ is an embedding of the adjacency matrix that represents the causal relationships among tokens in the input sequence. For instance, $a_t$ is directly influenced by $s_t$ and $R_t$, so there should be edges from $s_t$ to $a_t$ and from $R_t$ to $a_t$ in the casual graph. In this way, they incorporate the Markovian relationship into the DT model. (3) **DTd** (Wang et al. (2023c)) views the state, action, and RTG as distinct modalities. By analyzing attention values from a trained DT, they rank the importance of token interactions (for decision-making) within or cross modalities. Specifically, $s - s, a - a, R - R < s - R, a - R < s - a$, where $s - s$ denotes interactions (i.e., attending via the MHA) between states, and '$<$' denotes being less important. Consequently, they suggest a hierarchical DT structure, processing less crucial interactions before more significant ones to prevent important interactions from being distracted. The resulting structure is large in scale and outperforms DT, but its overall performance is close to traditional offline RL methods like CQL.

### 6.2.3 Mitigating Impacts from Environmental Stochasticity

Another category of research works that follow DT propose to change the conditional information of the DT to enhance its performance in stochastic environments. (1) In the absence of determinism, a high-return offline trajectory could be the outcome of uncontrollable environmental randomness, rather than the result of the agent's actions. Thus, goals that are independent from the environmental stochasticity are the only conditions that the agent can reliably achieve. **ESPER** (Paster et al. (2022)) and **DOC** (Yang et al. (2023c)) propose methods to learn such conditions automatically from the offline dataset. They share a common intuition: to ensure the conditional variable to contain sufficient information on action predictions but no information regarding the environment dynamics (e.g., on predicting future rewards or state transitions). Both algorithms use a contrastive learning framework, resulting in similar objectives. However, these methods require to learn at least five networks (so we choose not to show their objectives here), which lose the simplicity of DT and is even more complicated than traditional offline RL approaches. (2) **SPLT** (Villaflor et al. (2022)) proposes to model the policy as $\pi(\hat{a}_t | (s_{t-k}, a_{t-k}, \cdots, s_t), z_t^\pi)$, where the condition $z_t^\pi$ is sampled from an encoder that takes $(s_{t-k}, a_{t-k}, \cdots, s_t, a_t)$ as input. Both the encoder and $\pi$ have DT-like architectures and are trained jointly in a CVAE framework (introduced in Section 3.1), where $\pi$ works as the decoder, $(s_{t-k}, a_{t-k}, \cdots, s_t)$ and $a_t$ work as the condition and data point respectively. $z_t^\pi$ is a $d_\pi$-dim discrete vector, each dimension of which has $c$ categories. Extracted by the CVAE, each specific $z_t^\pi$ can be viewed as a mode contained in the offline data and can be assigned to $\pi$ as a generation condition. In the same manner, they learn a environment model $\omega(\hat{r}_t, \hat{s}_{t+1}, \widehat{R}_{t+1} | (s_{t-k}, a_{t-k}, \cdots, s_t, a_t), z_t^\omega)$. During inference, with $(\pi, \omega)$ and by enumerating $(z_t^\pi, z_t^\omega)$, $c^{d_\pi \times d_\omega}$ predictions on future $h$-length trajectories $\hat{\tau}_t^h$ can be obtained, and the next action is selected from the mode: $\arg\max_{z_t^\pi} \min_{z_t^\omega} \widehat{R}(\hat{\tau}_t^h)$ for robustness, where $\widehat{R}(\hat{\tau}_t^h) = \sum_{i=0}^{h-1} \hat{r}_{t+i} + \widehat{R}_{t+h}$. Through thorough enumeration on the future predictions, the influence brought by the environmental randomness can be mitigated. (3) As in QDT (introduced in Section 6.2.2), **CGDT** (Wang et al. (2023d)) suggests training an additional Q-network to guide the learning of DT. The rationale behind this method is that the Q-value, representing the expected return for each state-action pair, can help mitigate the impact of environmental stochasticity that could be wrongly captured by single-trajectory RTG values.

### 6.2.4 Transformers in Extended Offline Reinforcement Learning Setups

DT has been extended to various offline RL setups, including model-based, hierarchical, multi-agent, and multi-task offline RL. Unlike the previous three subsections, extensions in this part primarily expand on the problem setup rather than improving the algorithm design with respect to DT. Thus, this part is not our primary focus and we only provide a taxonomy and brief introductions of the related works.

**Model-based Offline RL:** Wang et al. (2023b) propose **Environment Transformer** to model the transition dynamics and reward function $P(s_{t+1}, r_t | s_t, a_t)$ from the offline dataset, which can be viewed as simply replacing the output of DT, i.e., $a_t$, with $(s_{t+1}, r_t)$. The learned environment model can then be utilized as a simulator for policy learning with any RL algorithm. **TransDreamer** (Chen et al. (2022)) suggests replacing the RNN module within Dreamer (Hafner et al. (2020)), which is a state-of-the-art (SOTA) model-based RL algorithm, with a transformer, for improved performance on tasks requiring modeling long-term temporal dependency. This reflects the advantage of transformers over RNNs, as mentioned in Section 6.1.

**Hierarchical Offline RL: HDT** (Correia & Alexandre (2022)) employs two DTs for hierarchical decision making. The high-level DT models the distribution $\pi_{\text{high}}(g_t | s_{t-k}, g_{t-k}, \cdots, s_t)$ for selecting a subgoal, while the low-level DT captures the corresponding action distribution $\pi_{\text{low}}(a_t | s_{t-k}, g_{t-k}, a_{t-k}, \cdots, s_t, g_t)$. Moreover, they propose a heuristic approach for extracting subgoals $g_{1:T}$ from the offline trajectories, when these subgoals are not labeled. **Skill DT** (Sudhakaran & Risi (2023)) adopts a similar hierarchical framework, but proposes to use a VQ-VAE (introduced in Section 3.1) as the high-level policy to decide on the skill choices for the low-level policy to condition on.

**Multi-agent Offline RL: MADT+Distillation** (Tseng et al. (2022)) is proposed to cast offline MARL as a sequence modelling problem. First, a teacher (joint) policy $\pi(a_t^{1:n} | o_{t-k}^{1:n}, R_{t-k}^{1:n}, a_{t-k}^{1:n}, \cdots, o_t^{1:n}, R_t^{1:n})$ is learned. Its training process is the same as DT, viewing the concatenation of observations (RTGs/actions) from $n$ agents as a single joint observation (RTG/action). Then, $\pi$ is distilled to $n$ student (individual) policies $\pi^i(a_t^i | o_{t-k}^i, R_{t-k}^i, a_{t-k}^i, \cdots, o_t^i, R_t^i)$, to enable decentralized execution. The policy distillation process is designed for $\pi^{1:n}$ to imitate $\pi$ while maintaining the structural relation among these agents as in the joint policy $\pi$, which does not rely on transformers. There are some other explorations for the multi-agent scenario: **MADT** (Meng et al. (2021)) investigates combining offline pretraining using DT and online adaptation with gradient-based MARL algorithms; **SCT** (Li et al. (2023b)) proposes approaches for handling non-cooperative MARL scenarios. However, these two works are still in early stages of development.

**Multi-task Offline RL:** Transformers have demonstrated exceptional generalization capabilities in NLP and CV tasks. Thus, there is a branch of research works on enhancing DT to enable policies pretrained on source tasks to be quickly adapted to target tasks through fine-tuning. The target task may be the same as the source task, differ but still fall within the same task distribution with the source task, or belong to entirely different modalities from the source task. We provide a brief review of these three categories as follows, each progressively demanding greater generalization capabilities. **(1)** Further adaption on the same task is required when the provided offline data is highly sub-optimal or only covers a limited part of the state space, for which **ODT** (Zheng et al. (2022)) introduces a technique to fine-tune the pretrained DT through further online interactions with the environment. Another possible scenario is when the high-quality and labelled offline data is scarce while there are abundant unlabeled (i.e., reward-free) trajectories. In this case, **PDT** (independently proposed by Cang et al. (2022) and Xie et al. (2023)) can be used for unsupervised pretraining on the unlabeled data, followed by fine-tuning using the limited high-quality data. **(2)** Adaptation to unseen tasks within the same task distribution is a primary focus in multi-task/meta (offline) RL. To extend DT in this realm, some studies suggest conditioning the DT on task-specific information $z \sim P_Z(\cdot)$ and training the DT across a range of tasks, each associated with unique embeddings $z_{\text{train}} \sim P_Z(\cdot)$. Consequently, when encountering new tasks sampled from $P_Z(\cdot)$ and conditioning on the corresponding $z_{\text{test}}$, the policy is expected to be effective in these previously unseen tasks after few/zero-shot fine-tuning. $z$ can be a task id (Lin et al. (2022a)), segment of a task trajectory (Xu et al. (2022b)), or trajectory embedding from an encoder (Lin et al. (2022b)). Alternatively, Boustati et al. (2021) suggest running the source policy in similar but counterfactual environments to gather augmented data for DT training, to enhance the robustness of the policy in unseen tasks. **(3)** So far, the transformer has been successfully adopted in NLP, CV, and RL, which makes it possible to build a transformer-based generalist agent across these three modalities. **Gato** (Reed et al. (2022)) shows that a single transformer with the same set of weights can handle 604 distinct tasks with varying modalities. Considering that texts, images, and decision trajectories can all be tokenized and embedded, the transformer can learn to generate all modalities within a unified training framework – predicting the next token based on previously generated ones. Specifically, the training objective aligns with Eq. (78), but substitutes the training data with sequences of words for text generation and sequences of image patches for image understanding. During training, each batch of training data incorporates sequences

from various modalities and each sequence is concatenated with a demonstration for the same task, serving as a prompt to assist the agent in differentiating between modalities. Gato utilizes a decoder-only transformer with 1.2 billion parameters and is trained on a dataset containing 1.76 trillion tokens. The empirical results show that Gato achieves performance exceeding 50% of the expert score threshold in 450 of the 604 evaluation tasks. Gato represents a significant attempt in developing generalist agents, yet its efficacy heavily depends on the quantity and quality of the training data. Especially, when learning control policies, it doesn't utilize reward signals, so, through imitation only, its performance would be capped by the provided demonstrations. **Multi-Game DT** (Lee et al. (2022)) adopts a similar protocol with Gato and suggests how to utilize reward signals in both the training and inference process. Another critical question regrading multi-modal learning is whether the learning in distinct modalities can enhance each other. Reid et al. (2022) empirically show that pretrained language models can be effectively adapted to offline RL tasks with boosted performance. Following this work, Takagi (2022) provides an excellent in-depth empirical study on the effect of cross-modality pretraining for DT. They find that, after fine-tuning, parameters of the language-pretrained model do not change as much as a randomly-initialized model, and these stable parameters preserves context-equivalent information such that the agent can make decisions even in the absence of contexts (i.e., without using the previous $k$ transitions as input). Further, the way to efficiently utilize these context-equivalent information is also preserved by those stable parameters. As a result, they claim that pretrained context-like information could help the DT training by enabling the model to predict actions more accurately. In contrast, a transformer pretrained on images performs significantly worse than a standard DT, partly due to the significant differences in data characteristics.

### 6.2.5 Reflections and Future Works on Transformer-based Offline Reinforcement Learning

We notice that there is a series of works on applying transformers to **online** partially-observable RL (Yuan et al. (2023a)). They adopt the transformer as the policy network in gradient-based or actor-critic RL algorithms, with the hope to harness its ability to process long-horizon historical information for decision making. These works focus on architectural modifications to the standard transformer, which are specifically tailored for RL. For example, **GTrXL** (Parisotto et al. (2020)) and **Catformer** (Davis et al. (2021)) suggest adjustments to the MHA module to facilitate a more stable RL training process; **ALD** (Parisotto & Salakhutdinov (2021)) adopts model distillation to improve the computation efficiency in transformer-based distributed RL; **STT** (Yang et al. (2022e)) and **WMG** (Loynd et al. (2020)) propose adaptions for the transformer to process spatiotemporal coupling observations and factored observations, respectively, for enhanced sample efficiency. These architectural modifications, or potentially new ones, could be integrated with policy gradient offline RL algorithms or DT-like return-conditioned supervised learning (RCSL) algorithms for performance improvement. Conducting an (empirical) comparison between these two categories (i.e., RCSL and policy gradient offline RL), both utilizing the same transformer architecture, would also be an intriguing study.

Finally, we reflect on these DT-like RCSL methods. Although a series of improvements have been made, there are still fundamental limitations regarding these algorithms. According to (Brandfonbrener et al. (2022)), RCSL returns near-optimal policy under a set of assumptions that are stronger than those needed for traditional RL algorithms, and RCSL alone is unlikely to be a general solution for offline RL problems. In particular, RCSL offers guarantees only when the environment dynamics (including the state transition and reward functions) are nearly deterministic, a priori knowledge of the optimal conditioning function (i.e., the RTG values) is available, and the return distribution of the provided offline trajectories can cover the possible values of the conditioning function. These findings inspire several directions for future research. First, adapting RCSL algorithms to stochastic environments, ideally backed by theoretical optimality guarantees, is a clear necessity. Second, developing effective strategies for selecting RTG values as the conditions during inference, or alternatively, exploring new conditioning functions, is crucial. Last, a significant challenge lies in addressing out-of-distribution scenarios in case that the offline dataset lacks adequate coverage of the task scenarios. Another fundamental problem regarding RCSL is whether it is necessary to learn low-return behaviors from the offline dataset and to require strong alignment between the target return (i.e., the condition of the DT) and realized return. As indicated in Eq. (78), the policy training involves imitating trajectories across a range of returns. However, only high-return policy is required during inference. This

| Algorithm | Key Novelty | Evaluation Task |
|---|---|---|
| DT | Solving offline RL as sequence modeling | Atari, D4RL (L) |
| TT | Tokenization, discretization and beam search | D4RL (L, M) |
| Q-Transformer | Transformer-based Q learning | Real Robot |
| Boot Transformer | Data augmentation with self-generated data | D4RL (L, A) |
| SS-DT | Data augmentation via an inverse dynamic model | D4RL (L, M2d) |
| QDT | Replacing RTGs with Q-values to enable stitching | D4RL (L, M2d) |
| CDT | Data augmentation for safe RL in constrained MDP | Bullet-safety-gym |
| DTRD | Data augmentation for sparse, delayed reward setups | Atari, MiniGrid, D4RL (L, K, M2d) |
| StARformer | Processing $(r_{t-1}, a_{t-1}, s_t)$ as a group | Atari, dm_control |
| GDT | Embedding the causal relation among $s, a, R$ | Atari, D4RL (L) |
| DTd | Processing $s, a, R$ as three distinct modalities | D4RL (L, M) |
| ESPER | Replacing RTGs with a stochasticity-free conditioner | Stochastic Benchmark |
| DOC | Replacing RTGs with a stochasticity-free conditioner | MuJoCo, D4RL (M), FrozenLake |
| SPLT | Enumerating future trajs to mitigate stochasticity | CARLA |
| CGDT | Critic-guided DT training | D4RL (L, M) |
| Env Trans (MB) | Modeling the dynamic and reward functions | D4RL (L) |
| TransDreamer (MB) | Integrating Dreamer with a transformer architecture | Hidden Order Discovery, dm_control, Atari |
| HDT (HRL) | Subgoal-conditioned decision-making | D4RL (L, A, K) |
| Skill DT (HRL) | Skill extraction via a VQ-VAE | D4RL (L, M) |
| MADT+Dist (MA) | A centralized training with decentralized execution (CTDE) framework for MARL based on policy distillation | Fill-In, Equal Space, SMAC, Grid-World, Highway |
| MADT (MA) | Applying a shared DT to each agent's sequence | SMAC |
| SCT (MA) | A DT-based method for non-cooperative MARL | simple-tag, simple-world |
| ODT (PT) | An online fine-tuning method for DT | D4RL (L, M) |
| PDT (PT) | Pretraining on large-scale reward-free trajectories | D4RL (L) |
| MG DT (MT) | A single transformer agent for 46 Atari games | Atari |
| Gato (MT) | A generalist agent for 604 cross-modality tasks | See Reed et al. (2022) |

Table 8: Summary of transformer-based offline RL algorithms. In Column 1, we list representative (but not all) algorithms in this section, where Boot Transformer, Env Trans, MADT+Dist, and MG DT correspond to Bootstrapped Transformer, Environment Transformer, MADT+Distillation, and Multi-Game DT, respectively. These algorithms are grouped by their categories, with abbreviations MB, HRL, MA, PT, and MT denoting model-based, hierarchical, multi-agent, pretraining-based, and multi-task offline RL, respectively. The evaluation tasks are listed in Column 3. Most works are evaluated on D4RL (Fu et al. (2020)), which provides offline datasets for various tasks, including Locomotion (L), AntMaze (M), Adroit (A), Kitchen (K), Maze2d (M2d). Regarding the other benchmarks, we provide their references here: Atari (Bellemare et al. (2013)), Real Robot (Chebotar et al. (2023)), Bullet-safety-gym (Gronauer (2022)), MiniGrid (Chevalier-Boisvert et al. (2018)), dm_control (Tassa et al. (2018)), Stochastic Benchmark (Paster et al. (2022)), MuJoCo (Todorov et al. (2012)), FrozenLake (Foundation (2023)), CARLA (Dosovitskiy et al. (2017)), Hidden Order Discovery (Chen et al. (2022)), SMAC (Samvelyan et al. (2019)), Fill-In & Equal Space & Grid-World & Highway (Meng et al. (2021)), simple-tag & simple-world (Li et al. (2023b)).

raises the possibility of developing a more efficient mechanism that strategically utilizes low-return samples to enhance the learning of a high-return policy.

## 6.3 Transformers in Imitation Learning

Similarly with DT, transformer-based IL utilizes a supervised learning paradigm (see Section 6.3.1), where the transformer is adopted as the policy backbone (see Section 6.3.2). We differentiate works for IL and offline RL simply by if the reward or return is used as a condition of the policy. Notably, a significant advantage of the transformer is its capability to process large quantities of training data across multiple modalities, which enables the development of generalist agents, as detailed in Section 6.3.3.

### 6.3.1 A Paradigm of Transformer-based Imitation Learning

Given expert demonstrations $D_E = \{\tau_E = (s_0, a_0, \cdots, s_T, a_T)\}$, transformer-based IL algorithms are designed to learn a policy $\pi$ to replicate expert behaviors. The objective function is typically formulated as follows:

$$\min_\pi \mathbb{E}_{D_E} \left[ (a_t - \log \pi(s_t, x_{t-1} \cdots, x_{t-k}))^2 \right], \text{ or } \min_\pi \mathbb{E}_{D_E} \left[ -\log \pi(a_t | s_t, x_{t-1}, \cdots, x_{t-k}) \right] \quad (81)$$

Here, $x_t \triangleq (s_t, a_t)$ or $s_t$, and $\pi$ is implemented as a decoder-only transformer (as introduced in Section 6.1). This objective is similar in form with the one of DT (i.e., Eq. (78)) but replaces $(s_t, a_t, R_t)$ with $x_t$. IL algorithms do not require reward signals but place an emphasis on the quality of demonstrations, which ideally should come from experts, as the learning process relies solely on imitation. Also, to make full use of demonstrations, auxiliary supervision objectives, such as prediction errors on the next state (i.e., the forward model loss) or the intermediate action between two consecutive states (i.e., the inverse model loss), are often employed. These objectives complement the action prediction error (i.e., Eq. (81)) to aid the policy learning process. As a representative, **Behavior Transformer** (**BeT**, Shafiullah et al. (2022)) empirically shows that a standard transformer architecture (specifically minGPT (Brown et al. (2020))) can significantly outperform commonly-used policy networks, like MLP and LSTM, in learning from large-scale, human-generated demonstrations, which typically exhibit high variance and multiple modalities. To cover the multiple modes in expert behaviors, the authors suggest dividing the action $a$ into two components: its corresponding action center $\lfloor a \rfloor$ and the residual action $\langle a \rangle$, such that $a = \lfloor a \rfloor + \langle a \rangle$. The set of action centers can be acquired by applying K-means clustering to the expert actions. Correspondingly, they apply a policy network $\pi$ with two prediction heads, one for the action center and the other for the residual action. Explicitly employing clustering to identify the various modes in expert actions and utilizing dual-head predictions to reason the mode of each instance significantly aids in modelling multi-modal actions. However, this approach does not model the multi-modality that may exist in the joint distribution of $(s, a)$.

### 6.3.2 Policy Approximation using Transformers

A series of studies (Kim et al. (2021a); Pan et al. (2022); Kim et al. (2022); Zhu et al. (2022); Chen et al. (2023a); Kim et al. (2023); Liang et al. (2023a)) have adopted transformers as the policy backbone for imitation learning in complex control tasks, such as vision-based robotic manipulation and end-to-end self-driving. As introduced in Section 6.1, the transformer has various advantages over CNNs and RNNs. Firstly, the transformer is adept at processing time-series data (Kim et al. (2022)) by capturing the long-term temporal dependencies between inputs from different time steps. By using the current state as the query and historical data as keys and values, the transformer-based agent can pinpoint vital information in the history for the current decision-making through the attention mechanism. Secondly, the transformer can concurrently process various types of input data, such as texts and images (Kamath et al. (2023)), texts and voxels (Shridhar et al. (2022)), point clouds (Pan et al. (2022)), enabling it to be the backbone of an end-to-end deep learning agent. Each type of data can be embedded to vectors by (pretrained) domain-specific neural networks. Then, instead of simply concatenating all these embeddings for subsequent processing, the transformer treats each embedding as distinct tokens. As detailed in Section 6.1, each token possesses its own query, key, and value, and can attend to the other tokens for more informative aggregation. Thirdly, the transformer is efficient in processing input that contains multiple entities by treating each entity's

representation as a token and modeling their interrelations through the attention mechanism. For example, **TSE** (Liang et al. (2023a)) takes the state of the ego vehicle, the status of surrounding vehicles, and lane information as input, explicitly reasoning the relationship between the ego vehicle and its surrounding entities for self-driving decision-making; **Silver-Bullet-3D** (Pan et al. (2022)) models the relationship of embeddings corresponding to different parts of the manipulated objects and robotic arms through the transformer for complex manipulation tasks; **VIOLA** (Zhu et al. (2022)) extracts a series of object-centric representations from the visual observation, introduces an extra action token (whose corresponding output is the action prediction), and applies a transformer on top of these tokens, so that the action token can learn to attend to and focus on task-relevant objects for improved performance in action prediction.

### 6.3.3 Transformer-based Generalist Imitation Learning Agents

Another research focus in this field is language-conditioned IL, aiming at training (robot) agents to follow human instructions. A language instruction, comprising a sequence of words, can be embedded into a sequence of tokens $(l_1, \cdots, l_m)$ through a pretrained language model or a predefined embedding table. **(1)** A straightforward method for language-conditioned IL, as shown in **TDT** (Putterman et al. (2022)), is to incorporate language tokens into the policy input, i.e., $\pi(a_t|s_t, x_{t-1}, \cdots, x_{t-k}, l_1, \cdots, l_m)$. Still, $\pi$ is implemented as a decoder-only transformer, which, as previously mentioned, is capable of processing multiple types of input concurrently. **Perceiver-Actor** (Shridhar et al. (2022)) adopts a similar design, but suggests using the Perceiver Transformer (Jaegle et al. (2022)) as the policy backbone to manage extra long sequences of input (e.g., a sequence of image/voxel patches from the vision input and word tokens from the language input). **(2) MARVAL** (Kamath et al. (2023)) uses an encoder-only transformer to process inputs comprising four modalities: instruction texts $(l_1, \cdots, l_m)$, historical states and actions $(x_{t-k}, \cdots, x_{t-1})$, the current state $s_t$, and the current action candidates $\mathcal{A}_t$. As discussed in Section 6.1, without the Masked MHA component as in the transformer decoder, each input token can attend to every other token for more informative aggregation. However, each forward pass predicts only a single action $a_t$, making it less efficient in training than the decoder-only transformer. In decoder-only transformers, the output token corresponding to the state input $s_i$ is used to predict $a_i$ $(i = 0, \cdots, t)$. In contrast, encoder-only transformers require a special action token, i.e., CLS, as an input to attend to all other tokens and capture the fused representation of the entire sequence, and its corresponding output token is utilized to predict the current action $a_t$. Additionally, MARVAL introduces auxiliary tasks, including predicting the masked language tokens and predicting the proportion of the trajectory that has been completed, to increase the amount of supervision for more efficient use of the demonstrations. **(3)** Instead of using instruction texts as conditions, **Lang** (Hejna et al. (2023)) proposes to predict corresponding instructions from the state sequence as an auxiliary task, to realize language-guided IL. Specifically, each demonstration trajectory is divided into $m$ segments, with each segment $(s_{T_i}, \cdots, s_{T_{i+1}-1})$ associated with a subtask instruction $l^{(i)} = (l_1^{(i)}, \cdots, l_{b_i}^{(i)})$. A transformer encoder is applied to extract representations $(z_1, \cdots, z_t)$ from the state sequence $(s_1, \cdots, s_t)$. These representations are expected to encompass information essential for predicting both actions and instructions, achieved by minimizing the following equation: $-\sum_{t=1}^{T} \log \pi(a_t|z_1, \cdots, z_t) - \lambda \sum_{i=1}^{m} \sum_{j=1}^{b_i} \log P(l_j^{(i)}|l_1^{(i)}, \cdots, l_{j-1}^{(i)}, z_1, \cdots, z_{T_i-1})$. This approach offers an alternative way to incorporate language instructions into action predictions and has proven effective when training with a limited amount of demonstrations.

Language-conditioned IL can be viewed as an instance of multi-task IL, where language texts serve as task contexts. Next, we introduce three approaches to multi-task IL, including context-conditioned IL, pretraining & fine-tuning, and generalist agent development. **(1)** Generally speaking, $(l_1, \cdots, l_m)$ could be substituted with any form of task context (Furuta et al. (2022)). For example, Dasari & Gupta (2020) suggest using demonstration videos from human operators, denoted as $v$, as contexts for corresponding tasks. They train the policy $\pi(a_t|s_t, \cdots, s_{t-k}, v)$ using Eq. (81), supplemented by an inverse model auxiliary loss term. **(2) Transformer Adapter** (Liang et al. (2022)) presents an approach for pretraining and fine-tuning. Initially, a transformer policy $\pi(a_t|s_{t-k}, a_{t-k}, \cdots, s_{t-1}, a_{t-1}, s_t)$, without task contexts, is pretrained on extensive multi-task demonstrations using Eq. (81). It is then fine-tuned with a limited amount of task-specific demonstrations before application. For efficient fine-tuning, the pretrained model's parameters are kept unchanged. Instead, lightweight adapters (Houlsby et al. (2019)) are introduced between the pretrained model's layers, with only these adapters being updated using the task-specific demonstrations. The structure

| Algorithm | Key Novelty | Evaluation Task |
|---|---|---|
| BeT | Dual-head predictions for the action center and residual action, seperately. | CARLA, Block Push, Franka Kitchen |
| TSE | A transformer-based policy for end-to-end self-driving. | SMARTS |
| Silver-Bullet -3D | A transformer-based control policy for complex manipulation skills with a robotic arm. | ManiSkill |
| VIOLA | Adopting the transformer to identify task-relevant objects in a multi-object environment. | Sorting, Stacking, BUDS-Kitchen, real-world tasks |
| TDT | Language-conditioned IL with the transformer. | Atari Frostbite |
| Perceiver-Actor | Applying the Perceiver Transformer to manage long input sequences in Language-conditioned IL. | RLBench, real-world tasks |
| MARVAL | Segmenting the input into 4 modalities and applying various auxiliary tasks for more efficient use of the demonstrations. | Matterport, Gibson |
| Lang | Improving long-horizon imitation by representation learning based on instruction predictions. | BabyAI, Crafting, ALFREAD |
| Transformer Adapter | Introducing Adapters to pretrained transformers for lightweight task-specific fine-tuning. | MetaWorld |
| DualMind | A dual-phase training framework for generalist agents. | MetaWorld, Habitat |
| MIA | Training an interactive agent by imitation of human-human interactions and self-supervision via modality matching. | 3D Playhouse |

Table 9: Summary of transformer-based IL algorithms. Representative (but not all) algorithms in this section with their key novelties and evaluation tasks are listed in this table. These algorithms are all grounded in the Behavioral Cloning framework, hence the algorithmic and theoretical advancements in this section are relatively modest. However, the learned transformer-based agents are evaluated on much more diverse, realistic, and challenging tasks. First, there are some commonly-used benchmarks. CARLA (Dosovitskiy et al. (2017)) is a simulated environment for self-driving tasks. Block Push (Florence et al. (2021)), Franka Kitchen (Gupta et al. (2019a)), RLBench (James et al. (2020)), MetaWorld (Yu et al. (2019c)) provide a range of robotic manipulation tasks, featuring various types of robots and input modalities. Habitat (Ramakrishnan et al. (2021)) provides navigation tasks for embodied AI in large scale 3D environments. BabyAI (Chevalier-Boisvert et al. (2019)), Crafting (Chen et al. (2021c)), ALFREAD (Shridhar et al. (2020)) offer instruction-conditioned tasks and datasets. Second, some algorithms have shown superior performance on competitions. In the table, SMARTS refers to the NeurIPS 2022 Driving SMARTS Competition (Rasouli et al. (2022)), and ManiSkill refers to SAPIEN ManiSkill Challenge 2021 (Mu et al. (2021)). Third, in the table, VIOLA, TDT, MARVAL, and MIA each develop their own environments or create large-scale training datasets tailored to their specific purposes. These efforts can be regarded as significant contributions alongside their algorithm designs. Readers are encouraged to read the respective papers for details.

of the adapter can be $\text{Adapter}(X) = X + W_2^A(\text{GeLU}(W_1^A(X)))$, where GeLU (Hendrycks & Gimpel (2023)) is the activation function and $W_{1,2}^A$ are the weight matrices of the adapter, and it is inserted between the point-wise FFN layer and Add & Normalization layer of the transformer (as shown in Fig. 1). The intuition behind this paradigm is that agents can acquire a diverse set of behavioral priors through large-scale task-agnostic pretraining, which then enables them to be efficiently fine-tuned for specific tasks. **(3)** As introduced in Section 6.2.4, the ultimate goal of multi-task learning is a generalist agent, i.e., one model with a single set of weights that can be directly applied to a wide range of tasks. Different from Gato, which is directly trained on batches of prompt-conditioned demonstrations, **DualMind** (Wei et al. (2023)) introduces a dual-phase method. Specifically, in Phase I, the transformer policy $\pi(a_t|s_{t-k}, a_{t-k}, \cdots, s_{t-1}, a_{t-1}, s_t)$ is trained with self-supervised learning objectives to capture generic information of state-action transitions, where the self-supervision involves predicting the next state, predicting the current action, and reconstructing masked actions based on the other elements. In Phase II, a prompt $p$, which can be either language or image tokens,

is added as a condition to the policy, i.e., $\pi(a_t|p, s_{t-k}, a_{t-k}, \cdots, s_{t-1}, a_{t-1}, s_t)$. To achieve this, extra Cross Attention layers are incorporated into the transformer decoder, enabling state/action tokens to attend to the prompt tokens. A small fraction of the transformer, including the Cross Attention layers, would then be trained via the prompt-conditioned IL, which is similar with TDT, using expert trajectories with associated prompts. This method emulates how humans learn to act in the world, i.e., acquiring skills in a task-agnostic manner and then learning specific tasks based on the acquired knowledge, and achieves superior performance in a series of robotic manipulation and navigation tasks. Abramson et al. (2021) propose **MIA**, a multi-modal interactive agent that can naturally interact and communicate with humans. The agent can be simply trained by imitation learning of human-human interactions and self-supervised learning through an auxiliary task. Specifically, a multi-modal transformer is used to extract representations from the vision and language input tokens (i.e., $s_t^V, s_t^L$) and is trained through the following objective:

$$\max_{\pi, f, D} \mathbb{E}_{D_E} \left[ \sum_{t=0}^{T} \log \pi(a_t|f(s_t^V, s_t^L), \cdots, f(s_0^V, s_0^L)) + \lambda \sum_{t=0}^{T} \left[ \log D(f(s_t^V, s_t^L)) + \log(1 - D(f(s_t^V, \tilde{s}_t^L))) \right] \right] \quad (82)$$

Here, $f$ denotes the transformer to extract representations, $\pi$ is the overall policy which comprises multiple components besides $f$, $D$ is a discriminator (as in GANs) which is introduced to encourage $f$ to generate distinct representations for matched vision and text data (i.e., $(s_t^V, s_t^L)$) and unmatched ones (i.e., $(s_t^V, \tilde{s}_t^L)$). This modality matching auxiliary task, also used in (Mees et al. (2022a)), has shown to significantly improve the agent's performance.

We summarize the representative algorithms discussed in this section in Table 9, highlighting their key novelties and evaluation tasks. As mentioned in the beginning of this section, transformer-based IL algorithms fundamentally rely on the straightforward BC framework. However, these algorithms have demonstrated promising outcomes in complex robotic manipulation tasks, both simulated and real-world. This leads to an inspiring paradigm for robust robotic learning, that is, using a potent foundation model trained by imitation learning on extensive, high-quality demonstrations. Future research directions may include developing methods to synthesize high-quality training data at lower costs or enhancing/replacing the foundational algorithm BC with more advanced alternatives.

# 7 Diffusion Models in Offline Policy Learning

In this section, we present applications of Diffusion Models (DM) in IL (i.e., Section 7.2) and offline RL (i.e., Section 7.3). In particular, there is an emerging body of researches on DM-based offline RL algorithms, while more extensions/explorations could be made for DM-based IL.

## 7.1 Background on Diffusion Models

Diffusion models have demonstrated superior performance across multiple domains, such as computer vision (Amit et al., 2021; Baranchuk et al., 2021), natural language processing (Austin et al., 2021b; Hoogeboom et al., 2021), and multi-modal learning (Avrahami et al., 2022; Ramesh et al., 2022), showcasing their impressive capabilities in generating detailed and diverse instances. Diffusion models contain two interconnected processes: the forward diffusion process and the backward denoising process (Yang et al. (2022a)). Specifically, the forward process is predefined and transforms the data with a certain (but unknown) distribution, i.e., $x \sim P_X(\cdot)$, into a (standard Gaussian) random noise $z$. This process progressively corrupts the input data by adding a varying scale of noise at each diffusion step. Correspondingly, the reverse process uses a neural network as the denoising function to gradually undo the forward transformation to reconstruct the data $x$ from the random noise $z$. There exist three main formulations of diffusion models: Score-based Generative Models (SGM) (Song & Ermon, 2019; 2020), Denoised Diffusion Probabilistic Models (DDPM) (Sohl-Dickstein et al., 2015; Ho et al., 2020; Nichol & Dhariwal, 2021), and Stochastic-Differential-Equations-based models (Score SDE) (Song et al., 2020; 2021). Next, we introduce these formulations by illustrating their forward/backward processes and learning objectives, while discussing their connections with each other.

- **SGM** learns (Stein) score functions (Hyvärinen (2005)) for samples at each diffusion step, i.e., $S_\theta(x_t, t)$, $t \in [1, \cdots, T]$. As defined, the Stein score function of $x$ is the gradient of its log likelihood

$\nabla_x \log P(x)$, with which data samples $x \sim P(\cdot)$ can be generated via various efficient score-based sampling schemes such as (Song & Ermon (2019); Jolicoeur-Martineau et al. (2021)). Specifically, the forward process of SGM starts from $x_0 \sim P_X(\cdot)$ and injects intensifying Gaussian noise to generate $x_{1:T}$: ($\beta_{1:T}$ is a predefined schedule of variance levels.)

$$x_t \sim F(\cdot|x_0) = \mathcal{N}(x_0, \beta_t I), \ 0 < \beta_1 < \beta_2 < \cdots < \beta_T \tag{83}$$

The score functions for each step $S_\theta(x_t, t)$ are trained to approximate $\nabla_{x_t} \log P(x_t)$, where $P(x_t) = \int F(x_t|x_0)P_X(x_0)dx_0$. In particular, the training objective is as below:

$$\min_\theta \mathbb{E}_{t \sim \mathcal{U}[1,T], x_0 \sim P_X(\cdot), \epsilon \sim \mathcal{N}(0,I)} \left[ \lambda(t)||\epsilon + \sqrt{\beta_t}S_\theta(x_t, t)||^2 \right] \tag{84}$$

Here, $\mathcal{U}[1, T]$ is a uniform distribution on $[1, \cdots, T]$, $\lambda(t)$'s are positive weighting functions. This objective is derived based on the definition of the forward process, i.e., $x_t = x_0 + \sqrt{\beta_t}\epsilon$, $\epsilon \sim \mathcal{N}(0, I)$. For detailed derivations and the definition of $\lambda(t)$, please refer to (Song & Ermon (2019)). Regarding the backward generation process, the score functions $S_\theta(x_t, t)$, $t \in [T, \cdots, 1]$, are sequentially used as denoising functions to generate $x_{t-1}$ from $x_t$. Finally, the data samples $x_0 \sim P_X(\cdot)$ can be acquired. As mentioned, multiple score-based sampling schemes can be adopted in this process. We take the annealed Langevin dynamics sampling scheme (Song & Ermon (2019)) as an example: ($x_T = x_T^0$, $x_0 = x_0^0$)

$$x_t^{i+1} = x_t^i + \frac{1}{2}s_t S_\theta(x_t^i, t) + \sqrt{s_t}\epsilon, \ (i = 0, \cdots, N-1), \ x_{t-1}^0 = x_t^N \tag{85}$$

$N$ and $s_t$ are hyperparameters, denoting the number of iterations and step size for denoising $x_t^0$ to $x_{t-1}^0$ $(t = T \to 1)$.

- **DDPM** gradually adds random noise to the data over a series of time steps $(x_0, \cdots, x_T)$ in the forward process, where $x_0 = x$, $x_T = z$. In particular, the sample at each time step is drawn from a Gaussian distribution conditioned on the sample from the previous time step: ($\beta_{1:T}$ are predefined.)

$$x_t \sim F(\cdot|x_{t-1}) = \mathcal{N}(\sqrt{1-\beta_t}x_{t-1}, \beta_t I) \tag{86}$$

With Eq. (86), the sample at each step $t$ can be expressed as a function of $x_0$: $x_t = \sqrt{\alpha_t}x_0 + \sqrt{1-\alpha_t}\epsilon$, where $\alpha_t = \prod_{s=0}^t (1 - \beta_s)$, $\epsilon \sim \mathcal{N}(0, I)$ (Sohl-Dickstein et al. (2015)). $\alpha_T$ is designed to be close to 0, so that $z = x_T$ approximately follows $\mathcal{N}(0, I)$. Conversely, through the reverse denoising process, $x_T$ is converted back to $x_0$ step by step: $(t = T \to 1)$

$$x_{t-1} \sim G_\theta(\cdot|x_t) = \mathcal{N}(\mu_\theta(x_t, t), \Sigma_\theta(x_t, t)) \tag{87}$$

where $\mu_\theta$ and $\Sigma_\theta$ can be implemented as neural networks. The objective for learning this denoising function is to match the joint distributions of $x_{0:T}$ in the forward and backward processes, i.e., $\min_\theta D_{KL}(F(x_0, \cdots, x_T)||G_\theta(x_0, \cdots, x_T))$, which is equivalent to: (Sohl-Dickstein et al., 2015)

$$\min_\theta \mathbb{E}_{x_0 \sim P_X(\cdot), x_{1:T} \sim F(\cdot|x_0)} \left[ -\log P(x_T) - \sum_{t=1}^T \log \frac{G_\theta(x_{t-1}|x_t)}{F(x_t|x_{t-1})} \right] \tag{88}$$

Note that $P(x_T)$ and $F(x_t|x_{t-1})$ have analytical forms and this objective is an upper bound for the negative log-likelihood $\mathbb{E}_{x_0 \sim P_X(\cdot)}[-\log P_\theta(x_0)]$. Although this objective can be directly optimized through Monte Carlo sampling, Ho et al. (2020) propose a reformulation of it for variance reduction. Since all (forward or backward) transformations are based on Gaussian distributions, by specifying the variance schedule $\beta_{1:T}$ and fixing the backward variance $\Sigma_\theta(x_t, t)$ to be $\beta_t I$, Eq. (88) can be converted to: (Ho et al. (2020))

$$\min_\theta \mathbb{E}_{t \sim \mathcal{U}[1,T], x_0 \sim P_X(\cdot), \epsilon \sim \mathcal{N}(0,I)} \left[ \lambda(t)||\epsilon - \epsilon_\theta(\sqrt{\alpha_t}x_0 + \sqrt{1-\alpha_t}\epsilon, t)||^2 \right] \tag{89}$$

Here, $\lambda(t)$ is a positive weighting function which has a closed-form, but in practice, $\lambda(t)$ is set as 1 for all $t$ to improve sample quality. Intuitively, the denoising function $\epsilon_\theta$ is trained to predict the noise injected to samples at each step. Also, by setting $\epsilon_\theta(x,t) = -\sqrt{\beta_t}S_\theta(x,t)$, the objective forms of SGM (Eq. (84)) and DDPM (Eq. (89)) can be unified. Finally, with the learned $\epsilon_\theta$, the sampling process $x_{t-1} \sim G_\theta(\cdot|x_t)$ is equivalent to: $x_{t-1} = \frac{1}{\sqrt{1-\beta_t}}(x_t - \frac{\beta_t}{\sqrt{1-\alpha_t}}\epsilon_\theta(x_t,t)) + \sqrt{\beta_t}\epsilon, \ \epsilon \sim \mathcal{N}(0,I)$.

- **Score SDE** extends the discrete-time schemes of the previous two methods to a unified continuous-time framework, building upon stochastic differential equations (SDE). In the forward process, it perturbs data to noise with a diffusion process governed by the following differential equation:

$$dx = f(x,t)\,dt + g(t)\,dw, \ t \in [0,T] \tag{90}$$

where $w$ is the standard Wiener process (Ricciardi (1976)), $f(x,t)$, $g(t)$ are the diffusion and drift functions, respectively. In particular, the forward process of DDPM and SGM can be described by the following two equations (Song et al. (2020)), respectively, with specified $f(x,t)$ and $g(t)$:

$$dx = -\frac{1}{2}\beta(t)x\,dt + \sqrt{\beta(t)}\,dw, \ dx = \sqrt{\frac{d\beta(t)}{dt}}\,dw \tag{91}$$

The reverse process can be realized via solving the reverse-time SDE (Anderson, 1982):

$$dx = [f(x,t) - g^2(t)\nabla_x \log P_t(x)]\,dt + g(t)\,d\bar{w} \tag{92}$$

where $\bar{w}$ denotes the standard Wiener process when time flows backwards, $P_t(x)$ denotes the distribution of $x$ at time $t$ in the forward process. Similarly with SGM, the score function at each time $\nabla_{x_t} \log P_t(x_t)$ can be estimated with an NN-based function $S_\theta(x_t,t)$ through various score matching techniques (Vincent (2011); Song et al. (2019)). Further, in (Song et al. (2020)), they propose a more computationally-efficient denosing process by removing the random noise injection $g(t)\,d\bar{w}$ and prove that the resulting equation is a probability flow ordinary differential equation (ODE) which shares the same marginal densities as those of the reverse-time SDE. Both equations allow sampling from the required data distribution, i.e., $P_X(x)$.

## 7.2 Diffusion Models in Imitation Learning

Most works regarding applying diffusion models in IL are based on the Behavioral Cloning (BC) framework (Pomerleau (1991)), which is introduced in Section 2.2.1. Diffusion models have shown superior performance in modeling complex, high-dimensional data distributions. Thus, many recent works have proposed to improve BC by implementing the policy network as a diffusion model to model the conditional distribution $P_{A|S}(a|s)$ within the expert data $D_E$. Further, as shown in Section 7.1, the objective for training DM (i.e., Eq. (88)) provides an upper bound for the negative log-likelihood (i.e., the objective of BC), which naturally connects DM with BC. Next, we introduce these works in details, which are designed to address drawbacks of the original BC algorithm from multiple perspectives.

### 7.2.1 Policy Approximation using Diffusion Models

The first group of works (Pearce et al. (2023); Chi et al. (2023); Reuss et al. (2023)) try to improve the expressiveness of the learned policy. The BC policy $\pi_\theta$ is trained to give out point estimates [17], which precludes it from capturing the multi-modality of the state-action pairs in $D_E$. Moreover, $\pi_\theta$ is encouraged to learn an average distribution, as the objective (i.e., Eq. (11)) is to maximize an expectation, which would result in bias towards more frequently occurring actions. Additionally, $\pi_\theta(a|s)$ is usually implemented as a diagonal Gaussian policy, thus the prediction of each action dimension is independent, potentially leading to uncoordinated behaviours in high-dimensional action spaces. On the other hand, DM has shown great potential in generating high-dimensional samples that adhere to complex, multi-modal distributions, while ensuring the precision and diversity. In this case, **Diffusion BC** (Pearce et al. (2023)) is proposed to utilize

---

[17]Each data point corresponds to a state-action pair $(s,a)$.

the denoising (score) function $\epsilon_\theta$ in DDPM (Ho et al. (2020)) to generate actions $a$ at given states $s$. Similarly with Eq. (89), the IL objective is as follows: (Please refer to Section 7.1 for notation definitions.)

$$\min_\theta \mathbb{E}_{t \sim \mathcal{U}[1,T],(s,a) \sim D_E, \epsilon \sim \mathcal{N}(0,I)} \left[ ||\epsilon - \epsilon_\theta(a_t = \sqrt{\alpha_t}a + \sqrt{1 - \alpha_t}\epsilon, t; s)||^2 \right] \tag{93}$$

Note that $s$ is the conditioner which is not interrupted in the diffusion process. With the learned $\epsilon_\theta$, the estimated expert action $a = a_0$ can be generated following the denoising process in DDPM:

$$a_{t-1} = \frac{1}{\sqrt{1 - \beta_t}}(a_t - \frac{\beta_t}{\sqrt{1 - \alpha_t}}\epsilon_\theta(a_t, t; s)) + \sqrt{\beta_t}\epsilon, \ t = T, \cdots, 1, \ a_T \sim \mathcal{N}(0, I) \tag{94}$$

Further, **Diffusion Policy** (Chi et al. (2023)) is proposed to utilize DDPM for closed-loop action-sequence prediction. In particular, it models the distribution $P_{\vec{A}|\vec{S}}(\vec{a}|\vec{s})$ in $D_E$, where $\vec{s}$ and $\vec{a}$ denote the previous $T_s$ states and future $T_p$ actions, respectively. $T_a$ out of the $T_p$ actions are executed without replanning. This framework allows long-horizon planning, encourages temporal consistency of the action sequence, and remains responsive to the changing environment through receding horizon planning. In order to learn the joint distribution of the $T_p$ actions conditioned on the historical states – a task that is inherently high-dimensional – Chi et al. (2023) suggest modeling it as $\epsilon_\theta(\vec{a}_t, t|\vec{s}_t)$, which is trained and sampled with a DDPM framework as delineated in Eq. (93) and (94).

Last, Reuss et al. (2023) propose **BESO** for goal-conditioned IL. Specifically, one or more future states within the same trajectory as $(s, a)$ are used as the goal $g$. The objective is now to get a goal-conditioned policy $\pi_\theta(a|s, g)$, which is learned as a conditional score function $S_\theta(a_t, t; s, g)$ in a Score SDE framework (Karras et al. (2022)). With $S_\theta(a_t, t; s, g)$, the action at $s$ targeting $g$ can be generated with a Probability Flow ODE, as introduced in Section 7.1. Goal-conditioned policies distill useful, goal-oriented behaviors, which can be integrated with a high-level planner for various downstream tasks, as in hierarchical RL. However, incorporating the goal conditioner amplifies the multimodal nature of the demonstrations, since the same goal might be achieved via various distinct trajectories, underscoring the necessity of employing DM. It's worthy noting that all three works suggest using transformers as the denoising function (i.e., $\epsilon_\theta, S_\theta$) for sample quality, rather than the commonly-used U-Nets (Ronneberger et al. (2015)) for DM.

### 7.2.2 Addressing Common Issues of Imitation Learning with Diffusion Models

Beyond enhancing policy expressiveness, numerous studies have explored the use of DM to address other challenges in IL, including spurious correlations (Saxena et al. (2023)) and imperfect demonstrations (Wang et al. (2023e); Yuan et al. (2023b)). To be specific, due to spurious correlations, expressive models, such as DM, may focus on distractors that are irrelevant to action prediction and thus fragile in real-world deployment, which resembles the causal misidentification issue introduced in Section 3.3.5. Thus, Saxena et al. (2023) propose **C3DM** to improve the denoising process of Diffusion BC as follows:

$$s_t = C(s; \text{pos}(a_t), t), \ a_{t-1} = \frac{1}{\sqrt{1 - \beta_t}}(a_t - \frac{\beta_t}{\sqrt{1 - \alpha_t}}\epsilon_\theta(a_t, t; s_t)) + \sqrt{\beta_t}\epsilon, \ t = T, \cdots, 1, \ a_T \sim \mathcal{N}(0, I) \tag{95}$$

Compared with Eq. (94), the only difference is to replace the conditioner $s$ with $s_t$ which is updated with the denoising process. In particular, at each denosing iteration $t$, the image state $s$ is zoomed into the region around the intermediate action, i.e., $\text{pos}(a_t)$, to acquire more details for decision-making while ignoring distractors in other regions. Note that this work assume the access to a transformation between the action and state spaces to determine $\text{pos}(a_t)$. Accordingly, they modify the training process in Eq. (93) by replacing $s$ with $C(s; \text{pos}(a), t)$. Further, considering that expert demonstrations in real-world scenarios are often noisy, Wang et al. (2023e) propose **DP-IL** to adopt DM for purifying/denoising the demonstrations. Subsequently, any IL algorithm can be applied to these purified data. To be specific, based on the DDPM framework, they view $(s, a)$ as the data point $x$ (in Eq. (89)) and learn a denoising function $\epsilon_\theta(x_t, t)$ to recover the original state-action demonstrations (i.e., $x_0$) from samples interrupted with Gaussian noise (i.e., $x_t$). Note that this training process is based on a set of "clean" demonstartions. Then, for imperfect data points $\tilde{x}$, the learned DM can be used to purify them by first adding random noise to $\tilde{x}$ to get $\tilde{x}_t$ (i.e., via the forward process) and then using $\epsilon_\theta$ in the reverse denoising process to get the purified data $\tilde{x}_0$. Theoretically, they prove that $\tilde{x}_t$

| Algorithm | DM Type | DM Usage | IL Framework | Targeted IL Issues | Evaluation Task |
|---|---|---|---|---|---|
| Diffusion BC | DDPM | Policy Generator | BC | Multimodal distribution | Kitchen, CSGO |
| Diffusion Policy | DDPM | Policy Generator | BC | Multimodal Distribution & HD data | Kitchen, Push-T, Block-Push, Robomimic |
| BESO | Score SDE | Policy Generator | BC | Multimodal distribution | Kitchen, Block-Push, CALVIN |
| C3DM | DDPM | Policy Generator | BC | Spurious correlation | Autodesk, Real Robot |
| DP-IL | DDPM | Data Denoiser | Not limited | Noisy data | MuJoCo |
| SMILE | DDPM | Policy Generator | BC | Noisy data | MuJoCo |

Table 10: Summary of DM-based IL algorithms. Specifically, "HD" is short for "high-dimensional". Regarding the evaluation tasks, Kitchen (Gupta et al., 2019b), Block-Push & Push-T (Florence et al., 2022), Robomimic (Mandlekar et al., 2021b), CALVIN (Mees et al., 2022b) are robotic manipulation tasks in different scenarios; CSGO (Pearce & Zhu, 2022) is a 3D First-person Shooter video game; Autodesk (Koga et al., 2022) and Real Robot (Saxena et al., 2023) represent robotic manipulation tasks built on the Autodesk simulator and real robot platforms, respectively; MuJoCo (Todorov et al. (2012)) provides a series of robotic locomotion tasks.

and $x_t$ can be arbitrarily closed in distribution as $t$ increases, and so $\epsilon_\theta$ trained on $x_t$ can be used to convert $\tilde{x}_t$ into purified demonstrations as well. Different from the two-stage framework in DP-IL, **SMILE** (Yuan et al. (2023b)) is proposed to jointly perform the automatic filtering of noisy demonstrations and learning of the expert policy. It is based on Diffusion BC (i.e., Eq. (93) and (94)), but changes the forward diffusion process of DDPM as follows:

$$a_t \sim \pi^t(\cdot|s), \ \pi^t(a_t|s) = \int \pi^0(a_0|s)\mathcal{N}(a_t|a_0, \sigma_t^2 I) \ da_0, \ \sigma_t^2 = \sum_{k=1}^{t} \beta_k^2 \tag{96}$$

Here, $\pi^0$ denotes the underlying policy of the provided demonstrations. Through this equation, the diffusion process of SMILE is conducted on a policy level rather than on data points, and so the learned denoising function $\epsilon_\theta$ can be used to compare the suboptimality among behavior policies underlying different trajectories (i.e., with Eq. (13) in (Yuan et al. (2023b))). Trajectories generated using policies noisier than the currently learned one are then filtered out from the training dataset.

While not exclusively designed for DM-based IL, the studies by Shi et al. (2023) and Sridhar et al. (2023) introduce techniques — namely, **AWE** and **MCNN** — that further enhance the performance of Diffusion Policy and Diffusion BC, respectively. Both techniques focus on mitigating the compounding errors of IL, i.e., prediction errors compounded over the decision horizon. AWE can be applied to automatically extract waypoints within an expert trajectory. The waypoint sequences are notably shorter in horizon and can be used for waypoint-only imitation. On the other hand, MCNN does not modify the training data but enhances the policy for reduced compounding errors. During evaluation, it identifies the nearest neighbor state of the current state in the expert dataset, and then blends the corresponding expert action with the output of the learned policy network to formulate an error-constrained policy.

In Table 10, we present a summary of DM-based IL methods. This includes their base IL algorithms, the specific IL challenges they aim to address, and how DM is used in resolving these issues. There are several promising future research directions in this field. Firstly, while most current works rely on DDPM due to its robust performance, future studies could explore more advanced DMs, particularly those with

efficient sampling schemes which are crucial for sequential decision-making [18]. Secondly, given that these works target various IL issues, the development of a unified (DM-based) algorithm capable of simultaneously resolving multiple issues of IL (which are well summarized in Section 2.3) is also a promising direction. Finally, integrating DM with more advanced IL frameworks, as listed in Table 6, could offer solutions to the fundamental issues of BC.

### 7.3 Diffusion Models in Offline Reinforcement Learning

Recently, there is an emerging body of advancements in applying diffusion models to offline RL (Zhu et al. (2023c)). In particular, diffusion models can be adopted as the policy, planner, or data synthesizer in the context of offline RL, as introduced in Section 7.3.1 - 7.3.3. Additionally, we present the applications of DM for extended offline RL setups in Section 7.3.4. For each category, we introduce the seminal works in details as a tutorial on its paradigm, followed by a brief review of the extensions with a focus on their key novelties.

### 7.3.1 Diffusion Models as Policies

As introduced in Section 2.1.1, in order to alleviate the value overestimation issue caused by OOD actions, four offline RL schemes are developed. Among them, DM-based offline RL usually follows the policy constraint method, that is, constraining the learned policy $\pi_\theta$ to the behavior policy $\mu$ while maximizing the Q-function:

$$\max_\theta \mathbb{E}_{s \sim D_\mu} \left[ \mathbb{E}_{a \sim \pi_{\theta(\cdot|s)}} Q_\phi(s,a) - \lambda D_{KL}(\pi_\theta(\cdot|s)||\mu(\cdot|s)) \right] \tag{97}$$

where $Q_\phi$ is a learned Q-function for the current policy $\pi_\theta$, and $\lambda > 0$ is the Lagrangian multiplier [19]. Such an optimization problem has a closed-form solution, i.e., $\pi^*(a|s) = \frac{1}{Z(s)}\mu(a|s)\exp(\frac{1}{\lambda}Q_\phi(s,a))$, where $Z(s)$ is the partition function for normalization. Suppose that $\pi^*$ can be represented as a parametric function, then the optimal policy can be learned through weighted regression (Wang et al. (2020)) as below:

$$\max_\theta \mathbb{E}_{(s,a) \sim D_\mu} \left[ \frac{1}{Z(s)} \log \pi_\theta(a|s) \exp(\frac{1}{\lambda}Q_\phi(s,a)) \right] \tag{98}$$

Therefore, the parametric policy $\pi_\theta$ should be expressive enough to recover the optimal policy.

In this case, DM can be employed to model the policy $\pi_\theta$. **(1) SfBC** (Chen et al. (2023b)) provides a straightforward manner to realize this. It first imitates the behavior policy $\mu(a|s)$ with a DM $\pi_\theta(a|s)$ (specifically, Score SDE) in an IL scheme as introduced in Section 7.2.1. Then, for any state $s$, $N$ actions are sampled with $\pi_\theta(\cdot|s)$ as candidates, and one action is resampled from these candidates with $\exp(\frac{1}{\lambda}Q_\phi(s,a))$ being the sampling weights. In this way, $a \sim \pi^*(\cdot|s)$ is approximated via importance sampling. Note that $Q_\phi(s,a)$ can be learned with any offline RL protocol. **IDQL** (Hansen-Estruch et al. (2023)) adopts the same resampling scheme but imitates $\mu(a|s)$ with DDPM. Additionally, rather than using $\exp(\frac{1}{\lambda}Q_\phi(s,a))$, they design a sampling weight based on the Implicit Q-Learning framework (Kostrikov et al. (2022)), which is a SOTA offline RL algorithm. **(2)** Alternatively, the Q-function $Q_\phi(s,a)$ can be directly involved in training the DM policy $\pi_\theta(a|s)$. In particular, **Diffusion-QL** (Wang et al. (2023f)) learns a DDPM-based policy through the following objective:

$$\min_\theta \mathcal{L}_{DM}(\theta) - \frac{\eta}{\mathbb{E}_{(s,a) \sim D_\mu}[|Q_\phi(s,a)|]} \mathbb{E}_{s \sim D_\mu, a_0 \sim \pi_\theta(\cdot|s)}[Q_\phi(s,a_0)] \tag{99}$$

Here, $\mathcal{L}_{DM}(\theta)$ is defined as Eq. (93) (i.e., an IL objective) and can be replaced by the corresponding training objective if a different type of DM is employed, $\pi_\theta$ is implied by the denoising function $\epsilon_\theta$, $a_0$ is the action sample after being denoised for $T$ iterations (with Eq. (94)), $\mathbb{E}_{(s,a) \sim D_\mu}[|Q_\phi(s,a)|]$ is incorporated for adaption to Q-functions with different scales. Intuitively, the Q-function acts as guidance in the reverse generation process of the DM. Also, Eq. (99) mirrors Eq. (97), since both equations encourage the learned policy to maximize Q-values while being closed to the behavior policy, and it's noteworthy that **DiffCPS** (He et al. (2023b)) provides a theoretical connection between Eq. (99) and Eq. (97).

---

[18] At each time step, to sample $a \sim \pi_\theta(\cdot|s)$, the entire denoising process (with $T$ iterations) shown as Eq. (94) needs to be executed, thus the sampling efficiency of DM is essential.

[19] As a common practice, $\lambda$ can be fine-tuned as a hyperparameter, controlling the tradeoff between the two objective terms.

Following Diffusion-QL, several algorithms adopting a similar objective design with Eq. (99) have been proposed, including SRDP (Ada et al. (2023)), EDP (Kang et al. (2023)), and Consistency-AC (Ding & Jin (2023)). To be specific, **SRDP** additionally introduces a state-reconstruction loss term in Eq. (99) to enable generalization for OOD states through self-supervised learning (Liu et al. (2023a)). **EDP** improves the sampling efficiency of Diffusion-QL by modifying the sampling process $a_0 \sim \pi_\theta(\cdot|s)$ in Eq. (97). Instead of iterative sampling with Eq. (94), it first samples $a_t$ via the forward process and then approximates $a_0$ as $\frac{1}{\sqrt{\alpha_t}}a_t - \frac{\sqrt{1-\alpha_t}}{\sqrt{\alpha_t}}\epsilon_\theta(a_t, t; s)$, since $a_t = \sqrt{\alpha_t}a_0 + \sqrt{1-\alpha_t}\epsilon$ and $\epsilon_\theta$ is trained to estimate $\epsilon$. Similarly, as a new variant of DM, Consistency Models (Song et al. (2023)) aim to learn a consistency function $f_\theta$ that can map the noisy sample at any iteration $t$ to its original sample in just one iteration. **Consistency-AC** thus replaces DDPM in Diffusion-QL with a Consistency Model to improve the sampling efficiency.

At last, there are two works: QGPO (Lu et al. (2023)) and CPQL (Chen et al. (2023f)) aiming to directly learn the score function of the optimal policy: (This is derived from the definition of $\pi^*$, where $\alpha = 1/\lambda$.)

$$\nabla_{a_t} \log \pi^*(a_t|s) \propto \nabla_{a_t} \log \mu(a_t|s) + \nabla_{a_t} \alpha Q_\phi(s, a_t) \tag{100}$$

As mentioned in Section 7.1, with the score function $\nabla_{a_t} \log \pi^*(a_t|s)$, samples $a_t \sim \pi^*(\cdot|s)$ can be generated via various efficient score-based sampling schemes [20]. Here, the score function $\nabla_{a_t} \log \mu(a_t|s)$ can be learned with Score-based DM from $D_\mu$, which is introduced in Section 7.1, but $\nabla_{a_t} \alpha Q_\phi(s, a_t)$ relies on the Q-functions over intermediate samples $a_t$, which is intractable. As potential solutions, **QGPO** propose a contrastive learning framework to estimate $\alpha Q_\phi(s, a_t)$, $\forall t$. While, **CPQL** employs Consistency Models as the policy, reframes the score function estimation in Eq. (100) into an objective similar to Eq. (99), and theoretically establishes their equivalence when the weights of the two terms in Eq. (99) are properly chosen.

In Table 11, we provide a summary of these algorithms. All the algorithms in this category have been evaluated on the D4RL benchmark. Readers can refer to corresponding papers for the numeric evaluation results to compare their performance. In particular, we notice that three schemes of using diffusion models as policies for offline RL have been developed: SfBC & IDQL, QGPO & CPQL, and the other works that follow Diffusion-QL. Among them, QGPO & CPQL choose to directly learn the score function corresponding to the optimal policy, which is a quite challenging but promising direction for future works.

### 7.3.2 Diffusion Models as Planners

For model-based learning, a dynamic function $\mathcal{T}_\psi(s^{k+1}|s^k, a^k)$ needs to be approximated from the dataset $D_\mu$ first, and then planning over the optimal action sequence $(a_*^1, \cdots, a_*^K)$ can be done by maximizing the return while avoiding OOD actions, of which the objective is shown as Eq. (9). Note that, for clarity, we use superscript $k$ to indicate the time step and subscript $t$ to denote the iteration of the diffusion process. **SGP** (Suh et al. (2023)) proposes to implement the uncertainty measure $u(s, a)$ (in Eq. (9)) as the negative log-likelihood of $(s, a)$ under a perturbed distribution, i.e., $-\log P_\sigma((s, a); D_\mu)$ where $P_\sigma(x; D_\mu) = \frac{1}{|D_\mu|}\sum_{x_i \in D_\mu} \mathcal{N}(x; x_i, \sigma^2 I)$. Intuitively, this likelihood measures the distance of $(s, a)$ to the dataset $D_\mu$ and a large distance indicates a high uncertainty of the point $(s, a)$ as it may be OOD. Then, to optimize Eq. (9) for trajectory planning, the score function $\nabla_a \log P_\sigma((s, a); D_\mu)$ needs to be estimated. As detailed in (Suh et al. (2023)), this estimation can be obtained as a denoising function in DM (specifically, SGM).

A more widely-adopted manner for planning with DM, which is firstly proposed in **Diffuser** (Janner et al. (2022)), is to fold the two processes mentioned above: transition dynamic modeling and trajectory optimization regarding $a^{1:K}$, as a trajectory modeling process using DM. Viewing trajectories $\tau = ((s^1, a^1), \cdots, (s^K, a^K))$ as data points (i.e., $x$ in Eq. (89)), DDPM is used to model the distribution of $\tau$ in $D_\mu$, through Eq. (89) where a denoising function for trajectory generation $\epsilon_\theta(\tau_t, t)$ is learned. Then, the planning can be done by sampling trajectories starting from the current state using the learned DM. The sampling process is similar with Eq. (94). However, as an RL algorithm, this process is guided by a (separately-trained) return function of the trajectory samples, i.e., $J_\phi(\hat{\tau})$ [21]: ($\beta_t$, $\alpha_t$, and $\Sigma_t$ are hyperpa-

---

[20]Note that $a_t$ represents the sample at the $t$-th diffusion iteration rather than the action at time step $t$.

[21]$J_\phi(\hat{\tau})$ is trained to estimate the return of the original trajectory, i.e., $J(\tau) = \sum_{k=1}^{K} r(s^k, a^k)$, where $\hat{\tau}$ can be the trajectory sample at any diffusion iteration, i.e., $\tau_t$.

rameters and defined in Section 7.1.)

$$\tau_{t-1} \sim G_\theta(\cdot|\tau_t, g) = \mathcal{N}(\mu_t + \Sigma_t g, \Sigma_t), \ \mu_t = \frac{1}{\sqrt{1-\beta_t}}(\tau_t - \frac{\beta_t}{\sqrt{1-\alpha_t}}\epsilon_\theta(\tau_t, t)), \ g = \nabla_{\hat{\tau}} J_\phi(\hat{\tau})|_{\hat{\tau}=\mu_t} \qquad (101)$$

This process is also known as **classifier-guided sampling** (CG (Dhariwal & Nichol (2021))), which is widely adopted for conditional generations with diffusion models. Intuitively, the generation is guided by the gradient $\nabla_\tau J(\tau)$ along which the expected return $J(\tau)$ would be maximized, aligning it with RL. In (Janner et al. (2022)), the authors connect this guided sampling design with the control-as-inference framework of RL (Levine (2018)) and claim that trajectories from such a generation process follows the distribution $P(\tau|\mathcal{O}_{1:K} = 1)$, where $\mathcal{O}_k = 1$ indicates the optimality of the time step $k$. During evaluation, the learned DM can be applied as follows: ($\mathcal{T}$ is the real dynamic.)

$$[\tau_t(s^1) \leftarrow s^k, \ \tau_{t-1} \sim G_\theta(\cdot|\tau_t, g)]_{t=T}^1, \ a^k \leftarrow \tau_0(a^1), \ s^{k+1} \sim \mathcal{T}(\cdot|s^k, a^k), \ k = 1, \cdots, K \qquad (102)$$

To determine $a^k$, trajectory samples at each iteration are forced to start with the current state, i.e., $\tau_t(s^1) \leftarrow s^k$ $(t = 1, \cdots, T)$. This follows the idea for solving inpainting problems (Sohl-Dickstein et al. (2015)), where the generation for the unobserved part is in a manner consistent with the observed constraints. Moreover, only the first action in the generated plan, i.e., $\tau_0(a^1)$, is executed without replanning, which aligns with the receding horizon control (Mayne & Michalska (1988)). **Decision Diffuser** (Ajay et al. (2023)) adopts similar designs with two key modifications. First, instead of training a return function for classifier-guided sampling, they adopt **classifier-free guided sampling** (CFG (Ho & Salimans (2022))), for which a conditional and an unconditional denoising function, i.e., $\epsilon_\theta(\tau_t, t; J(\tau))$ and $\epsilon_\theta(\tau_t, t; \emptyset)$, are jointly trained (with Eq. (89)) by randomly dropping out the conditioner $J(\tau)$. The trajectory generation process is the same as Eq. (94), but replaces the standard denoising function $\epsilon_\theta(\tau_t, t)$ with $\omega\epsilon_\theta(\tau_t, t; J(\tau)) + (1-\omega)\epsilon_\theta(\tau_t, t; \emptyset)$. Increasing the value of $\omega$ would decrease the diversity of samples but aligns the trajectory distribution more closely with $P(\tau|\mathcal{O}_{1:K} = 1)$. Notably, both CG and CFG can be utilized for generating samples that satisfy specific conditions $y$. In the context of RL, $y$ can be the desired return $J(\tau)$, constraints to obey, goals or subgoals to achieve (for multi-task or hierarchical RL), and so on. Second, only state sequences are modelled and predicted with the DM, i.e, $\tau = (s^1, \cdots, s^K)$ and actions are predicted with a separate inverse dynamic model, i.e., $a^k = \mathcal{T}_{\text{inv}}(s^k, s^{k+1})$. This is because sequences over actions tend to be more high-frequency and less smooth, making them much harder to predict and model.

Following Diffuser and Decision Diffuser, improvements have been made in multiple aspects such as the sampling process (SafeDiffuser, Discrete Diffuser), network structure (EDGI), training objective (PlanCP), and conditioners (TCD). We present a brief overview of these advancements in comparison to Diffuser and Decision Diffuser as follows. Notably, only TCD follows Decision Diffuser and the others follow Diffuser. **SafeDiffuser** (Xiao et al. (2023)) aims to ensure the safe generation of data in the diffusion process. For each iteration $t$, the diffusion dynamic $u_t = \frac{\tau_{t-1} - \tau_t}{\Delta t}$ is re-optimized to satisfy certain safety constraints and the resulting dynamic $u_t^*$ is used to update $\tau_{t-1}$ as $\tau_t + \Delta t * u_t^*$. $\Delta t$ is the diffusion time interval which should be small enough. They theoretically show that $\tau_0$ generated in this manner fulfills the safety constraints with probability almost 1. **Discrete Diffuser** (Coleman et al. (2023)) proposes three diffusion-guidance sampling techniques for generation in discrete state and action spaces, where Gaussian-based DM cannot be applied. These guided sampling methods can work with discrete DM such as D3PM (Austin et al. (2021a)). **EDGI** (Brehmer et al. (2023)) proposes an equivariant network architecture for the scenario where an embodied agent operates $n$ objects in a 3D environment. This network design takes geometric structures into account and ensures equal likelihood of a trajectory and its counterpart for which specific spatial/temporal translations or permutations over objects are applied. Better generalization across scenarios, where such translations or permutations happen, can thus be realized. **PlanCP** (Sun et al. (2023)) proposes to quantify the uncertainty of DM using Conformal Prediction (Vovk et al. (2005)). In particular, $M$ trajectories are generated from DM corresponding to $M$ trajectories in $D_\mu$, each pair of which has a prediction error $e_i$. The conformity among $\{e_i\}$ reflects the uncertainty of the DM. Introducing this conformity term to the DDPM training objective (i.e., Eq. (89)) as an auxiliary term can potentially reduce the uncertainty of sampling with DM. Compared with Decision Diffuser, **TCD** (Hu et al. (2023a)) utilizes more temporal information as the conditioner of the denoising function $\epsilon_\theta$ to guide the sampling process. Specifically, following the insight of Decision Transformer (introduced in Section 6.2.1), the return-to-go and reward at the current time step

are involved as the prospective and immediate conditions, respectively, to reason about the sampling progress while focusing more closely on the current generation step.

### 7.3.3 Diffusion Models as Data Synthesizers

The performance of offline RL is limited by the provided static dataset $D_\mu$. Once trained with $D_\mu$, the DM can potentially generate a variety of high-quality trajectory data. This newly generated data can then be employed to augment $D_\mu$, thereby creating a feedback loop to further improve the DM. Moreover, in an unseen but related task, the DM's inherent generality allows for generation of well-performing trajectories for fine-tuning the model in this novel environment. Several works have been developed in this direction following either Diffuser (Hwang et al. (2023); Liang et al. (2023b)) or Decision Diffuser (He et al. (2023a)). Specifically, **AdaptDiffuser** (Liang et al. (2023b)) suggests a rule-based method for filtering and enhancing generated trajectories $\tau = ((s^1, a^1), \cdots, (s^K, a^K))$ from Diffuser, based on the return values and dynamic consistency. Starting with $k = 1$, a revised action $\tilde{a}^k$ is determined using a well-trained or defined inverse dynamic model $\mathcal{T}_{\text{inv}}(\tilde{s}^k, s^{k+1})$ (with $\tilde{s}^1 = s^1$). Subsequently, the next state is adjusted according to the (learned) environment dynamic function as $\tilde{s}^{k+1} = \mathcal{T}(\tilde{s}^k, \tilde{a}^k)$. Trajectories with states $s^k$ significantly deviating from their counterparts $\tilde{s}^k$ are filtered out. The remaining trajectories $\tilde{\tau}$ are further filtered based on their returns $J_\phi(\tilde{\tau}) = \sum_{k=1}^K r_\phi(s^k, a^k)$. Finally, the chosen trajectories are of high quality and can be used for effective data augmentation. In a more straightforward approach, **MTDiff** (He et al. (2023a)) models trajectories $\tau = ((s^1, a^1, r^1), \cdots, (s^K, a^K, r^K))$ conditioned on expert demonstrations from the same environment, denoted as $y$. Then, trajectory samples from the learned conditional denoising function $\epsilon_\theta(\tau_t, t; y)$ are directly used for data enhancement. For unseen tasks, a small amount of demonstrations can be used as prompts, i.e., $y$, for $\epsilon_\theta(\tau_t, t; y)$ to generate high-quality training data. These data can then be employed to adapt the DM policy to novel tasks.

### 7.3.4 Diffusion Models in Extended Offline Reinforcement Learning Setups

Diffusion models have been applied in multi-task, multi-agent, and hierarchical offline RL setups. Still, these methods are built on the foundational algorithms mentioned above, such as Diffusion-QL, Diffuser, and Decision Diffuser. MTDiff (He et al. (2023a)) and MetaDiffuser (Ni et al. (2023)) aim to learn task-conditioned planners to solve a distribution of tasks. They utilize offline data categorized by task, i.e., $\cup_i D_{\mu_i}$, where $\mu_i$ denotes the behavior policy of task $i$. Following Decision Diffuser, **MTDiff** incorporates an expert trajectory from task $i$, denoted as $y_i$, as an extra condition of the planner specific to that task, represented as $\epsilon_\theta(\tau_t, t; J(\tau), y_i)$. Through training, the agent is expected to implicitly capture the transition model and reward function stored in the prompt trajectory $y_i$, and perform task-specific planning based on these internalized knowledge. When being evaluated in a new but related task, the denoising function conditioned on a corresponding demonstration $y_i$ can be directly used for planning. In contrast, **MetaDiffuser** explicitly models the reward and transition dynamics by learning an encoder, denoted as $E_\eta$, to encode trajectories into compact representations. The encoder is jointly trained with the approximate reward and dynamic models, which serve as the decoder, within a VAE framework, resembling BoReL introduced in Section 3.2.4:

$$\max_{\eta, \psi, \phi} \mathbb{E}_{\tau \sim \cup_i D_{\mu_i}, y = E_\eta(\tau), (s^k, a^k, r^k, s^{k+1}) \sim \tau} \left[\log \mathcal{T}_\psi(s^{k+1}|s^k, a^k, y) + \log r_\phi(r^k|s^k, a^k, y)\right] \tag{103}$$

In this way, $y$ is trained to embed task-specific reward and dynamic functions. When planning for a specific task $i$, the representation $y_i = E_\eta(\tau_i)$, $\tau_i \sim D_{\mu_i}$ can be used as an extra condition of the planner. Notably, MetaDiffuser is based on Diffuser and it adopts the learned reward and dynamic models to define the sampling guidance $g$ in Eq. (101) to enhance the dynamics consistency of the generated trajectories while encouraging a high return, detailed as Eq. (7) in (Ni et al. (2023)).

DOM2 (Li et al. (2023d)) and MADiff (Zhu et al. (2023b)) target at fully cooperative, multi-agent offline RL. Built upon Diffusion-QL, **DOM2** learns a Q-function and DM-based policy for each agent following a fully decentralized framework. This policy is tailored to map the individual observation of each agent to its own action, which is trained with Eq. (99). Following Decision Diffuser, **MADiff** is developed for planning over the joint trajectories of all agents. Each agent $i$ has a separate denoising function $\epsilon_\theta^i$ for planning its corresponding trajectory. To encourage global information interchange for better coordination, the denoising

| Algorithm | DM Type | DM Usage | Evaluation Task |
|---|---|---|---|
| SfBC | Score SDE | Policy | D4RL (L, M, K) |
| IDQL | DDPM | Policy | D4RL (L, M) |
| Diffusion-QL | DDPM | Policy | D4RL (L, M, A, K) |
| SRDP | DDPM | Policy | D4RL (L, M) |
| DiffCPS | DDPM | Policy | D4RL (L, M) |
| EDP | DDPM | Policy | D4RL (L, M, A, K) |
| Consistency-AC | Consistency Model | Policy | D4RL (L, M, A, K) |
| QGPO | Score-SDE | Policy | D4RL (L, M) |
| CPQL | Consistency Model | Policy | D4RL (L, A), dm_control |
| DOM2 (MARL) | Score-SDE | Policy | MPE, MAMuJoCO |
| LDCQ (HRL) | DDPM | Policy | D4RL (L, M, A, K, C) |
| SGP | SGM | Planner | CartPole, D4RL (L), Pixel-Based Single Integrator, Box-Pushing |
| Diffuser | DDPM | Planner | D4RL (L, M), Kuka Robot |
| Decision Diffuser | DDPM | Planner | D4RL (L, K), Kuka Robot, Unitree-go-running |
| SafeDiffuser | DDPM | Planner (D) | D4RL (L, M) |
| Discrete Diffuser | D3PM | Planner (D) | D4RL (L) |
| EDGI | DDPM | Planner (D) | Kuka Robot |
| PlanCP | DDPM | Planner (D) | D4RL (L, M) |
| TCD | DDPM | Planner (DD) | D4RL (L) |
| MetaDiffuser (MTRL) | DDPM | Planner (D) | Multi-task MuJoCo |
| MADiff (MARL) | DDPM | Planner (DD) | MPE, SMAC, MATP |
| HDMI (HRL) | DDPM | Planner (DD) | D4RL (L, M), NeoRL |
| AdaptDiffuser | DDPM | Planner & Synthesizer (D) | D4RL (L, M), Kuka Robot |
| MTDiff (MTRL) | DDPM | Planner & Synthesizer (DD) | Meta-World-V2 |

Table 11: Summary of DM-based offline RL algorithms. Some algorithms in this table can be categorized as multi-agent RL (MARL), multi-task RL (MTRL), or hierarchical RL (HRL). When DM are used as planners, most algorithms can be viewed as extensions of Diffuser (D) or Decision Diffuser (DD). Regarding the benchmarks, nearly all algorithms in this category have been evaluated on D4RL (Fu et al. (2020)), which provides offline datasets for various data-driven RL tasks, including Locomotion (L), AntMaze (M), Adroit (A), Kitchen (K), and CARLA Autonomous Driving (C). dm_control (Tassa et al. (2018)), Kuka Robot (Janner et al. (2022)), Unitree-go-running Margolis & Agrawal (2022), CartPole (Tedrake (2023)), Box-Pushing (Manuelli et al. (2020)) are a series of continuous (robotic) control tasks. Pixel-Based Single Integrator (Chou & Tedrake (2023)) requires dynamic learning and control over the pixel space. NeoRL (Qin et al. (2022)) is an industrial benchmark on financial decision making. Lastly, Multi-task MuJoCo (Mitchell et al. (2021)) and Meta-World-V2 (Yu et al. (2019c)) are widely-used benchmarks for multi-task/meta RL. MPE (2D control Lowe et al. (2017)), MAMuJoCo (robotic locomotion Peng et al. (2021)), SMAC (video gaming Samvelyan et al. (2019)), and MATP (trajectory prediction Alcorn & Nguyen (2021)) provide diverse evaluation tasks for multi-agent RL.

function contains an attention layer to aggregate trajectory embeddings from other agents. Clearly, more extensions can be developed regarding DM-based MARL, such as MARL algorithms for fully competitive or mixed (partially cooperative/competitive) multi-agent tasks, and integration of DM with state-of-the-art centralized training & decentralized execution (CTDE) MARL methods.

LDCQ (Venkatraman et al. (2023)) and HDMI (Li et al. (2023c)) are proposed to learn a hierarchical policy/planner from an offline dataset, which can be especially beneficial for long-horizon decision-making. **LDCQ** learns a high-level policy $\pi_{\text{high}}(z|s)$ for skill selection, and low-level policies $\pi_{\text{low}}(a|s,z)$ for each skill

$z$. In particular, they employ a $\beta$-VAE for skill discovery from the offline dataset $D_\mu$: (This objective is the same as the one of OPAL introduced in Section 3.2.4.)

$$\max_{\phi,\theta} \mathbb{E}_{\tau \sim D_\mu, z \sim P_\phi(\cdot|\tau)} \left[ \sum_{k=1}^{K} \log P_\theta(a^k|s^k, z) - \beta D_{KL}(P_\phi(z|\tau)||\mathrm{prior}(z|s^1)) \right] \tag{104}$$

Similarly with OPAL, after training, $P_\theta$ is used as $\pi_{\mathrm{low}}$ and $P_\phi$ is adopted to convert $D_\mu$ to a set of transitions in the form of $(s^1, z \sim P_\phi(\cdot|\tau), \sum_{k=0}^{K-1} r^k, s^K)$ for training $\pi_{\mathrm{high}}$. $\pi_{\mathrm{high}}$ is modeled as DM and trained in a similar manner with SfBC (introduced in Section 7.3.1). Alternatively, for each trajectory in $D_\mu$, **HDMI** extracts its subgoal list following a heuristic method (Eysenbach et al. (2019a)). Then, two DMs are adopted to model the sequence of subgoals and trajectories targeting at each subgoal, respectively. In particular, based on Decision Diffuser, HDMI learns a high-level planner to generate the subgoal list $\tau_z$, guided by the trajectory return, i.e., $\epsilon_{\mathrm{high}}(\tau_{z,t}, t; J(\tau))$, and a low-level planner to produce the state sequence $\tau_s$ corresponding to each certain subgoal $z$, i.e., $\epsilon_{\mathrm{low}}(\tau_{s,t}, t; z)$.

It's worthy noting that fast sampling is essential for the application of DM for offline RL or IL. Besides adopting Consistency Models, for DDPM-based methods, they usually limit the number of iterations of the reverse generation process to be relatively small and adopt a carefully-designed variance schedule, i.e., $\beta_{0:T}$, as proposed in (Xiao et al. (2022)). For SDE-based methods, they would solve a probability flow ODE for the reverse generation process (as introduced in Section 7.1) with a fast solver: the DPM-solver (Lu et al. (2022a)). All algorithms mentioned in this section have been summarized in Table 11 and categorized based on the DM type and usage. When applying DM as planners, most algorithms are built upon DDPM and follow the design of either Diffuser (D) or Decision Diffuser (DD).

## 8 Discussions and Open Problems

In this section, we provide in-depth discussions on deep generative models (DGMs) in offline policy learning and our perspectives on future research directions of this area, based on the main text in Section 3 - 7. The discussions are centered around how DGMs have been used in offline policy learning, which provide a comprehensive summary of this paper and insightful ideas for future works. Note that, for simplicity, we use abbreviations of the DGMs: VAE - Variational Auto-Encoder, GAN - Generative Adversarial Network, NF - Normalizing Flow, DM - Diffusion Model.

### 8.1 Discussions on Deep Generative Models and Offline Policy Learning

**A common usage of DGMs in offline policy learning:** All DGMs can be utilized as policy functions in offline policy learning. In the context of IL, they can be the student policy learned by imitating the expert, while, in offline RL, they can either be the approximated behavior policy or the RL policy. To be specific, we list their mathematical forms as follows, in the order they were introduced [22]:

$$\begin{aligned}
&\text{VAE: } z \sim P_\theta(\cdot|s), \ a \sim P_\theta(\cdot|s, z); \ \text{GAN: } z \sim P_Z(\cdot), \ a = G(z|s); \\
&\text{NF: } z \sim P_Z(\cdot), \ a = G(z|s), \ G = G_1 \circ \cdots \circ G_N; \ \text{Transformer: } a_t \sim \pi(\cdot|\tau_{t-k:t}); \\
&\text{DM: } a_T \sim \mathcal{N}(0, I), \ a_{t-1} \sim G_\theta(\cdot|a_t, s), \ (t = T, \cdots, 1), \ a = a_0.
\end{aligned} \tag{105}$$

All these DGM-based policies show superior expressiveness compared with the one using only feed-forward neural networks, and each of them has unique advantages/disadvantages: (1) VAEs might be less expressive than the others, but offers a lightweight model choice and relatively stable training process; (2) As mentioned in Section 4.2.4, GAN-based polices can generate sharp data samples but may suffer from the mode collapse issue, meaning that they may not be able to cover the multiple modes in the $(s, a)$ distribution of the dataset

---

[22]$P_\theta(z|s)$ and $P_\theta(a|s, z)$ denote the prior and decoder of the VAE, respectively. $P_Z(z)$ is the assumed prior distribution of the latent variable $z$. $G(z|s)$ denotes the generator in the GAN or NF, and it is composed of a series of invertible and differentiable functions, i.e $G_1 \circ \cdots \circ G_N$, in the NF. $\pi(\cdot|\tau_{t-k:t})$ represents a transformer-based policy, where $\tau_{t-k:t}$ can be $(s_{t-k}, a_{t-k}, \cdots, s_t)$ in IL and $(s_{t-k}, a_{t-k}, R_{t-k}, \cdots, s_t, R_t)$ in offline RL. $G_\theta(a_{t-1}|a_t, s)$ represents the denosing process of DM, and the subscript $t$ denotes the denoising iteration.

like the other DGMs; (3) With its special architecture (e.g., the attention mechanism), the transformer can process time series or data encompassing multiple entities, as mentioned in Section 6.3.2; (4) As for NF and DM, they excel in modelling/generating high-dimensional and complex behaviors, due to their multi-component or multi-iteration designs shown in Eq. (105). As a spotlight, we note that both VAE- and transformer-based policies have been used to make decisions on multi-modal input (e.g., information from different sensors or in different formats like images and languages), but they adopt different strategies. In particular, VAEs try to embed input from different modalities into a unified latent space, and then a single latent-conditioned policy (coupled with encoders for each modality) can be used to predict actions based on multiple types of input, as introduced in Section 3.3.3. Transformers, as a universal foundation model, do not make assumptions on the structure of the input data, and so can be used in various task scenarios such as CV and NLP, as introduced in Section 6.1. Thus, a transformer can directly take a data sequence containing different modalities as input, treat each modality as a separate token, and integrate them via the attention mechanism, as shown in Section 6.3.2.

**The unique usages of each DGM for offline policy learning:** We notice that each DGM has its own unique usage for offline policy learning. (1) VAEs can learn **latent representations** of the data samples. These representations, which are potentially disentangled, can be used as data transformation for learning efficiency (Section 3.2.3), task or subtask representations for multi-task or hierarchical learning (Section 3.2.4), similarity measure of data samples for data augmentation (Section 3.3.2), unified representations of multi-modal input (Section 3.3.3), or disentangled representations of the states for mitigating causal confusion (Section 3.3.5), etc. (2) GANs utilize an adversarial learning framework to practically realize **distribution matching**. Specifically, GANs can be used to minimize the discrepancy between the learned policy and expert policy, i.e., $d(\rho_\pi(s,a)||\rho_{\pi_E}(s,a))$, for IL, or the discrepancy between the learned policy and behavior policy, i.e., $d(\rho_\pi(s,a)||\rho_\mu(s,a))$, to mitigate the OOD issue for offline RL, as introduced in Section 4.2.1 and 4.3.1, respectively. (3) NFs can provide **exact density estimation** due to its special architecture design (i.e., invertiable and differentiable components), enabling them to be incorporated with various advanced IL frameworks, as detailed in Section 5.2.1. We highlight that whenever there is a requirement for exact density estimation, NFs can be utilized. For example, SOPT, as introduced in Section 3.2.2, proposes to approximate the behavior policy's density function, i.e., $\mu(a|s)$, using a VAE to apply support-constraint offline RL. However, the NF is a better choice for this purpose since it can provide more accurate estimations. (4) Due to its ability for **sequence modelling**, transformers enable trajectory-optimization-based offline RL/IL, as introduced in Section 6.2.1 and 6.3.1. Also, as shown in Section 6.3.3, owing to its **high capacity and generalization ability**, it can be used as the core component of a generalist agent to handle a range of tasks with a single set of parameters, and it allows for pretraining and fine-tuning as in CV and NLP. (5) Finally, DM can be used to **purify noisy training data** with its special denoising process, as illustrated in Section 7.2.2, and solve model-based offline RL as **trajectory modelling**, as shown in Section 7.3.2, due to its efficiency in generating high-dimensional data and its special conditional sampling mechanisms (i.e., classifier guided or classifier-free guided sampling).

**Integrated use of different DGMs for offline policy learning:** The unique usages/advantages of each DGM form the basis of an integrated use of different DGMs in offline policy learning, and there are already some works exploring in this direction. Here, we list some (but not all) examples introduced in this paper. (1) In Section 4.2.4, we present a spotlight on integrating VAEs and GANs for IL to synthesize their advantages, i.e., to mitigate the mode collapse issue of GANs and the blurry sample issue of VAEs. (2) In Section 5.3, APAC adopts a Flow-GAN to model the distribution of $(s,a)$ in $D_\mu$, which can be viewed as an integration of NFs and GANs; SAFER adopts a VAE to extract the safety context from the state sequence, which is then used as a condition of an NF-based policy for safe generation. (3) In Section 6.2.3, SPLT shows a coherent use of VAEs and transformers, where the VAE is used to capture the multiple modes within the data/environment as latent variables, which are then used as conditioners of a Decision Transformer to mitigate the impact from environmental stochasticity. (4) MIA, as introduced in Section 6.3.3, adopts a GAN-based auxiliary loss term to improve the data usage in transformer-based IL. (5) In Section 7.2.1, all three DM-based IL works implement their denoising functions as transformers, rather than commonly-used U-nets, for sample quality and policy expressiveness.

| DGM | VAE | GAN | Normalizing Flow | Transformer | Diffusion Model |
|---|---|---|---|---|---|
| IL | BC (VIB), Bayesian IRL | MaxEntIRL (AIL) | KL, LIL, AIL | BC (self-supervision) | BC |
| Offline RL | DP-based offline RL | Model-based offline RL | DP-based offline RL | Trajectory optimization | DP/Model-based offline RL, Trajectory optimization |

Table 12: A summary of the base offline RL/IL algorithms for DGM-based offline policy learning. MaxEntIRL (Section 4.2.1) is short for maximum causal entropy inverse RL. KL, LIL, and AIL (Section 5.2.1) denote the KL-divergence-based, Likelihood-based, and Adversarial IL framework, respectively. DP-based offline RL (Section 2.1.1) refers to the Dynamic-Programming-based offline RL.

**A summary of the base IL/offline RL algorithms for each DGM:** We provide this summary as Table 12. **(1)** Regarding IL, most DGMs, including VAEs, Transformers, and DMs, select BC as the base algorithm, of which the objective (Eq. (11)) is simply to maximize the log likelihood of expert state-action pairs. However, there are differences in the realization. Specifically, as introduced in Section 3.3.1, VAE-based IL algorithms either adopt a VAE ELBO, which is a lower bound of the BC objective, or a VIB-based framework, which is close in form with the ELBO. For transformer-based IL, besides the BC term, to make full use of demonstrations, auxiliary supervision objectives, such as prediction errors on the next state (i.e., the forward model loss) or the intermediate action between two consecutive states (i.e., the inverse model loss), are often employed. As for DM-based IL, the objective for training DM (i.e., Eq. (88)) provides a lower bound of the log-likelihood, which naturally connects DM with BC, as mentioned in the beginning of Section 7.2. On the other hand, VAEs, GANs, and NFs have been integrated with more advanced IL frameworks. As introduced in Section 4.2.1, the fundamental GAN-based IL algorithms are derived from MaxEntIRL and practically implemented as Adversarial IL (AIL) frameworks; NFs, as exact density estimators, have the flexibility to be adopted in various IL frameworks such as KL, LIL, and AIL, as illustrated in Section 5.2.1. In the representative work, AVRIL (introduced in Section 2.2.2), a VAE is employed to scale Bayesian IRL, but this direction remains relatively underexplored. **(2)** Three categories of offline RL algorithms, including dynamic-programming-based, model-based, and trajectory-optimization-based offline RL, are introduced in Section 2.1 as necessary background. The corresponding categories for each DGM are listed in Table 12. More specifically, we note that VAE-based offline RL mainly follows policy penalty, support constraint, and pessimistic value methods, which are subcategories of dynamic-programming-based offline RL; while, DM-based offline RL mainly follows another subcategory of dynamic-programming-based offline RL – policy constraint methods.

**Seminal works of DGM-based offline policy learning:** Regarding the applications in offline policy learning, developments for different DGMs are quite unbalanced, and even for the same type of DGM, the applications in IL may be significantly more than the ones in offline RL, or vice versa. One main factor is whether there are seminal works in that category, since many research works would follow the seminal ones for extensions. Here, we highlight the seminal works introduced in this paper: VAE - offline RL - BCQ (Fujimoto et al. (2019)), GAN - IL - GAIL (Ho & Ermon (2016)) and AIRL (Fu et al. (2017)), Transformer - offline RL - Decision Transformer (Chen et al. (2021b)) and Trajectory Transformer (Janner et al. (2021)), Diffusion Model - offline RL - Diffuser (Janner et al. (2022)) and Decision Diffuser (Ajay et al. (2023)). There are still no seminal works for NF-based offline policy learning, which explains why there are relatively fewer works in Section 5. Notably, some seminal works pioneer new paradigms for offline policy learning. For example, GAIL converts IL/IRL to a distribution matching problem [23]; Decision Transformer realizes offline RL via trajectory optimization, which eliminates the necessity to fit value functions through dynamic programming or to compute policy gradients; Diffuser folds the two processes in model-based offline

---

[23]Although GAIL- or AIRL-based algorithms are used for imitation learning from offline expert data, these algorithms also rely on simulators in their learning process, because there is an inner (online) RL process in their algorithm designs. Similarly, most algorithms within Section 5.2.1 require simulators, as they also realize imitation learning via distribution matching (like GAIL and AIRL), i.e., $\min_\pi d(\rho_\pi(s,a)||\rho_{\pi_E}(s,a))$, and RL is a natural choice for optimization regarding the occupancy measure of the policy, i.e., $\rho_\pi(s,a)$.

| DGM | VAE | GAN | Normalizing Flow | Transformer | Diffusion Model |
|---|---|---|---|---|---|
| IL issues | Insufficient demonstrations, Causal confusion, Multi-modal input | Insufficient demonstrations, Compounding error, Multi-modal demonstrations | Exact density estimation, Learning from observations, Policy expressiveness | Sequential/ multi-modal/ multi-entity input, Multi-modal actions, Insufficient demonstrations | High-dim/ multi-modal state-action, Spurious correlations, Noisy demonstrations, Compounding error |
| IL extensions | Hier | MT, MA, Hier, MB, Safety | N/A | MT, Generalization | Hier |
| Offline RL issues | OOD actions, Low-quality training data, High-dim states, Continuous actions | OOD actions, (Vision-based) world model estimation, Segment stitching | OOD actions, Over conservatism, Insufficient training data, Sparse reward | OOD actions, Insufficient training data, Segment stitching, Sparse reward, Partial observability | OOD actions, Insufficient training data |
| Offline RL extensions | MT, Hier | Hier, MB | Hier, Safety | MT, MA, Hier, MB, Safety, Generalization | MT, MA, Hier, MB, Safety |

Table 13: A summary of the issues and extensions of offline policy learning targeted by the DGMs. MT, MA, Hier, MB represents multi-task, multi-agent, hierarchical, model-based learning, respectively. Safety and generalization refer to if the learned policy can be safely applied to risky environments or be generalizable to unseen environments.

RL: transition dynamic modeling and trajectory optimization, into a trajectory modeling process. These paradigm shifts are closely related to corresponding DGMs and make full use of their unique advantages, which open up promising directions for offline policy learning. On the other hand, simply using DGMs as function estimators in traditional offline RL or IL methods can be less attractive, unless the application of DGMs offers superior scalability or generalization capabilities.

**A summary of the issues and extensions of offline policy learning that have been targeted by the DGMs:** In Table 13, we enumerate the IL/offline RL issues that each DGM has tried to solve, as well as the extended IL/offline RL setups that have been explored by each DGM. Specifically, there are in total six setup extensions, including multi-task (MT), multi-agent (MA), hierarchical (Hier), model-based (MB) learning, policy safety, and policy generalization. Readers can easily find the research works targeting at specific issues/extensions in corresponding sections. However, we note that the listed issues/extensions should not be considered as completely resolved by the DGMs, and future works can focus on the unsolved or underexplored issues/extensions as detailed in Section 8.2.4. Here, we provide some remarks on this table. (1) Low-quality training data is characterized by its limited coverage of the state-action space, a lack of diversity in behavior patterns, or the inclusion of suboptimal/noisy demonstrations. (2) Multi-modal demonstrations/actions refer to demonstrations/actions containing multiple distributional modes, while by multi-modal input, we mean input from multiple sensors or in various formats (e.g., images and texts). (3) Learning from observations refer to imitating from sequences of states only, which is notably more challenging than usual imitation learning. (4) Compared with BC, GAN-based IL can mitigate the issues of insufficient demonstrations and compounding errors. This is because GAN-based IL methods do more than just mimicking observed behaviors – they reason about the underlying reward function from demonstrations and utilizes it for further RL training, which complements the static demonstrations and enables the agent

to act reasonably at unseen states. However, GAN-based IL methods rely on simulators for RL training and suffer from issues regarding sample efficiency and training stability, as introduced in Section 4.2.2. (5) Decision Transformer has lower learning difficulty and is less susceptible to the OOD actions issue compared to traditional offline RL algorithms, because it is based on trajectory optimization, which eliminates the need for dynamic programming or policy gradient calculations, and multiple state and action anchors throughout the trajectory prevent the learned policy from deviating too far from the behavior policy. However, there are still pending issues for Decision Transformer, such as the impacts from environmental stochasticity (Section 6.2.3) and its theoretical shortcomings as listed in Section 6.2.5. (6) All DGMs have been applied to mitigate the OOD actions issue of offline RL. These models (except the transformer) rely on dynamic-programming-based offline RL but employ various methods to approximate the behavior policy, the support of the behavior policy, or the discrepancy between the behavior policy and the learned policy, based on their unique usages as generative models. Readers can refer to Section 3.2.2, 4.3.1, 5.3.1, 7.3.1 for more insights.

**Comparisons on implementation practicality:** Due to the use of different types of DGMs, algorithms reviewed in Section $3-7$ could vary in terms of computation efficiency, implementation difficulty, training stability, and difficulty of hyperparameter tuning. However, there are no empirical studies yet on comparing these aspects across all DGMs, so the following discussions are based on their architectural/objective designs and our own implementation experience, which might be subjective. **(1) Computation efficiency:** VAE-based algorithms (denoted as VAEs for simplicity, and similarly for other DGMs) typically have lower computational overhead due to their simpler architectures. GANs can suffer from high simple complexity, since they adopt an adversarial learning framework and actually solve a minimax optimization problem. NFs involve a series of invertible and differentiable neural network components, which can be computationally intensive. This is especially true when calculating the exact likelihoods, as it requires computation of Jacobian matrices and their determinants. The computation cost for applying transformers is influenced by the number of modules or attention heads used in the encoder and decoder, as illustrated in Figure 1, and the length of the input sequence. Specifically, its time complexity is quadratic with respect to the input length. DMs require a high computational cost due to the iterative nature of their generative and diffusion processes. Notably, certain variants of NFs and DMs need to solve stochastic differential equations (SDEs) or ordinary differential equations (ODEs) during the data generation process. Generally speaking, in terms of computation efficiency, (NFs, Transformers, DMs) < GANs < VAEs, where '<' denotes 'worse than'. **(2) Implementation difficulty:** Both VAEs and GANs consist of only two network components and are relatively easy to implement. Although transformers have complex architectures, most deep learning frameworks, such as TensorFlow (Abadi et al. (2015)) and PyTorch (Paszke et al. (2019)), offer APIs that allow users to directly define a transformer by specifying the number of layers (a.k.a., modules) and attention heads for the encoder and decoder. On the other hand, implementing NFs and DMs can be complicated. NFs utilize invertible and differentiable neural network components, which differ from standard layers such as MLPs or RNNs. Both NFs and DMs involve a normalization direction and a generation direction, and they may require the use of an ODE or SDE solver during the generation process. Thus, for implementation difficulty, (VAEs, GANs, Transformers) < (NFs, DMs). **(3) Training stability:** Both NFs and Transformers utilize supervised learning objectives, which results in relatively stable training processes. VAEs optimize an Evidence Lower Bound (ELBO), like Eq. (17). To avoid potential issues such as posterior collapse (Lucas et al. (2019)), where the latent variable $z$ fails to capture meaningful information about data reconstruction, a careful balance must be maintained between the reconstruction accuracy and the KL divergence term. GANs suffer from low training stability because the discriminator and generator (i.e., policy) networks compete against each other, attempting to reach a Nash equilibrium. This adversarial process is inherently challenging and can lead to issues like mode collapse (Thanh-Tung & Tran (2020)). For DMs, the training stability can vary depending on the specific variant of DM selected, the setup of the noise schedule, and the numerical solver used. Particularly, both the training and generation processes of DMs are sensitive to the configuration of the numerical solver. Thus, their training stability can be ranked as (DMs, GANs) < VAEs < (NFs, Transformers). **(4) Hyperparameter tuning:** The performance of VAEs relies heavily on the weight parameter $\beta$ (as in Eq. (17)), which controls the importance of the regularization term. For GANs, it is crucial to balance the training and convergence rates of the discriminator and policy networks. Typically, the discriminator converges faster, so it should have a larger network structure and a smaller learning rate or fewer updating epochs to maintain this balance. For Transformers, the number of attention heads and layers

should be compatible with the dataset size to avoid overfitting or underfitting, and the sequence length or time window size must be adjusted according to the computation budget. DMs demand careful adjustments in the noise schedule, the number of diffusion steps, and the setup of numerical solvers. Similarly for NFs, if continuous flows are used, specially-designed schemes for hyperparameter tuning may be required (Vidal et al. (2023)). Thus, regarding the difficulty of hyperparameter tuning, (VAEs, Transformers) < GANs < (NFs, DMs).

**Foundation model & Algorithm & Data:** Among all the applications of DGMs, transformer-based IL/offline RL have shown some of the most promising and exciting evaluation results. (1) As listed in Table 9, most algorithms have seen success in challenging robotic simulators, real-world tasks, or large scale competitions. (2) As shown in Section 6.3.3, the use of transformer enables the process of input encompassing extra long time sequences or multiple modalities/entities, and (robotic) agents that are able to follow language instructions or interact with humans can be developed solely by imitation and self-supervision. (3) As introduced in Section 6.2.4, Gato (Reed et al. (2022)), as a single transformer with the same set of weights, can handle 604 distinct tasks with varying modalities (i.e., images, texts, robotic control, and video gaming); further, Reid et al. (2022) and Takagi (2022) explore pretraining the transformer on language tasks and fine-tuning it on RL tasks, and demonstrate promising cross-modality pretraining effects. All these advancements are based on simple supervised learning objectives and are not related to any theoretical breakthrough. However, they do rely on a high-capacity and computation-efficient foundation model – transformer, and a large amount of diverse training data from multiple modalities. This leads to an inspiring paradigm for developing generalist agents, that is, using a potent foundation model trained by (self-) supervised learning from extensive, high-quality demonstrations. It also prompts the question of whether future advancements in artificial general intelligence (AGI) should be driven by data or algorithms. Nevertheless, it is clear that the development of new foundation models or algorithms should prioritize scalability and computational efficiency, key factors in the success of the transformer.

## 8.2 Perspectives on Future Research Directions

Based on the discussions above and the main content in Section 3 - 7, we present some perspectives on future research directions, and we categorize the content in this section as four aspects: data, benchmarking, theories, and algorithms. We believe more open problems can be gleaned from the detailed introductions in the main content, and we have included comments on future works specific to certain DGMs in corresponding sections, such as Section 6.2.5 and the last paragraphs of Section 3, 5, 6.3, 7.2, 7.3.1.

### 8.2.1 Future Works on Data-centric Research

**How would the offline policy learning agent evolve with the quality and quantity of the training data?** The performance of offline policy learning is closely related to the quality and quantity of the provided offline data. Therefore, one potential research direction would be investigating how the dataset's characteristics would impact the learned policy, as instructions for building datasets. (1) First, the static dataset may contain noise (i.e., disturbed or misleading information) or suboptimal behaviors [24]. In this case, open questions include how to measure the noise or suboptimality of the provided data and how robust can the policy learning be to these data imperfections. The answers to these questions would vary with the used algorithms, DGMs, or evaluation tasks, but the imperfection measure and robustness evaluation protocol can be general and beneficial for algorithm development. These questions are important because perfect datasets are costly to build in scale and the tolerance for imperfection can greatly reduce the burden for data collection and process. (2) Second, the performance of an offline policy learning agent on evaluation tasks relies heavily on the coverage and diversity of the training data. An effective set of behaviors for policy learning should cover the possible task scenarios and provide various behavioral patterns/skills for the agent to learn. How to measure the coverage and diversity of the dataset and how these properties would influence the generalization of the learned policy on evaluation tasks would be interesting questions.

---

[24]This would be an issue for IL, since IL requires expert-level demonstartions. However, for offline RL, the dataset could contain suboptimal behaviors as long as their rewards are correctly labeled. If there are only suboptimal trajectories in the dataset for offline RL, the learning difficulty would be high, since the agent needs to learn to stitch segments from various trajectories to form an optimal strategy.

A recent empirical study (Park et al. (2024)) highlights that the generalization of the learned policy to out-of-distribution (OOD) states (Kirk et al. (2023)) is often the main bottleneck in offline RL. Further, they suggest that the most straightforward approach for improvement is to use a training dataset with extensive coverage, and practitioners should prioritize high coverage over high optimality when collecting training data for offline RL. (3) One key factor contributing to the success of certain DGMs in CV and NLP is their ability to leverage extensive training datasets. Notably, their training performance would continue to improve as the quantity of the training data increases. Thus, for each variant of DGM-based offline policy learning, it is essential to study their scalability, namely, how the learned agent evolves with an increasing amount of training data. Algorithms with superior scalability would be more promising in challenging, real-life applications.

**How to construct a training dataset for effective offline policy learning?** With efforts for the open problems mentioned above, we could establish measurements for noise, suboptimality, diversity, and coverage of the dataset and get some insights on the relationship between these measurements and the performance of the learned policy, which can guide the construction of training datasets for effective offline policy learning. However, there are some other problems that need to be explored. First, the original state and action spaces may be high-dimensional and continuous, which would require exponentially more data for policy learning. Utilizing compact representations, which can be obtained from DGMs like VAEs and encoder-only transformers, or discretization could help improve the learning efficiency, but also would introduce inaccuracy. Second, for offline RL datasets, sparse rewards can bring training difficulty but dense rewards are costly to annotate. Thus, a tradeoff needs to be achieved between them and strategically annotating part of the dataset with well-designed rewards could be a promising direction. Similarly, although there are algorithms for learning multi-task, hierarchical, or safe policies from data without the task, skill, or safe action labels, having these labels either fully or partially annotated can significantly aid in the policy learning process. However, given that real-life datasets can be enormous in scale, the labeling should be only for critical points or regions in the state-action space, which necessitates the development of techniques for their identification.

**Data Augmentation & Data Synthesis.** VAEs, transformers, and DMs have been used for data augmentation or synthesis, as shown in Section 3.2.3, 6.2.2, and 7.3.3, respectively, of which the purpose is simply to involve more data for training. The data used for augmentation may already exist but require transformation or supplementation to be effectively utilized, or could be synthesized using DGMs. DGMs exhibit remarkable capability in synthesizing new data that shares statistical properties with the real data. Thus, this approach has the potential to bridge the gap between the demand for large datasets and the feasibility of acquiring them. Multiple research directions in this domain could be developed. First, based on the coverage or diversity indicators of the dataset, targeted data synthesis techniques to supplement the under-covered areas or to avoid repeated behavioral patterns for better diversity could be developed. Second, the synthesized data usually cannot be directly employed without filtering. In this case, necessary quality measure, such as the similarity of the synthesized data with the real (optimal) data, should be established. Ideally, there should be (theoretical) studies on the relationship between the suboptimality of the learned policy and the quality/quantity of the synthesized data. All in all, it would be quite exciting to see the creation of a positive feedback loop: enhanced policies emerging from newly synthesized data, which in turn are leveraged to generate even higher quality data for further training.

### 8.2.2 Future Works on Benchmarking

**Development of more realistic, challenging benchmarks.** We notice that most DGM-based offline RL algorithms are evaluated on D4RL, as shown in the tables of Section 3 - 7, which is a standard benchmark for offline RL. However, widely-used benchmarks like D4RL, cannot fully show or evaluate the advantages of DGM-based offline policy learning algorithms, as we have seen agents, which are implemented as two-layer neural networks and trained with a moderate amount of data, can already achieve excellent performance on these benchmarks. To fully explore the potential and guide the development of DGM-based offline policy learning, more challenging benchmarks are required. (1) First, the benchmark dataset should be large in size to evaluate the scalability of the DGM-based algorithms. This includes evaluating the trend of performance improvement relative to the volume of training data, and determining whether DGM-based algorithms can

significantly outperform others when provided with extensive training data. (2) Second, the dataset should contain multiple modalities. For example, state inputs could be collected from various sources, including language instructions, visual information from different sensors, or even videos; moreover, the state-action space should feature multiple distributional modes, such as containing distinct trajectories for achieving the same goal. DGM-based algorithms are expected to effectively harness information from diverse modalities and learn policies that cover the multiple modes within the policy space. Further, benchmarks for evaluating generalist agents should include a mixture of challenging tasks from different modalities, such as CV, NLP, robotic control, and strategic decision-making. (3) The task scenarios should be more realistic. A main reason that CV and NLP techniques, which are usually DGM-based, can be widely adopted in real life is that these algorithms are directly evaluated on real-life tasks/datasets rather than simplified simulators. Thus, realistic offline dataset in a large scale could greatly advance the development of offline policy learning and showcase the superiority of DGM-based algorithms. (4) The benchmark should evaluate more than just the policy return; it should also assess aspects such as the generalization, safety of the learned policy, and the robustness of policy learning in environments with noise or sparse rewards.

**Comparisons among different DGMs.** Although there are numerous offline RL or IL algorithms for each type of DGMs and they have shared benchmarks, comparisons among them are quite rare. It would be insightful to see fair and thorough comparisons among different DGMs with the same usage (e.g., as policy backbones) on the same set of tasks. Also, as task complexity or dataset size escalates, analyzing the trends in policy performance and computational cost across different DGMs can provide useful insights. Such comparisons provide suggestions on the choice of DGMs, conditioning on the task difficulty and dataset scale. Moreover, as mentioned in the main text of this paper, algorithms that fall in the same category, such as NF-based offline RL algorithms, are also rarely compared with each other. Comparisons within the same category can also be valuable, as they can indicate which usage of a certain DGM is more effective.

### 8.2.3 Future Works on Theories

Most works on applying DGMs in offline policy learning focus on algorithm designs rather than theoretical analysis, so more theoretical research can be done for this field. One promising direction is derivations of the convergence rate or performance guarantee for the seminal works in DGM-based offline policy learning, such as (Zhang et al. (2020b)) for GAIL and (Brandfonbrener et al. (2022)) for return-conditioned supervised learning algorithms like Decision Transformer, based on the theoretical results from either DGMs or offline RL/IL. Moreover, we notice that there have been efforts on theoretically unifying the objectives of different DGMs for improved learning performance and a deeper understanding of the relationship among DGMs, such as (Nielsen et al. (2020); Kingma & Gao (2023)). This group of works can greatly inspire development of new DGM-based offline policy learning algorithms or unified analysis of existing ones.

Next, we outline some specific theoretical problems in this field. (1) First, regarding GAN-based IL (i.e., Section 4.2), so far, no approach has been developed that is both computationally efficient and can theoretically guarantee the recovery of the expert reward function. Also, many works propose to replace the on-policy RL within GAIL/AIRL (i.e., TRPO) with off-policy RL algorithms (e.g., DDPG) for improved sample efficiency. However, this improvement lacks theoretical backup. With off-policy RL, the policy training at each iteration is based on samples from multiple past iterations, during which the reward function keeps changing. As a result, the RL training is conducted in an unstationary MDP where typical RL algorithms would theoretically fail. (2) Second, as mentioned in Section 6.2.5, return-conditioned supervised learning (e.g., Decision Transformer) alone is unlikely to be a general solution for offline RL problems, and it offers guarantees only when the environment dynamics are nearly deterministic. Thus, adapting these algorithms to stochastic environments, ideally backed by theoretical optimality guarantees, is a clear necessity. Also, Decision Transformer would imitate both high-return and low-return trajectories, and claims that the learning performance in this manner is better than imitating high-return trajectories only. A theoretical explanation on how the low-return behavior learning would benefit the overall performance can be insightful. (3) Third, as introduced in Section 7.3.2, the algorithm design of Diffuser is mainly based on intuitions from similar problems, such as the inpainting problem and receding horizon control, but lacks theoretical support. To be specific, in Eq. (102), to decide on the action choice $a$ at state $s$, the trajectory generation is hardcoded to start with $s$, and then the action $\hat{a}$ that is right after $s$ in the generated trajectory is adopted as $a$.

The problem here is how to ensure there is causal relationship between $s$ and $a$, which is crucial for (RL) sequential decision-making, and whether it is necessary to generate an entire trajectory segment in case that only the first action prediction is utilized.

### 8.2.4 Future Works on Algorithm Designs

From the discussions in Section 8.1 and the main content in Section 3 - 7, we can identify many potential future directions regarding the algorithm design. Here, we list some of them as examples.

**Underexplored categories in the main text.** For existing categories of algorithms, some of them are still underexplored, such as GAN-based IL algorithms that do not rely on simulators (Section 4.2), NF-based IL where the NF works as the policy (Section 5.2.2), NF-based offline RL algorithms (Section 5.3.1), efficient algorithms on mitigating impacts from environmental stochasticity for transformer-based offline RL (Section 6.2.3), transformer-based offline RL which integrates traditional offline RL methods with transformer architectures (Section 6.2.5), DM-based offline policy learning where DMs work as data synthesizers (Section 7.3.3), and so on.

**Integrated use of various DGMs for offline policy learning.** As mentioned in Section 8.1, each DGM has distinct usages due to their special design and there have been a few algorithms managing to integrate two DGMs together for improved offline policy learning performance. More in-depth exploration can be done in this direction, especially for involving more DGMs in a unified offline policy learning framework. For example, we observe from Table 12 that most DGM-based IL algorithms are based on BC, which is a relatively basic IL algorithm and has several fundamental issues [25], while NFs are integrated with various advanced IL frameworks due to its ability in exact density estimations. In this case, NFs can be integrated with transformer-based DMs, where NFs extend the base IL framework and transformer-based DMs work as a core component of a generalist agent that can deal with multi-modal input and high-dimensional generations. Further, they can be integrated with VAEs, which can be used to extract task or subtask representations for multi-task or hierarchical learning, as introduced in Section 3.2.4. In addition, we notice that there are some research working on integrating the advantages of various DGMs for computer vision tasks, such as VQ-GAN (VAE+GAN+Transformer, Esser et al. (2021)), TransGAN (GAN+Transformer, Jiang et al. (2021)), DiffuseVAE (VAE+DM, Pandey et al. (2022)), DiTs (Transformer+DM, Peebles & Xie (2023)), which can be potentially utilized in offline policy learning.

**Unsolved issues and extensions of DGM-based offline policy learning.** Table 13 provides a summary of the issues and extensions of IL and offline RL that have been explored by DGM-based algorithms, which provides indications for future works. (1) By checking the outlined challenges in Section 2.3, we can pinpoint unresolved or inadequately-addressed issues by existing methods as future directions. For example, although there have been GAN-based or DM-based algorithms targeting at the compounding error issue, the GAN-based ones rely on simulators and DM-based ones (i.e., AWE and MCNN in Section 7.2.2) are still in the early stages of development. Similarly, in offline RL, reward sparsity is a significant issue, for which the explorations made by NFs (i.e., NF Shaping in Section 5.3.2) and transformers (i.e., DTRD in Section 6.2.2) have tried reward shaping and reward function learning, which are both interesting directions awaiting further development. (2) Regarding the extensions, there can be in total six (or more) setup extensions: multi-task, multi-agent, hierarchical, model-based, safety, and generalization. Table 13 illustrates the extensions that are awaiting exploration for each DGM. The extension method can be potentially transferred among DGMs. For instance, as shown in Section 7.3.4, DM-based multi-task and hierarchical offline RL algorithms adopt the same algorithm ideas as BOReL and OPAL (i.e., the VAE-based methods introduced in Section 3.2.4). Notably, multi-agent extensions for DGM-based offline policy learning require significant development, and existing explorations (such as MAGAIL, MA-AIRL in Section 4.2 and DOM2, MADiff in Section 7.3.4) are still quite immature. In particular, developments of MARL algorithms for fully competitive or mixed

---

[25]As introduced in Section 2.2.1, BC is implemented as supervised learning. However, predictions made by $\pi$, which is learned with BC and used for sequential decision-making, can impact future observations, thus breaching a fundamental assumption of supervised learning (Ross & Bagnell (2010)): training inputs should be drawn from an independent and identically distributed population. Consequently, errors and deviations from the demonstrated behavior tend to accumulate over time, as minor mistakes lead the agent into areas of the observation space that the expert has not ventured into, known as the compounding error. Additionally, BC learns policies merely through imitation, without engaging in reasoning, which restricts the generalization capability of the learned policy.

(partially cooperative/competitive) multi-agent scenarios based on game theory, and integrations of DGMs with state-of-the-art CTDE MARL methods are essential future directions.

**Development of generalist agents.** One of the most exciting future directions for DGM-based offline policy learning is developing a decision-making agent that can continually evolve with the amount of training data & the size of the DGM-based policy network and can handle a range of tasks (including unseen ones). CV and NLP have made great progress in this direction, owing to the use of DGMs. To realize this for offline policy learning, several things should be done. (1) First, the base IL or offline RL algorithm should be compatible with deep and broad neural networks, so that its performance can grow with the size of the foundation model. This is also the reason why mainstream transformer-based and DM-based offline policy learning methods (e.g., Decision Transformer and Diffuser) are based on supervised learning as in CV and NLP. Future works can work on developing deep foundation models that are compatible with policy-gradient methods or dynamic programming. (2) Second, besides constructing large-scale, high-quality training datasets as mentioned in Section 8.2.1, some algorithm-driven efforts can be done. For example, algorithms that are robust to suboptimal training data or capable of processing multi-modal inputs should be developed. Specifically, learning from demonstration videos (in third-person views) is a quite challenging but promising research direction, as large-scale video datasets are more accessible and videos usually contain more learnable information than images or vectorized data. Agents capable of learning from videos have the potential to continuously evolve by observing their surrounding environments during deployment. (3) Third, to enhance generalization, employing advanced meta-learning or multi-task learning techniques is advisable. Adopting the "pretraining + fine-tuning" approach, similar to that used in large language and vision models, is also promising, since it enables the accumulation of training efforts. Notably, the more significant the difference between the source and target tasks, the more challenging it becomes to make pretraining effective, yet the more data can be utilized for training the agent.

**Future works in related areas.** Finally, there are some related areas that remain to be explored. **First**, reviews on the applications of DGMs in Inverse Reinforcement Learning (IRL) or (Online) Reinforcement Learning have not yet been developed. IRL and RL are important approaches for sequential decision-making, but they both require online interactions with the environment. **Second**, as shown in Table 12, DGMs have been applied to dynamic-programming-based, model-based, and trajectory-optimization-based offline RL. There are some other branches in offline RL (Levine et al. (2020)), such as importance-sampling-based and uncertainty-estimation-based offline RL, which have the potential to be enhanced with DGMs. Notably, there is a subclass of dynamic-programming-based offline RL called one-step offline RL (Gülçehre et al. (2021); Brandfonbrener et al. (2021); Li et al. (2023a)). Instead of iteratively executing policy evaluation and policy improvement as in Eq. (1), one-step offline RL starts from learning an approximation of the behavior policy, i.e., $\hat{\mu}$, from the offline dataset $D_\mu$, and its corresponding Q function $Q^{\hat{\mu}}$ (using temporal difference updates). Once the Q learning phase converges, a policy $\pi$ can be learned with certain policy improvement operators (e.g., the second line of Eq. (1)) until convergence, while keeping the Q function $Q^{\hat{\mu}}$ fixed. Since the Q-learning stage is only executed once and does not involve the being-learned policy $\pi$, the issues of OOD behaviors and overestimation (as mentioned in Section 2.1.1) can be strategically avoided. Empirical results from (Brandfonbrener et al. (2021)) show that one-step offline RL outperforms the mainstream iterative offline RL algorithms (introduced in Section 2.1.1) in terms of training stability and policy efficacy, especially when the offline dataset lacks coverage of the state-action space. However, we haven't seen works applying DGMs to one-step offline RL yet, signaling potential areas for future research. Given the reliance of $\pi$'s performance on the accurately approximated $\hat{\beta}$ and its corresponding Q function, DGMs could be used to model the behavior policy, leveraging their expressiveness and ability to handle high-dimensional states & multi-modal actions (as detailed in Section 8.1). DGMs can also be adopted to learn the Q function, as exemplified by Q-transformer (Chebotar et al. (2023)) introduced in Section 6.2.1. Specifically, Li et al. (2023a) propose closed-form policy improvement operators that can be used in one-step offline RL, based on first-order approximations of the policy objective, which offer enhanced stability over gradient descent updates. However, their theoretical analyses assume a Gaussian behavior policy. Extending these closed-form solutions to accommodate non-Gaussian behavior policies would enable the use of more advanced DGMs, such as diffusion models and normalizing flows, as policy approximators. **Third**, we notice that new foundation models and DGMs are continually emerging, such as Mamba (Gu & Dao (2023)) and

Poisson Flows (Xu et al. (2022c; 2023)). Applying these SOTA models in offline policy learning is certainly a promising research direction.

# 9 Conclusion

In this paper, we show a systematic review on the applications of DGMs in offline policy learning. In Section 2, we provide an overview on offline policy learning methods and challenges. Then, in Section 3 - 7, we introduce the applications of each DGM in both offline RL and IL, i.e., the two primary branches of offline policy learning. Each section includes both a tutorial and a survey on the respective topic. Notably, we cover nearly all mainstream DGMs, including Variational Auto-Encoders, Generative Adversarial Networks, Normalizing Flows, Transformers, and Diffusion Models. Following these main content, we provide a summary on existing works and our perspectives on future research directions in Section 8. As the first review paper in this field, we wish this work to be a hands-on reference for better understanding the current research progress regarding DGMs in offline policy learning and help researchers develop improved DGM-based offline RL/IL algorithms.

# Acknowledgement

We would like to thank Hanhan Zhou for participating in discussions and assisting with material collection. We also express our special thanks to the editor and anonymous reviewers of TMLR for their valuable comments, which significantly improved the manuscript.

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
