# OpenReview forum: "Deep Generative Models for Offline Policy Learning: Tutorial, Survey, and Perspectives on Future Directions"
_TMLR — Accepted by TMLR_

### Review · Reviewer_hunT · 2024-04-27

**Summary Of Contributions:**

This review paper is truly a tour de force. It covers applications of 5 different kinds of deep generative models to offline policy learning (including offline reinforcement learning and imitation learning). The paper is a mix of tutorials, reference guides, and systematic paper review. For the generative models, it covers Variational Autoencoders (VAEs) and their variants, Generative Adversarial Networks (GANs), Normalizing Flows (NFs), Transformers, and Diffusion Models (DMs). The paper is structured such that the technical background for all generative models is presented first, and then each paper section is focused on one of these generative models, and its application to offline policy learning. Every section includes a summary table highlighting various aspects of the cited papers within the section. The paper concludes by drawing similarities and differences across different generative model classes in how they are used within offline policy learning, and highlighting areas for future work.

**Audience:**

Yes

**Broader Impact Concerns:**

No concerns about broader impacts.

**Claims And Evidence:**

Yes

**Requested Changes:**

Critical for securing my recommendation
* As noted in the weaknesses section, the paper would be much clearer with a restructuring of how offline policy learning is introduced relative to deep generative models. Concretely, offline policy learning would be introduced as the "Background" section, with the details of each DGM introduced in their respective sections instead.
* In Section 8.1, equation 102 is very difficult to read. It would be helpful to split it into multiple equations for each of VAEs, GANs, etc. listed separately.
* "Regarding IL, most DGMs, including VAEs, GANs, and DMs, select BC as the base algorithm..."
  * This doesn’t match the table – GANs are using MaxEntIRL. Transformers are using BC. Please fix this typo.
* Please make the tables consistent with whether they use the rows or columns to define different algorithms. Currently it is a mix between the two and this made interpreting the tables more complicated than it needed to be.
* Please provide a definition of generative models and give some motivation for why transformers (which are a fundamentally different kind of "generative") are discussed in the context of the review.

Strengthening the work
* In section 8.1, there are two very nice tables (Table 12 and 13) which provide comparisons between DGMs for different offline policy learning setups. Moving these to the background section with offline policy learning would be more useful than having them at the end, as they could signpost for the reader which sections to look at if they want to solve a particular problem in offline policy learning.

**Strengths And Weaknesses:**

Strengths
* This paper is exceptionally comprehensive and current. There is truly impressive coverage of an extremely diverse field which has also seen enormous change in the past 3 years. Given the recent explosion of research papers in this area, review papers like this are critical for distilling and synthesizing insights from the literature. It will be a very helpful reference for new researchers entering this field.
* The authors have given careful thought as to how to present results both comprehensively but also succinctly. The authors have done a nice job of using a few key papers to highlight a broad approach, and then presenting other works as deltas on these few key works. Similarly, for every section, there is a table provided that helps classify and cluster research works together. This helps a reader who may not have time to read the entire paper to pinpoint key works to look for within the section. I particularly appreciated the emphasis on which environments different algorithms were tested on within the tables.
* Several sections are well written, with key insights being clearly elucidated in the text.
* Caveats are clearly outlined for where there is conflicting evidence about the value of a particular approach.
* Concluding sections detailing similarities and differences between generative model classes are well written, and the authors illuminate areas of the field that are under-explored (determined as a result of the literature review).

Weaknesses
* Some of the paper's organization is confusing. In particular, the introduction of the generative modelling methods *before* their respective sections seems not ideal, especially because their notation was not shared. It would be easier for the reader if these techniques were introduced in the sections that they belong to. Instead, the offline policy learning methods could be outlined where the generative models are currently outlined since they are used across many sections. This alternative organization of the paper would make navigating it easier for readers.
  * For example, in the section on policy constraint methods in VAEs, the paper notes "Offline RL algorithms based on diffusion models typically adhere to policy constraint methods. Therefore, we will provide further details on policy constraint methods in Section 7." This is a good example of how a restructuring to put the offline policy method details *first* would help with motivating and signposting how generative models can be used, irrespective of their model class.
  * This restructuring would also help clarify the similarities and differences of how different generative models can be used in similar offline policy learning algorithms. For example, as the paper notes in the concluding sections, all DGMs can be used to represent the policy function. Motivating this as part of the background (by introducing offline policy learning algorithms first) would be helpful, as it provides a framework for the reader to consider the different opportunities for DGMs to play a role.
  * This restructuring would also provide an opportunity to enumerate what the challenges for offline policy learning are which can be addressed by DGMs (actually very nicely summarized in Table 13). This is introduced in individual sections but ends up being very repetitive as the same challenges come up across many sections. Introducing these challenges before going into the specific DGMs would help motivate the rest of the review article.
  * As another example, in "7.2.1 Diffusion Models as Policies", Section 3.1.1 is mentioned (policy constraint methods). The paper states "DM can be employed to model the policy pi_theta". If the policy constraint method was mentioned at the beginning, this would be a more natural comparison, and the paper could include an explanation for why DMs would be preferable to VAEs for modeling pi_theta.

* Generative models are never defined. Four out of 5 models are fundamentally based on sampling from a probabilistic distribution (VAEs, GANs, NFs, DMs) while the fifth, the transformer, is not. As a result, the section on transformers seemed somewhat out of place with respect to the other works, since it is generative in a fundamentally different way (it predicts next tokens based on previous tokens, rather than learning a distribution such that sampling from it gives data points that resemble true data points from training).
* Some sections are not about offline policy learning. Section 6.1.5 in particular focuses on online, partially observable RL. This is not within scope for the paper, and should therefore be removed. Likely the online RL with offline data section of NFs should also be removed, but this is more tangentially related so there could be an argument for keeping it.
* Some terms are not defined. For example "return-to-go" and "MAF" in 5.1.1.

---

> ### Author Response · Authors · 2024-05-28
>
> Thank you for your appreciation and the valuable suggestions. We have restructured the paper and added more explanations and details as requested, which have significantly enhanced our paper. Please download our latest submission and see our responses below.
>
> ## For requested changes:
>
> 1. We have reorganized the details of each DGM and moved them to their respective sections, specifically Sections 3.1, 4.1, 5.1, 6.1, and 7.1. Additionally, we introduced a new section (i.e., Section 2) to provide an overview of offline policy learning methods and challenges. We also swept through the paper to eliminate repetitive content, refined the language use, and added more details to some related works to enhance clarity. We have highlighted significant modifications in blue. Extensively revised sections, such as Sections 2, 4.3.2, and 7.3.4, have their titles marked in blue as well.
>
> 2. We have fixed Eq. (102) as requested, and it is now Eq. (105).
>
> 3. We have corrected the typo. The revised sentence now reads: "Regarding IL, most DGMs, including VAEs, Transformers, and DMs, select BC as the base algorithm."
>
> 4. We have revised Tables 6, 7, and 10 to ensure consistency, with all tables now using rows to define different algorithms.

---

> ### Author Response · Authors · 2024-05-28
>
> ## For requested changes:
>
> 5. We have included definitions of generative models and deep generative models at the beginning of the second paragraph in Section 1, as follows:
>
>    _“Generative models are a class of statistical models that are capable of generating new data instances. They learn the underlying distribution of a dataset and can generate new data points with similar characteristics. Deep generative models (DGMs) are a subset of generative models that leverage the powerful representational capabilities of deep neural networks to model complex distributions and generate high-quality, realistic samples. Notably, the application of DGMs in computer vision (CV) and natural language processing (NLP) has been highly successful.”_
>
>    Regarding the reason to include transformers in our review, we consider the transformer as a representative of autoregressive generative models. Autoregressive models generate future data points based on the previously-generated ones, and are commonly used for generating sequential data such as texts or time series. Other examples of autoregressive models include PixelRNN [1] and PixelCNN for image data. As we have mentioned in the third paragraph of Section 1:
>
>    _“Specifically, the main content is categorized by the type of DGMs, and we cover nearly all mainstream DGMs in this paper, including Variational Auto-Encoders, Generative Adversarial Networks, Normalizing Flows, Transformers, and Diffusion Models, where Transformers and Diffusion Models are representatives of autoregressive and score-based generative models, respectively.”_
>
>    Additionally, transformer-based offline policy learning is an emerging research area, as evidenced by the large volume of the corresponding summarization tables, i.e., Tables 8 and 9. In particular, Decision Transformer is a seminal work of trajectory-optimization-based offline RL, as introduced in Sections 2.1.3 and 6.2.1, and it has become one of the most popular offline RL algorithms today. Further, as discussed in the last paragraph of Section 8.1, among all the applications of DGMs, transformer-based IL/offline RL have shown some of the most promising and exciting evaluation results. For your convenience, we quote that paragraph here.
>
>    _“Foundation model & Algorithm & Data: Among all the applications of DGMs, transformer-based IL/offline RL have shown some of the most promising and exciting evaluation results. (1) As listed in Table 9, most algorithms have seen success in challenging robotic simulators, real-world tasks, or large scale competitions. (2) As shown in Section 6.3.3, the use of transformer enables the process of input encompassing extra long time sequences or multiple modalities/entities, and (robotic) agents that are able to follow language instructions or interact with humans can be developed solely by imitation and self-supervision. (3) As introduced in Section 6.2.4, Gato, as a single transformer with the same set of weights, can handle 604 distinct tasks with varying modalities (i.e., images, texts, robotic control, and video gaming); further, Reid et al. (2022) and Takagi (2022) explore pretraining the transformer on language tasks and fine-tuning it on RL tasks, and demonstrate promising cross-modality pretraining effects. All these advancements are based on simple supervised learning objectives and are not related to any theoretical breakthrough. However, they do rely on a high-capacity and computation-efficient foundation model -- transformer, and a large amount of diverse training data from multiple modalities. This leads to an inspiring paradigm for developing generalist agents, that is, using a potent foundation model trained by (self-) supervised learning from extensive, high-quality demonstrations. It also prompts the question of whether future advancements in artificial general intelligence (AGI) should be driven by data or algorithms. Nevertheless, it is clear that the development of new foundation models or algorithms should prioritize scalability and computational efficiency, key factors in the success of the transformer.”_
>
>    Thus, we believe it is important to retain the content relating to transformers.
>
>    [1] Van Den Oord, Aäron, Nal Kalchbrenner, and Koray Kavukcuoglu. "Pixel recurrent neural networks." In International conference on machine learning, pp. 1747-1756. PMLR, 2016.
>
>    [2] Van den Oord, Aaron, Nal Kalchbrenner, Lasse Espeholt, Oriol Vinyals, and Alex Graves. "Conditional image generation with pixelcnn decoders." Advances in neural information processing systems 29 (2016).

---

> ### Author Response · Authors · 2024-05-28
>
> ## For requested changes:
>
> 6. We believe that it would be beneficial to keep Tables 12 and 13 in Section 8.1, together with corresponding discussions, i.e., the fourth and sixth paragraphs of Section 8.1, for which readers need to go through the main content in Sections 3 – 7 to get a better understanding.
>
>    We appreciate your valuable suggestion and have added Section 2.3, where we thoroughly reviewed the challenges of offline policy learning. Additionally, in the last paragraph of Section 2.3, we directed readers to Tables 12 and 13 for those interested in addressing specific problems in offline policy learning. We quote that paragraph here.
>
>    _“In Section 8.1, we provide an in-depth discussion on offline policy learning and deep generative models (DGMs). In particular, we outline the base IL/offline RL frameworks to which each type of DGM has been applied in Table 12, and the issues or extensions that each DGM has tried to resolve in Table 13, corresponding to the challenges introduced in this section. We encourage readers interested in solving specific problems of offline policy learning to look up these tables and refer directly to the corresponding sections.”_

---

> ### Author Response · Authors · 2024-05-28
>
> ## For weaknesses:
>
> 1. We have restructured the paper as requested. Thanks for your detailed suggestions.
>
> 2. We have included definitions of generative models and explained our motivations to include transformer-related content.
>
> 3. We have included content relating to online RL, such as in Sections 5.3.2 and 6.2.5. Specifically, within Section 6.2.5, we have briefly covered some architectural improvements regarding transformers that were originally proposed for online partially-observable RL. However, these methodologies have the potential to be extended to the offline policy learning setting as future work. That's why we have chosen to retain this content. Here is the modified version of Section 6.2.5:
>
>    _“We notice that there is a series of works on applying transformers to online partially-observable RL. They adopt the transformer as the policy network in gradient-based or actor-critic RL algorithms, with the hope to harness its ability to process long-horizon historical information for decision making. These works focus on architectural modifications to the standard transformer, which are specifically tailored for RL. For example, GTrXL and Catformer suggest adjustments to the MHA module to facilitate a more stable RL training process; ALD adopts model distillation to improve the computation efficiency in transformer-based distributed RL; STT and WMG propose adaptations for the transformer to process spatiotemporal coupling observations and factored observations, respectively, for enhanced sample efficiency. These architectural modifications, or potentially new ones, could be integrated with policy gradient offline RL algorithms or DT-like return-conditioned supervised learning (RCSL) algorithms for performance improvement. Conducting an (empirical) comparison between these two categories (i.e., RCSL and policy gradient offline RL), both utilizing the same transformer architecture, would also be an intriguing study.”_
>
>    In Section 5.3.2, we have introduced some works on online RL that utilize offline data based on Normalizing Flows (NFs). Since NF-based offline RL algorithms, discussed in Section 5.3.1, are relatively scarce—with only three identified—we have slightly broadened the scope to include Section 5.3.2. The works presented in Section 5.3.2 also focus on how to utilize offline data for RL based on NFs, providing valuable insights for NF-based offline RL. For example, CEIP, as discussed in Section 5.3.2, proposes a strategy to leverage offline data from multiple related tasks to enhance data availability.
>
> 4. MAF, referring to Masked Autoregressive Flow, has been added to the revised manuscript. Additionally, the return-to-go $R_t = \sum_{i=t}^{T}$ is defined in the first paragraph of Section 6.2.1.

---

### Review · Reviewer_x4Zp · 2024-05-01

**Summary Of Contributions:**

This paper presents a survey of different deep generative models used for offline policy learning (offline RL and imitation learning). 5 types of models are discussed: VAE, GAN, Normalizing Flows, Transformers and Diffusion models. The paper first gives a background discussion for each of these model families, then for each of them a section is dedicated to discuss the relevant works in offline RL and imitation learning settings. The paper is the first review/survey paper on using deep generative models for offline policy learning, and serves as a tutorial for researchers interested in this topic.

**Audience:**

Yes

**Claims And Evidence:**

Yes

**Requested Changes:**

Would be nice to also have in the discussion section to compare these families of models in aspects such as general computation efficiency, implementation difficulty, training stability, difficult or easy to tune hyperparameters, etc.

**Strengths And Weaknesses:**

Strengths:

- The paper is written and organized clearly and is easy to follow
- The summary tables present a clear overview of different algorithms, the generative model variants that they used, and the evaluation tasks, which is quite useful.
- Discussion section and table 13 provides insight on the main issues of each family of models and their extensions, which is interesting.

Weaknesses:

- As a survey paper, it provides limited technical novelty or results, the main contribution is in providing a tutorial and guidance for researchers interested in the related topics.
- The survey itself does not have major weaknesses.

---

> ### Author Response · Authors · 2024-05-28
>
> Thank you for your valuable suggestions. In this paper, we do not aim to propose new algorithms, but rather to provide a tutorial and comprehensive survey on the applications of deep generative models (DGMs) in offline reinforcement learning and imitation learning, offering perspectives on future work in this area. This is surely a very important topic with a lot of recent advances. As the first review paper in this field, we hope this work will serve as a hands-on reference for better understanding the current research progress regarding DGMs in offline policy learning and assist researchers in developing improved DGM-based offline RL/IL algorithms.
>
> For this revision, we have reorganized the details of each DGM and moved them to their respective sections, specifically Sections 3.1, 4.1, 5.1, 6.1, and 7.1. Additionally, we introduced a new section (i.e., Section 2) to provide an overview of offline policy learning methods and challenges. We also swept through the paper to eliminate repetitive content, refined the language use, and added more details to some related works to enhance clarity. We have highlighted significant modifications in blue. Extensively revised sections, such as Sections 2, 4.3.2, and 7.3.4, have their titles marked in blue as well. Please download our latest submission and see our responses below.
>
> ## For Requested Changes:
>
> As requested, we have added a new paragraph in Section 8.1 (specifically, the second-to-last paragraph) to compare the deep generative models covered in the paper. This comparison addresses aspects such as general computation efficiency, implementation difficulty, training stability, and difficulty in hyperparameter tuning. For your convenience, we quote the paragraph in the next chat box.

---

> ### Author Response · Authors · 2024-05-28
>
> ### The newly added discussions:
>
> _“Comparisons on implementation practicality: Due to the use of different types of DGMs, algorithms reviewed in Section 3 -- 7 could vary in terms of computation efficiency, implementation difficulty, training stability, and difficulty of hyperparameter tuning. However, there are no empirical studies yet on comparing these aspects across all DGMs, so the following discussions are based on their architectural/objective designs and our own implementation experience, which might be subjective. (1) Computation efficiency: VAE-based algorithms (denoted as VAEs for simplicity, and similarly for other DGMs) typically have lower computational overhead due to their simpler architectures. GANs can suffer from high simple complexity, since they adopt an adversarial learning framework and actually solve a minimax optimization problem. NFs involve a series of invertible and differentiable neural network components, which can be computationally intensive. This is especially true when calculating the exact likelihoods, as it requires computation of Jacobian matrices and their determinants. The computation cost for applying transformers is influenced by the number of modules or attention heads used in the encoder and decoder, as illustrated in Figure 1, and the length of the input sequence. Specifically, its time complexity is quadratic with respect to the input length. DMs require a high computational cost due to the iterative nature of their generative and diffusion processes. Notably, certain variants of NFs and DMs need to solve stochastic differential equations (SDEs) or ordinary differential equations (ODEs) during the data generation process. Generally speaking, in terms of computation efficiency, (NFs, Transformers, DMs) < GANs < VAEs, where ‘<' denotes ‘worse than'. (2) Implementation difficulty: Both VAEs and GANs consist of only two network components and are relatively easy to implement. Although transformers have complex architectures, most deep learning frameworks, such as TensorFlow and PyTorch, offer APIs that allow users to directly define a transformer by specifying the number of layers (a.k.a., modules) and attention heads for the encoder and decoder. On the other hand, implementing NFs and DMs can be complicated. NFs utilize invertible and differentiable neural network components, which differ from standard layers such as MLPs or RNNs. Both NFs and DMs involve a normalization direction and a generation direction, and they may require the use of an ODE or SDE solver during the generation process. Thus, for implementation difficulty, (VAEs, GANs, Transformers) < (NFs, DMs). (3) Training stability: Both NFs and Transformers utilize supervised learning objectives, which results in relatively stable training processes. VAEs optimize an Evidence Lower Bound (ELBO), like Eq. (17). To avoid potential issues such as posterior collapse, where the latent variable $z$ fails to capture meaningful information about data reconstruction, a careful balance must be maintained between the reconstruction accuracy and the KL divergence term. GANs suffer from low training stability because the discriminator and generator (i.e., policy) networks compete against each other, attempting to reach a Nash equilibrium. This adversarial process is inherently challenging and can lead to issues like mode collapse. For DMs, the training stability can vary depending on the specific variant of DM selected, the setup of the noise schedule, and the numerical solver used. Particularly, both the training and generation processes of DMs are sensitive to the configuration of the numerical solver. Thus, their training stability can be ranked as (DMs, GANs) < VAEs < (NFs, Transformers).  (4) Hyperparameter tuning: The performance of VAEs relies heavily on the weight parameter $\beta$ (as in Eq. (17)), which controls the importance of the regularization term. For GANs, it is crucial to balance the training and convergence rates of the discriminator and policy networks. Typically, the discriminator converges faster, so it should have a larger network structure and a smaller learning rate or fewer updating epochs to maintain this balance. For Transformers, the number of attention heads and layers should be compatible with the dataset size to avoid overfitting or underfitting, and the sequence length or time window size must be adjusted according to the computation budget. DMs demand careful adjustments in the noise schedule, the number of diffusion steps, and the setup of numerical solvers. Similarly for NFs, if continuous flows are used, specially-designed schemes for hyperparameter tuning may be required. Thus, regarding the difficulty of hyperparameter tuning, (VAEs, Transformers) < GANs < (NFs, DMs).”_

---

### Review · Reviewer_HMCN · 2024-06-05

**Summary Of Contributions:**

This paper provides a thorough review for and topic of generative models + OPL, covers the topic such as variational auto-encoders, generative adversarial networks, normalizing flows, transformers and diffusion models. Concretely, this paper is the first to review Deep Generative Models (DGMs) for Offline Policy Learning, covering a wide range of topics, including five major DGMs and their applications in Offline Reinforcement Learning and Imitation Learning. It distills key algorithmic schemes and paradigms, highlighting seminal research works as tutorials. Additionally, it showcases the evolution of DGM-based offline policy learning in parallel with advancements in generative models and provides valuable insights and future directions in the concluding summary.

**Audience:**

Yes

**Broader Impact Concerns:**

No.

**Claims And Evidence:**

Yes

**Requested Changes:**

I don't have other questions, just answer the question I presented above.

**Strengths And Weaknesses:**

Strengths: The discussion is thorough, and the covered topics are wide, the discussion for future works is also comprehensive.

Weakness: I believe the authors missed one particular important class of algorithms in offline RL, one-step algorithms. For instance, in [1] offline can be formulated using one-step framework with appropriate approximation. What is the potential of one-step algorithm in with deep generative models as approximator? I think the current paper should at least explain it (and other related papers) in the future work section.


[1] Offline Reinforcement Learning with Closed-Form Policy Improvement Operators, ICML23.

---

> ### Author Response · Authors · 2024-06-05
>
> Thank you for the valuable suggestions. We agree that one-step offline RL is an important category of offline RL. We have incorporated a discussion on one-step offline RL and explored the potential application of deep generative models (DGMs) to it in the final paragraph of Section 8.2.4. Please see our latest submission. For your convenience, we quote that paragraph here.
>
> "Notably, there is a subclass of dynamic-programming-based offline RL called one-step offline RL (Gülçehre et al.
> (2021); Brandfonbrener et al. (2021); Li et al. (2023a)). Instead of iteratively executing policy evaluation and policy improvement as in Eq. (1), one-step offline RL starts from learning an approximation of the behavior policy, i.e., $\hat{\mu}$, from the offline dataset $D_\mu$, and its corresponding Q function $Q^{\hat{\mu}}$ (using temporal difference updates).  Once the Q learning phase converges, a policy $\pi$ can be learned with certain policy improvement operators (e.g., the second line of Eq. (1)) until convergence, while keeping the Q function $Q^{\hat{\mu}}$ fixed. Since the Q-learning stage is only executed once and does not involve the being-learned policy $\pi$, the issues of OOD behaviors and overestimation (as mentioned in Section 2.1.1) can be strategically avoided. Empirical results from (Brandfonbrener et al. (2021)) show that one-step offline RL outperforms the mainstream iterative offline RL algorithms (introduced in Section 2.1.1) in terms of training stability and policy efficacy, especially when the offline dataset lacks coverage of the state-action space. However, we haven't seen works applying DGMs to one-step offline RL yet, signaling potential areas for future research. Given the reliance of $\pi$’s performance on the accurately approximated $\hat{\beta}$ and its corresponding Q function, DGMs could be used to model the behavior policy, leveraging their expressiveness and ability to handle high-dimensional states \& multi-modal actions (as detailed in Section 8.1). DGMs can also be adopted to learn the Q function, as exemplified by Q-transformer (Chebotar et al. (2023)) introduced in Section 6.2.1. Specifically, Li et al. (2023a) propose closed-form policy improvement operators that can be used in one-step offline RL, based on first-order approximations of the policy objective, which offer enhanced stability over gradient descent updates. However, their theoretical analyses assume a Gaussian behavior policy. Extending these closed-form solutions to accommodate non-Gaussian behavior policies would enable the use of more advanced DGMs, such as diffusion models and normalizing flows, as policy approximators."

---

### Decision · Action_Editor_vTx6 · 2024-07-27

**Recommendation:** Accept as is

**Comment:**

The reviewers have consensus that this is a well written overview/survey of the topic. The authors engaged during discussion, provided clarifications and substantially revised the draft in response to feedback.

**Audience:**

Yes, the topics are of substantial interest currently across multiple fields (RL, Gen-AI, Robotics, etc), and survey papers should be welcome for consolidation.

**Claims And Evidence:**

This paper provides an extensive survey of the use of generative models for offline policy learning.